# Straggler-Resilient Personalized Federated Learning

**Isidoros Tziotis**                                            *isidoros_13@utexas.edu*
*The University of Texas at Austin*

**Zebang Shen**                                                *zebang.shen@inf.ethz.ch*
*ETH Zurich*

**Ramtin Pedarsani**                                          *ramtin@ece.ucsb.edu*
*The University of California, Santa Barbara*

**Hamed Hassani**                                            *hassani@seas.upenn.edu*
*The University of Pennsylvania*

**Aryan Mokhtari**                                          *mokhtari@austin.utexas.edu*
*The University of Texas at Austin*

**Reviewed on OpenReview:** *https://openreview.net/forum?id=gxEpUFxIgz&referrer=%5BAuthor%20*

## Abstract

*Federated Learning* is an emerging learning paradigm that allows training models from samples distributed across a large network of clients while respecting privacy and communication restrictions. Despite its success, federated learning faces several challenges related to its decentralized nature. In this work, we develop a novel algorithmic procedure with theoretical speedup guarantees that simultaneously handles two of these hurdles, namely (i) *data heterogeneity*, i.e., data distributions can vary substantially across clients, and (ii) *system heterogeneity*, i.e., the computational power of the clients could differ significantly. By leveraging previous works in the realm of representation learning (Collins et al., 2021; Liang et al., 2020), our method constructs a global common representation utilizing the data from all clients. Additionally, it learns a user-specific set of parameters, resulting in a personalized solution for each individual client. Furthermore, it mitigates the effects of stragglers by adaptively selecting clients based on their computational characteristics, thus achieving for the first time near-optimal sample complexity and provable logarithmic speedup. Experimental results support our theoretical findings showing the superiority of our method over alternative personalized federated schemes in system and data heterogeneous environments.

## 1 Introduction

Due to growing concerns on data privacy and communication cost, Federated Learning (FL) has become an emerging learning paradigm as it allows for training machine learning models without collecting local data from the clients. Due to its decentralized nature, a major challenge in designing efficient solvers for FL is heterogeneity of local devices which can be categorized into two different types: (i) *data heterogeneity* where the underlying data distributions of clients vary substantially, and (ii) *system heterogeneity* where the computational and storage capabilities of devices differ significantly. In fact, it has been observed that the seminal Federated Averaging (`FedAvg`) method suffers from slow convergence to a high quality solution when facing highly heterogeneous datasets (McMahan et al., 2017) as well as heterogeneous systems (Li et al., 2020; Kairouz et al., 2021).

In this paper, we aim to address these two challenges simultaneously by introducing a generic framework that includes algorithms with robust performance in the presence of those forms of clients' heterogeneity. Inspired by prior works in the literature of FL (Collins et al., 2021; Liang et al., 2020) that utilized representation

learning theory to tackle data heterogeneity, we propose a meta-algorithm that produces personalized solutions and handles data heterogeneity by leveraging a global representation shared among all clients. Further, our method circumvents the delays introduced due to the presence of stragglers by carefully selecting participating nodes based on their computational speeds. In early stages, only a few of the fastest nodes are chosen to participate and sequentially slower devices are included in the training process until the target accuracy is achieved. Although the disproportional selection of nodes raises fairness and accuracy concerns, we highlight that our method achieves speedup without compromising the resulting solution. The most significant contribution of our work is achieving near-optimal sample complexity in regimes with data and system heterogeneity, alongside a provable logarithmic speedup guarantee in terms of running time. Next we summarize our contributions:

1. **`SRPFL` Algorithm**. We propose Straggler-Resilient Personalized Federated Learning (`SRPFL`), an adaptive node participation meta-algorithm that builds upon subroutines that fall into the representation learning framework (Collins et al., 2021; Liang et al., 2020) enhancing their resilience to stragglers and performance.

2. **Logarithmic Speedup**. Assuming that clients' speeds are drawn from the exponential distribution, we prove that `SRPFL` guarantees logarithmic speedup in the linear representation setting, outperforming established, straggler-prone benchmarks while maintaining the state of the art sample complexity per client $m = \mathcal{O}((d/N + \log(N)))$, where $d$ and $N$ denote the feature vector size and number of active nodes. Our results hold for non-convex loss functions, heterogeneous data and dynamically changing client's speeds.

3. **Numerical Results**. Experiments on various datasets (CIFAR10, CIFAR100, EMNIST, FEMNIST, Sent140) support our theoretical results showing that: (i) `SRPFL` significantly boosts the performance of different subroutines designed for personalized FL both in full and partial participation settings and (ii) `SRPFL` exhibits superior performance in system and data heterogeneous settings compared to state-of-the-art baselines.

## 1.1 Related Work

**Data heterogeneity.** In data heterogeneous settings, if one aims at minimizing the aggregate loss in the network using the classic `FedAvg` method or more advanced algorithms, which utilize control-variate techniques, such as `SCAFFOLD` (Karimireddy et al., 2019), `FEDGATE` (Haddadpour et al., 2021), `FedDyn` (Acar et al., 2021) or `FEDSHUFFLE` (Horváth et al., 2022) the resulting solution could perform poorly for some of the clients. This is an unavoidable hurdle due to the fact that no single model works well for all clients when their underlying data distributions are diverse. A common technique that addresses this issue is fine-tuning the derived global model to each local task by following a few steps of SGD updates (Wang et al., 2019; Yu et al., 2020). Based on this observation, Fallah et al. (2020b) showed that one might need to train models that work well after fine-tuning and showed its connections to Model-Agnostic Meta-Learning (MAML). In (Cho et al., 2022; Balakrishnan et al., 2021) novel client-sampling schemes were explored achieving increased efficiency in regimes with data heterogeneity. Another line of work for personalized FL is learning additive mixtures of local and global models (Deng et al., 2020; Mansour et al., 2020; Hanzely and Richtárik, 2020). These methods learn local models for clients that are close to each other in some norm, an idea closely related to multi-task FL (Smith et al., 2017; Hu et al., 2021). The works in (Chen et al., 2022; Lee et al., 2022) studied the interplay of local and global models utilizing Bayesian hierarchical models and partial participation, respectively. An alternative approach was presented by Collins et al. (2021), where instead of enforcing local models to be close, the authors assumed that models across clients share a common representation. Using this perspective, they presented `FedRep` a method that provably learns this underlying structure in the linear representation setting. Building upon the idea of a common representation Zhu et al. (2021) and Jiang and Lin (2022) proposed federated methods that can handle data heterogeneity while exhibiting robustness to distribution shifts. Recently, a novel framework was proposed allowing the comparison of personalized FL methods under various metrics (Wu et al., 2022). In all of the aforementioned methods however, a subset of clients participate regardless of their computational capabilities. Thus, in the presence of stragglers, the

speed of the training process significantly goes down as the server waits, at every communication round, for the slowest participating node to complete its local updates.

**System heterogeneity.** Several works have attempted to address the issue of system heterogeneity. Specifically, asynchronous methods, which rely on bounded staleness of slow clients, have demonstrated significant gains in distributed data centers (Xie et al., 2019; Stich, 2019; So et al., 2021). In FL frameworks, however, stragglers could be arbitrarily slow casting these methods inefficient. In an attempt to manually control staleness, deadline-based computation has been proposed (Reisizadeh et al., 2019) as well as aggregation of a fixed number of models per round (Nguyen et al., 2022). However, in the worst case scenario the performance of these methods is still determined by the slowest client in the network. Active sampling is another approach where the server aggregates as many local updates as possible within a predefined time span (Nishio and Yonetani, 2019). In a different line of work the effects of stragglers are mitigated by utilizing computation/gradient coding schemes (Tandon et al., 2017; Wang et al., 2018; 2020b). In (Cho et al., 2021) clients use heterogeneous model-architectures to transfer knowledge to nodes with similar data distributions while Yang et al. (2022) proposed a new framework where clients are free to choose their participation scheme. More recently, normalized averaging methods were proposed in (Wang et al., 2020a; Horváth et al., 2022) that rectify the objective inconsistency created by the mismatch in clients' updates. `FedLin` (Mitra et al., 2021) instead utilizes gradient correction and error-feedback mechanisms to circumvent the speed-accuracy conflict. A novel approach to mitigate the effects of stragglers was proposed by Reisizadeh et al. (2022), where clients are selected to take part in different stages of the training according to their computational characteristics. Alas, all of the above methods yield improvement only in data-homogeneous settings and they are not applicable in regimes with data heterogeneity.

## 2 Problem Formulation

In this section, we introduce our setting and define the data and system heterogeneity model that we study. Consider the FL framework where $M$ clients interact with a central server. We focus on a supervised, data-heterogeneous setting where client $i$ draws samples from distribution $\mathcal{D}_i$, potentially with $\mathcal{D}_i \neq \mathcal{D}_j$. Further, consider the learning model of the $i$-th client as $q_i : \mathbb{R}^d \to \mathcal{Y}$ which maps inputs $\mathbf{x}_i \in \mathbb{R}^d$ to predicted labels $q_i(\mathbf{x}_i) \in \mathcal{Y}$. The objective function of client $i$ is defined as $f_i(q_i) := \mathbb{E}_{(\mathbf{x}_i, y_i) \sim \mathcal{D}_i} [\ell(q_i(\mathbf{x}_i), y_i))]$, where the loss $\ell : \mathcal{Y} \times \mathcal{Y} \to \mathbb{R}$ penalizes the gap between the predicted label $q_i(\mathbf{x}_i)$ and true label $y_i$. In the most general setting clients aim to solve

$$\min_{(q_1, \dots, q_M) \in \mathcal{Q}} \quad \frac{1}{M} \sum_{i=1}^{M} f_i(q_i), \tag{1}$$

with $\mathcal{Q}$ the space of feasible tuples of mappings $(q_1, \dots, q_M)$. Traditionally in FL, methods focus on learning a single model $q = q_1 = \dots = q_M$ that performs well on average across clients (Li et al., 2020; McMahan et al., 2017). Although such a solution may be satisfactory in data-homogeneous settings, it leads to undesirable local models when the data distributions are diverse. Indeed, in the presence of data heterogeneity the loss functions $f_i$ have different forms and their minimizers could be far from each other. This justifies the formulation in (1) and necessitates the search for personalized solutions that can be learned in federated fashion.

**Low Dimensional Common Representation**. There have been numerous examples in image classification and word prediction where tasks with heterogeneous data share a common, low dimensional representation, despite having different labels (Bengio et al., 2013; LeCun et al., 2015; Pillutla et al., 2022). Based on that, a reasonable choice for $\mathcal{Q}$ is a set in which all $q_i$ share a common map, coupled with a personalized map that fits their local data. To formalize this, suppose the ground-truth map can be written for each client $i$ as $q_i = h_i \circ \phi$ where $\phi : \mathbb{R}^d \to \mathbb{R}^k$ is a shared global representation which maps $d$-dimensional data points to a lower dimensional space of size $k$ and $h_i : \mathbb{R}^k \to \mathcal{Y}$, which maps from the lower dimensional subspace to the space of labels. Typically $k \ll d$ and thus given any fixed representation $\phi$, the client specific heads $h_i : \mathbb{R}^k \to \mathcal{Y}$ are easy to optimize locally. With this common structure into consideration, (1) can be reformulated as $\min_{\phi \in \Phi} \frac{1}{M} \sum_{i=1}^{M} \min_{h_i \in \mathcal{H}} f_i(h_i \circ \phi)$, where $\Phi$ is the class of feasible representation and $\mathcal{H}$ is the class of feasible heads. This formulation leads to good local solutions, if the underlying data generation models for the clients share a low dimensional common representation, i.e., $y_i = h_i^* \circ \phi^*(\mathbf{x}_i) + z_i$, where $z_i$ is some additive noise.

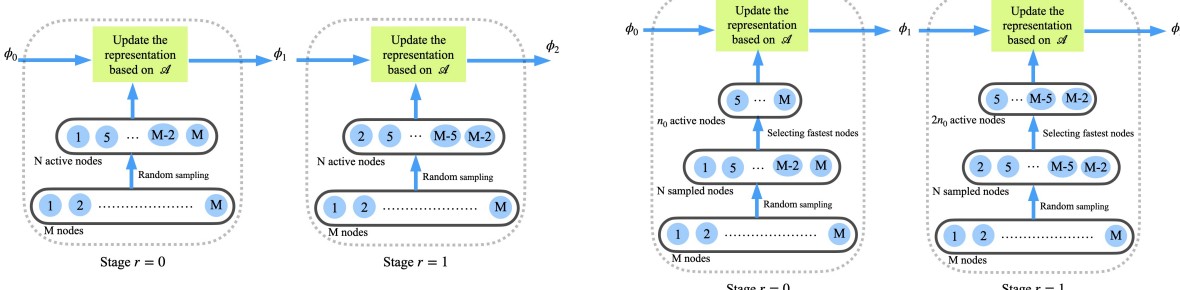

Figure 1: Classic FL schemes for solving (2)

Figure 2: `SRPFL` for solving (2)

The server and clients collaborate to learn the common representation $\phi$, while locally each client learns their unique head $h_i$. Since clients often do not have access to their true data distributions, instead of minimizing their expected loss, they settle for minimizing the empirical loss associated with their local samples. Specifically, we assume client $i$ has access to $S_i$ samples $\{\mathbf{x}_i^1, \mathbf{x}_i^2, ..., \mathbf{x}_i^{S_i}\}$, and its local empirical loss is $\hat{f}_i(h_i \circ \phi) = \frac{1}{S_i} \sum_{s=1}^{S_i} \ell(h_i \circ \phi(\mathbf{x}_i^s), y_i^s)$. Hence, the global problem becomes

$$\min_{\phi \in \Phi} \frac{1}{M} \sum_{i=1}^{M} \min_{h_i \in \mathcal{H}} \left\{ \hat{f}_i(h_i \circ \phi) := \frac{1}{S_i} \sum_{s=1}^{S_i} \ell(h_i \circ \phi(\mathbf{x}_i^s), y_i^s) \right\} \tag{2}$$

**System Heterogeneity Model**. In most FL settings, thousands of clients participate in the training process, each with different computational capabilities. Thus, fixed computational tasks such as gradient computations or local model updates require different processing time, for different clients. Formally, for each client $i \in [M]$, we denote by $\mathcal{T}_i$ the time required to compute a local model update. When a subset of clients participate in the learning process, the computational time of each round is determined by the slowest participating node. Naturally, as the number of nodes in the network grows, we expect the number of stragglers to increase. This phenomenon calls for the development of straggler-resilient methods in system-heterogeneous settings.

## 3 Straggler-Resilient Personalized FL

In the shared representation setting, we face the challenge of finding an algorithm that coordinates server and clients in order to learn a common representation and a set of personalized parameters in a federated and straggler-resilient fashion. To this end, we propose a method that tackles problem (2) with limited sample access and provably superior performance over naive, straggler-prone methods. Specifically, we propose the Straggler-Resilient Personalized Federated Learning (`SRPFL`) meta-algorithm, designed to mitigate the effects of system heterogeneity in environments with non-convex loss functions and heterogeneous data while accommodating a variety of methods as subroutines. In a nutshell, `SRPFL` iteratively solves problem (2), while adaptively increasing the set of participating nodes based on their computational capabilities. As a result the process of learning the common representation across clients' tasks is accelerated, without compromising the resulting accuracy.

To simplify the exposition, henceforth we denote by $\mathcal{A}$ some federated algorithm of choice, designed to solve (2). As depicted in Figure 1, in standard FL frameworks, out of all $M$ clients in the network, the server often selects uniformly at random a subset of size $N$. Subsequently, a few iterations of algorithm $\mathcal{A}$ are performed to approximately solve a version of (2) which corresponds to those $N$ selected clients i.e., $\min_{\phi \in \Phi} \frac{1}{N} \sum_{i=1}^{N} \min_{h_i \in \mathcal{H}} \hat{f}_i(h_i \circ \phi)$. In every following stage, a new subset of $N$ nodes is sampled and the same process is repeated. Although such a procedure eventually learns the global representation across all tasks, it is susceptible to delays caused by stragglers, as the server has to wait for the slowest client (among the $N$ selected ones) to complete its updates. Hence, when $N$ is large the training procedure could become prohibitively slow.

SRPFL takes a different approach in order to mitigate the effects of straggling clients. An overview of the selecting scheme is provided in Figure 2. At each stage, $N$ clients are randomly selected, but only a small, fast subset of them is used to solve their corresponding learning problem. More precisely, suppose that at stage $r$, each client $i$ in the sampled subset of size $N$, is associated with a computational time $\mathcal{T}_i^r$. For simplicity we assume that the nodes are re-indexed at every stage so that they maintain a decreasing ordering w.r.t. their times, i.e. $\mathcal{T}_1^r \leq \mathcal{T}_2^r \leq ... \leq \mathcal{T}_N^r$. Initially, only the $n_0$ fastest clients, $\{1, 2, ..., n_0\}$, are included in the learning procedure, with $n_0$ much smaller than $N$. At every communication round, the set of $n_0$ nodes perform the model updates indicated by algorithm $\mathcal{A}$, to solve their version of (2), i.e., $\min_{\phi \in \Phi} \frac{1}{n_0} \sum_{i=1}^{n_0} \min_{h_i \in \mathcal{H}} \hat{f}_i(h_i \circ \phi)$. We note that during this stage of the algorithm, the server waits only for the slowest client among the participating ones, i.e., client $n_0$ which is potentially much faster than node $N$.

*Remark* 3.1. In practice, the knowledge of clients' computational power is not required to figure out the $n_0$ fastest nodes. Instead, the server sends the global representation model to all $N$ sampled clients and updates the common representation once the first $n_0$ new models are received. Indeed, these representations belong to the fastest $n_0$ nodes.

Once the current stage terminates, a new batch of $N$ clients is sampled and the $2n_0$ fastest nodes are chosen to participate in the learning process. Since speeds vary across stages, consecutive sets of participating nodes could have small or no overlap. However, the representations learned in previous stages still operate as good starting points for subsequent stages which is possible since nodes are homogeneous w.r.t. their representations (see Section 2). Thus, utilizing the representation model learned from the previous stage, nodes $\{1, 2, ..., 2n_0\}$ continue the learning process deriving a model of improved accuracy. The procedure of geometrically increasing the number of participating nodes continues until the target accuracy is achieved. Hence, SRPFL uses the data of stragglers only at the latest stages of the algorithm when an accurate approximation is required.

*Remark* 3.2. For simplicity of exposition we assume that the set of $N$ sampled nodes maintains connectivity to the server throughout each stage. However, our analysis remains unaffected even if a new set of nodes is sampled at every round.

We proceed to characterize the class of federated algorithms able to employ SRPFL to enhance their performance in system heterogeneous environments. Any iterative method that solves an instance of (2) can be combined with our adaptive node participation scheme, however in this paper, we focus on a broad class of alternating gradient-based update methods, presented in (Collins et al., 2021). As the name suggests, in each round, clients update their heads and representation models in an alternative fashion. After a certain number of gradient updates is completed, all clients send their derived representations to the server where the models are averaged and broadcasted back to the clients. Next, we rigorously illustrate this procedure.

**Alternating Update Scheme**. At round $t$, the server communicates a common representation $\phi^t$ to the clients and a subset of them $\mathcal{I}^t$, are selected to participate by performing the following updates.

*Client Head Update.* Each client $i \in \mathcal{I}^t$ performs $\tau_h$ local gradient-based updates optimizing their head parameter, given the received representation $\phi^t$. Concretely, for $s = 1, ..., \tau_h$ client $i$ updates their head model as follows

$$h_i^{t,s} = GRD\left(f_i\left(h_i^{t,s-1}, \phi^t\right), h_i^{t,s-1}, \eta\right). \tag{3}$$

*Client Representation Update.* Once the updated local heads $h_i^{t,\tau_h}$ are obtained, each client $i$ executes $\tau_\phi$ local updates on their representation parameters. That is for $s = 1, ..., \tau_\phi$

$$\phi_i^{t,s} = GRD\left(f_i\left(h_i^{t,\tau_h}, \phi_i^{t,s-1}\right), \phi_i^{t,s-1}, \eta\right). \tag{4}$$

In the above expressions, $GRD(f, h, \eta)$ captures the generic notion of an update of variable $h$ using the gradient of function $f$ with respect to $h$ and step size $\eta$. This notation allows the inclusion of a large class of algorithms such as Gradient Descent with momentum, SGD, etc.

*Server Update.* Each client $i$ sends their derived representation models $\phi_i^{t,\tau_\phi}$ to the server, where they are averaged producing the next representation model $\phi^{t+1}$.

Coupling SRPFL with a generic subroutine that falls into the Alternating Update Scheme, gives rise to Algorithm 1. Every stage $r$ is characterized by a participating set of size $2^r \cdot n_0$, denoted by $\mathcal{I}^r$. At the

---

**Algorithm 1** SRPFL
___
1: **Input:** Initial number of nodes $n_0$; step size $\eta$;
   number of local updates for head $\tau_h$; number of local updates for representation $\tau_\phi$.
2: **Initialization:** $n \leftarrow n_0$, $\phi_0, h_1^{0,\tau_h}, ..., h_N^{0,\tau_h}$
3: **for** $r = 0, 1, 2, \ldots, \log(N/n_0)$ **do**
4:     $\phi^0 \leftarrow \phi_r$
5:     **for** $t = 1, 2, \ldots, \tau_r$ **do**
6:        Server sends representation $\phi^{t-1}$ to $N$ clients sampled from $[M]$.
7:        **for** $i \in \mathcal{I}^r$ **do**
8:           Client $i$ initializes $h_i^{t,0} \leftarrow h_i^{t-1,\tau_h}$ and runs $\tau_h$ updates $h_i^t = h_i^t - \eta \nabla_{h_i^t} f_i(h_i^t, \phi^{t-1})$.
9:           Client $i$ initializes $\phi_i^{t,0} \leftarrow \phi^{t-1}$ and runs $\tau_\phi$ updates $\phi_i^t = \phi_i^t - \eta \nabla_{\phi_i^t} f_i(h_i^t, \phi_i^t)$.
10:          Client $i$ sends $\phi_i^{t,\tau_\phi}$ to the server.
11:        **end for**
12:        **for** each client $i \notin \mathcal{I}^r$ **do**
13:          $h_i^{t,\tau_h} \leftarrow h_i^{t-1,\tau_h}$
14:        **end for**
15:        Server computes $\phi^t \leftarrow \frac{1}{n} \sum_{i=1}^n \phi_i^{t,\tau_\phi}$.
16:     **end for**
17:     Server sets $n \leftarrow \min\{N, 2n\}$ and $\phi_{r+1} \leftarrow \phi^{\tau_r}$.
18: **end for**

---

beginning of each round the server provides a representation model to the participating clients (line 6). The clients update their models (lines 8 and 9) and sent their representations back to the server where they are averaged producing a new global model. The numbers of local updates $\tau_h$, $\tau_\phi$ depend on the subroutine method of choice and the number of rounds per stage is denoted by $\tau_r$. At the end of every stage the set of participating nodes is doubling in size until a set of size $N$ is reached (line 17).

Remark 3.3 summarizes the technical novelties of our work and highlights crucial benefits enjoyed by SRPFL.

*Remark* 3.3. Reisizadeh et al. (2022) proposed a similar participation scheme, however their approach differs from ours and their results apply to significantly more restrictive regimes. Specifically, in (Reisizadeh et al., 2022) the analysis heavily relies on deriving a connection between the ERM solutions of consecutive stages. In order to control the statistical accuracy of the corresponding ERM problems (i) data homogeneity across all clients is necessary and further (ii) clients who participate in early stages are required to remain active and connected to the server in all subsequent stages, maintaining fixed computational speeds throughout the whole training process. (iii) The results of their analysis hold only for strongly convex loss functions and (iv) their stage termination criterion requires the knowledge of the strong convexity parameter. The above restrictions are detrimental in the FL regime and severely undermine the applicability of the resulting algorithm.

In our work we follow a different approach controlling -in terms of principal angle distance - a quantity analogous to statistical accuracy, therefore directly connecting the common representation (and overall solution) at every stage to the ground truth representation. This novel approach allows our algorithm to accommodate (i) data heterogeneity, (ii) clients with dynamically changing speeds or equivalently clients that are replaced by new ones at every round, and (iii) non-convex loss functions. Additionally, a major part of our technical contribution focuses on (iv) analytically deriving the optimal number of rounds per stage, thus producing a simple and efficient doubling scheme.

## 4   SRPFL in the Linear Representation Case

Our theoretical analysis focuses on a specific instance of (1), where clients strive to solve a linear representation learning problem. Concretely we assume that $f_i$ is the quadratic loss, $\phi$ is a projection onto a $k$-dimensional subspace of $\mathbb{R}^d$, given by matrix $\mathbf{B} \in \mathbb{R}^{d \times k}$ and the $i$-th client's head $h_i$, is a vector $\mathbf{w}_i \in \mathbb{R}^k$. We model local data of client $i$ such that $y_i = \mathbf{w}_i^{*\top} \mathbf{B}^{*\top} \mathbf{x}_i + z_i$, for some ground truth representation $\mathbf{B}^* \in \mathbb{R}^{d \times k}$, local heads $\mathbf{w}_i^* \in \mathbb{R}^k$ and $z_i \sim \mathcal{N}(0, \sigma^2)$ capturing the noise in the measurements. Hence, all clients' optimal solutions lie

---

**Algorithm 2** FedRep-SRPFL (Linear Representation)

---

1: **Input:** Step size $\eta$; Batch size $m$; Initial nodes $n_0$;
2: **Initialization:** Client $i \in [N]$ sends to server: $\qquad \mathbf{P}_i := \frac{1}{m}\sum_{j=1}^{m}(y_i^{0,j})^2 \mathbf{x}_i^{0,j}(\mathbf{x}_i^{0,j})^\top, \qquad n \leftarrow n_0.$
3: Server finds $\mathbf{UDU}^\top \leftarrow$ rank-$k$ SVD$(\frac{1}{N}\sum_{i=1}^{N}\mathbf{P}_i)$.
4: **for** $r = 0, 1, 2, \ldots, \log(N/n_0)$ **do**
5: $\quad$ Server initializes representation $\mathbf{B}^{r,0} \leftarrow \mathbf{U}$.
6: $\quad$ **for** $t = 1, 2, \ldots, \tau_r$ **do**
7: $\quad\quad$ Server sends $\mathbf{B}^{r,t}$ to $N$ clients sampled from $[M]$.
8: $\quad\quad$ **for** $i \in \{1, .., n\}$ **do**
9: $\quad\quad\quad$ Client $i$ samples a fresh batch of $m$ samples.
10: $\quad\quad\quad$ Client $i$ computes $\mathbf{w}_i^{r,t+1} \leftarrow \arg\min_{\mathbf{w}} \hat{f}_i^t(\mathbf{w}, \mathbf{B}^{r,t})$.
11: $\quad\quad\quad$ Client $i$ computes $\mathbf{B}_i^{r,t+1} \leftarrow \mathbf{B}^{r,t} - \eta\nabla_{\mathbf{B}}\hat{f}_i^t(\mathbf{w}_i^{t+1}, \mathbf{B}^{r,t})$ and sends it back to the server.
12: $\quad\quad$ **end for**
13: $\quad\quad$ Server computes $\bar{\mathbf{B}}^{r,t+1} \leftarrow \frac{1}{n}\sum_{i \in \mathcal{I}^t}\mathbf{B}_i^{r,t+1}$.
14: $\quad\quad$ Server computes $\mathbf{B}^{r,t+1}, \mathbf{R}^{r,t+1} \leftarrow$ QR$(\bar{\mathbf{B}}^{r,t+1})$.
15: $\quad$ **end for**
16: $\quad$ Server sets $\mathbf{U} \leftarrow \bar{\mathbf{B}}^{r,t+1}$ and $n \leftarrow \min\{N, 2n\}$.
17: **end for**

---

in the same $k$-dimensional subspace. Under these assumptions the global expected risk is

$$\min_{\mathbf{B},\mathbf{W}} \frac{1}{2M}\sum_{i=1}^{M} \mathbb{E}_{(\mathbf{x}_i,y_i)\sim\mathcal{D}_i}\left[\left(y_i - \mathbf{w}_i^\top\mathbf{B}^\top\mathbf{x}_i\right)^2\right], \tag{5}$$

where $\mathbf{W} = [\mathbf{w}_1^\top, ..., \mathbf{w}_N^\top] \in \mathbb{R}^{N \times k}$ is the concatenation of the client-specific heads. Since the true distributions $\mathcal{D}_i$'s are unknown, algorithms strive to minimize the empirical risk instead. The global empirical risk over all clients is

$$\frac{1}{M}\sum_{i=1}^{M}\hat{f}_i(\mathbf{w}_i, \mathbf{B}) = \frac{1}{2Mm}\sum_{i=1}^{M}\sum_{j=1}^{m}\left(y_i^j - \mathbf{w}_i^{t\top}\mathbf{B}^\top\mathbf{x}_i^j\right)^2, \tag{6}$$

where $m$ is the number of samples at each client. The global loss in (6) is nonconvex and has many global minima, including all pairs of $(\mathbf{W}^*\mathbf{Q}^{-1}, \mathbf{B}^*\mathbf{Q}^\top)$, where $\mathbf{Q} \in \mathbb{R}^{k \times k}$ is some invertible matrix. Thus, the server aims to retrieve the column space of $\mathbf{B}^*$, instead of the ground truth factors $(\mathbf{W}^*, \mathbf{B}^*)$. To measure closeness between column spaces, we adopt the metric of principal angle distance (Jain et al., 2013).

**Definition 4.1.** Let matrices $\mathbf{B}_1, \mathbf{B}_2 \in \mathbb{R}^{d \times k}$ and $\hat{\mathbf{B}}_{1,\perp}, \hat{\mathbf{B}}_2$ orthonormal matrices s.t. $span(\hat{\mathbf{B}}_{1,\perp}) = span(\mathbf{B}_1)^\perp$ and $span(\hat{\mathbf{B}}_2) = span(\mathbf{B}_2)$. The principle angle distance between the column spaces of $\mathbf{B}_1$ and $\mathbf{B}_2$ is defined to be dist$(\mathbf{B}_1, \mathbf{B}_2) := \|\hat{\mathbf{B}}_{1,\perp}^\top\hat{\mathbf{B}}_2\|_2$.

Federated Representation Learning (FedRep) is an alternating minimization-descent algorithm, recently proposed in (Collins et al., 2021) for the Linear Shared Representation framework. SRPFL coupled with FedRep gives rise to Algorithm 2. Below we highlight the main points of interest.

In the initialization phase (lines 2 and 3) a model of bounded distance from the optimal representation is obtained, via the Method of Moments (Tripuraneni et al., 2021). Subsequently, at every round $t$, client $i$ samples a fresh batch of samples $\{\mathbf{x}_i^{t,j}, y_i^{t,j}\}_{j=1}^{m}$ from its local distribution (line 9) and thus the corresponding loss function becomes $\hat{f}_i(\mathbf{w}_i \circ \mathbf{B}) := \frac{1}{2m}\sum_{j=1}^{m}(y_i^{t,j} - \mathbf{w}_i^\top\mathbf{B}^\top\mathbf{x}_i^{t,j})^2$. Utilizing the global representation provided by the server, client $i$ computes the optimal head $\mathbf{w}_i$ (line 10). Fixing the newly computed head, client $i$ proceeds to update its global representation model with one step of gradient descent (line 11) and transmits it back to the server. As depicted in lines 13 and 14 the parameter server averages the models received and orthogonalizes the resulting matrix before broadcasting it to the clients, a component of crucial importance required in our analysis.

Mapping this method back to Algorithm 1 we note that the number of representation model updates $\tau_\phi$ is set to 1, whereas the number of head updates $\tau_h$ is sufficiently large to derive (approximately) the optimal solutions. This imbalance is designed to take advantage of the inherent structure of our problem where the

size of $\mathbf{w}_i$'s is significantly smaller than $d$. Finally, we point out that the number of the communication rounds per stage $\tau_r$ is a small and a priori known to the algorithm constant, specified by our analysis.

*Remark* 4.2. Although our proposed method utilizes `FedRep` as a backbone, our analysis and framework differ substantially from the ones in (Collins et al., 2021). Specifically, Collins et al. (2021) assume access to (i) infinite samples, (ii) without the presence of noise. Additionally, (iii) the number of participating nodes remains fixed throughout the training and (iv) the focus lies solely on handling heterogeneous data with system heterogeneity being an orthogonal direction to their work. In contrast, our analysis requires only (i) finite and (ii) noisy samples, and our theoretical results (Theorems 4.7 and 4.9) revolve around (iii) participating subsets of different sizes and (iv) regimes where both data and system heterogeneity is prevalent.

## 4.1 Theoretical Results

In this subsection, we provide rigorous analysis of `FedRep-SRPFL` in the linear representation setting. First, we present the notion of Wall Clock Time (WCT) which is the measure of the performance for our algorithm. Subsequently, we illustrate the contraction inequality that determines the rate at which the distance to the optimal representation diminishes. We conclude showing that `FedRep-SRPFL` outperforms its straggler-prone variant by a factor of $\mathcal{O}(\log N)$, under the standard assumption that clients' computational times follow the exponential distribution. Before we proceed, we introduce the necessary notation and the assumptions.

$$E_0 := 1 - \text{dist}^2\left(\mathbf{B}^0, \mathbf{B}^*\right), \tag{7}$$

$$\bar{\sigma}_{\max,*} := \max_{\mathcal{I} \in [N], |\mathcal{I}|=n, n_0 \leq n \leq N} \sigma_{\max}\left(\frac{1}{\sqrt{n}}\mathbf{W}^*_\mathcal{I}\right), \tag{8}$$

$$\bar{\sigma}_{\min,*} := \min_{\mathcal{I} \in [N], |\mathcal{I}|=n, n_0 \leq n \leq N} \sigma_{\min}\left(\frac{1}{\sqrt{n}}\mathbf{W}^*_\mathcal{I}\right) \tag{9}$$

**Assumption 4.3.** (Sub-gaussian design). The local samples $\mathbf{x}_i \in \mathbb{R}^d$ are i.i.d. with mean 0, covariance $\mathbf{I}_d$ and are $\mathbf{I}_d$-sub-gaussian, i.e. $\mathbb{E}[e^{\mathbf{v}^\top \mathbf{x}_i}] \leq e^{\|\mathbf{v}\|_2^2/2}$ for all $\mathbf{v} \in \mathbb{R}^d$.

**Assumption 4.4.** (Client diversity). Let $\bar{\sigma}_{\min,*}$ defined in (9), be the minimum singular value of any matrix that can be obtained by taking $n$ rows of $\frac{1}{\sqrt{n}}\mathbf{W}^*$. Then $\bar{\sigma}_{\min,*} > 0$.

Specifically, our theoretical analysis requires Assumption 4.4 to be satisfied for every $n \geq n_0$. Assumption 4.4 implies that the optimal heads, of the participating clients, span $\mathbb{R}^k$. This is true in many FL regimes as the number of clients is usually much larger than the dimension of the shared representation.

*Remark* 4.5. In this work we consider client speeds being independent of the local data available to them. This is a natural assumption since the computational power of each client crucially depends on their device characteristics (battery, CPU, etc.), whereas any connection to their local data is unclear. However, in the presence of strong correlation between data and system heterogeneity, Assumption 4.4 may not hold, which can be seen as a potential limitation of our work and an interesting future direction to explore.

**Assumption 4.6.** (Client normalization). The ground-truth client specific parameters satisfy $\|\mathbf{w}_i^*\|_2 = \sqrt{k}$ for all $i \in [n]$ and $\mathbf{B}^*$ has orthonormal columns.

Assumption 4.6 ensures the ground-truth matrix $\mathbf{W}^*\mathbf{B}^{*\top}$ is row-wise *incoherent*, i.e. its row norms have similar magnitudes. This is of vital importance since our measurement matrices are row-wise sparse and incoherence is a common requirement in sensing problems with sparse measurements.

**Wall Clock Time.** To measure the speedup that our meta-algorithm enjoys we use the concept of real time or WCT as described below. `FedRep-SRPFL` runs in communication rounds grouped into stages. Consider such a round $t$ at stage $r$, with nodes $\{1, 2, ..., n_r\}$ participating in the learning process. Here $n_r$ denotes the slowest participating node. The expected amount of time that the server has to wait for the updates to take place is $\mathbb{E}\left[\mathcal{T}^r_{n_r}\right]$. Put simply, the expected computational time of the slowest node acts as the bottleneck for the round. Further, at the beginning and at the end of every round, models are exchanged between the server and the clients. This incurs an additional, fixed communication cost $\mathcal{C}$. If $\tau_r$ communication rounds take place at every stage $r$, then the overall expected WCT for `FedRep-SRPFL` is $\mathbb{E}[T_{SRPFL}] = \sum_{r=0}^{\log(N/n_0)} \tau_r \cdot \left(\mathbb{E}\left[\mathcal{T}^r_{n_r}\right] + \mathcal{C}\right)$. Similarly, the total expected runtime for `FedRep` can be expressed in terms of the total number of rounds, $T_{FR}$, as $\mathbb{E}[T_{FedRep}] = T_{FR}(\mathbb{E}[\mathcal{T}^r_N] + \mathcal{C})$. Taking the ratio of these quantities derives the desired speedup guarantee.

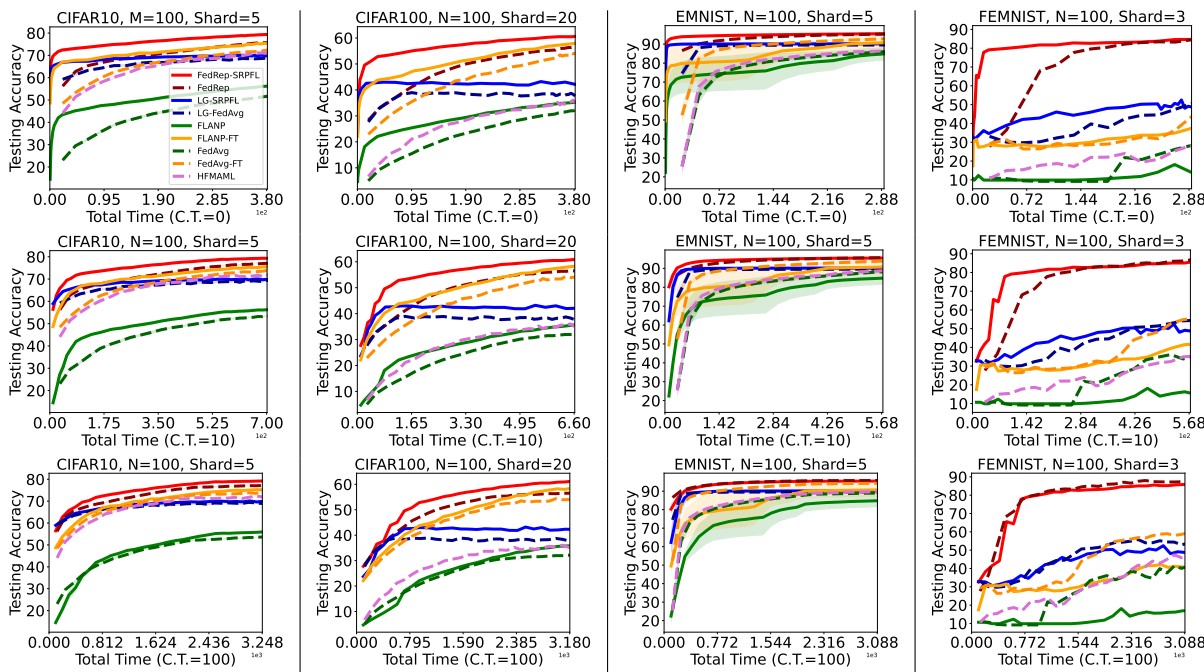

Figure 3: Numerical results on CIFAR10, CIFAR100, EMNIST, FEMNIST with full participation ($M = N$) in the fixed computation speeds setting. 'Shard' denotes the number of classes per client. 'C.T.' denotes the communication cost per round.

**Contraction Inequality.** Theorem 4.7 captures the progress made between two consecutive rounds of `FedRep-SRPFL`. It follows that the rate of convergence to the optimal representation is exponentially fast, provided that the number of participating nodes and the batch size are sufficiently large.

**Theorem 4.7.** *Let Assumptions 4.3-4.6 hold. Further, let the following inequalities hold for the number of participating nodes and the batch size respectively, $n \geq n_0$ and $m \geq c_0 \frac{(1+\sigma^2)k^3\kappa^4}{E_0^2} \max\{\log(N), d/n_0\}$, for some absolute constant $c_0$. Then `FedRep-SRPFL` with stepsize $\eta \leq \frac{1}{8\bar{\sigma}^2_{\max,*}}$, satisfies the following contraction inequality:*

$$dist\left(\mathbf{B}^{t+1}, \mathbf{B}^*\right) \leq dist\left(\mathbf{B}^t, \mathbf{B}^*\right)\sqrt{1-a} + \frac{a}{\sqrt{\frac{n}{n_0}(1-a)}}, \tag{10}$$

*w.p. at least $1 - T \cdot \exp\left(-90\min\left\{d, k^2\log(N)\right\}\right)$, where $a = \frac{1}{2}\eta E_0\bar{\sigma}^2_{\min,*} \leq \frac{1}{4}$.*

Here $T$ denotes the total number of communication rounds which is logarithmic w.r.t. the target error $\epsilon$. The initial representation computed by the Method of Moments satisfies $dist\left(\mathbf{B}^0, \mathbf{B}^*\right) \leq 1 - C_M$, for some constant $C_M$. Since $E_0$ is strictly greater than zero inequality (10) ensures contraction.

*Remark* 4.8. Theorem 4.7 suggests that the server can learn the ground-truth representation before some of the clients update their local heads or participate in the learning process. This might raise concerns about fairness or accuracy, however it is important to highlight that such concerns are unfounded. This is because, after obtaining it the server shares the ground-truth representation with all the clients in the system. Thus, even if a client $i$ was not selected in the representation learning process, it can still optimize its low-dimensional head $w_i \in \mathbb{R}^k$ using its local samples through a few local updates (given that $k$ is a small constant). Consequently, the derivation of the ground-truth model benefits both the clients that already participated in the learning procedure as well as new clients who opt to join the federated system at a later time.

**Logarithmic Speedup.** Algorithm 2 sets off with $n_0$ participating clients and follows a doubling scheme so that at stage $r$ only the fastest $2^r n_0$ nodes contribute to the learning process. Thus at stage $r$, inequality (10) can be written as:

$$\text{dist}^+ \leq \text{dist} \cdot \sqrt{1-\alpha} + \frac{\alpha}{\sqrt{2^r(1-\alpha)}} \tag{11}$$

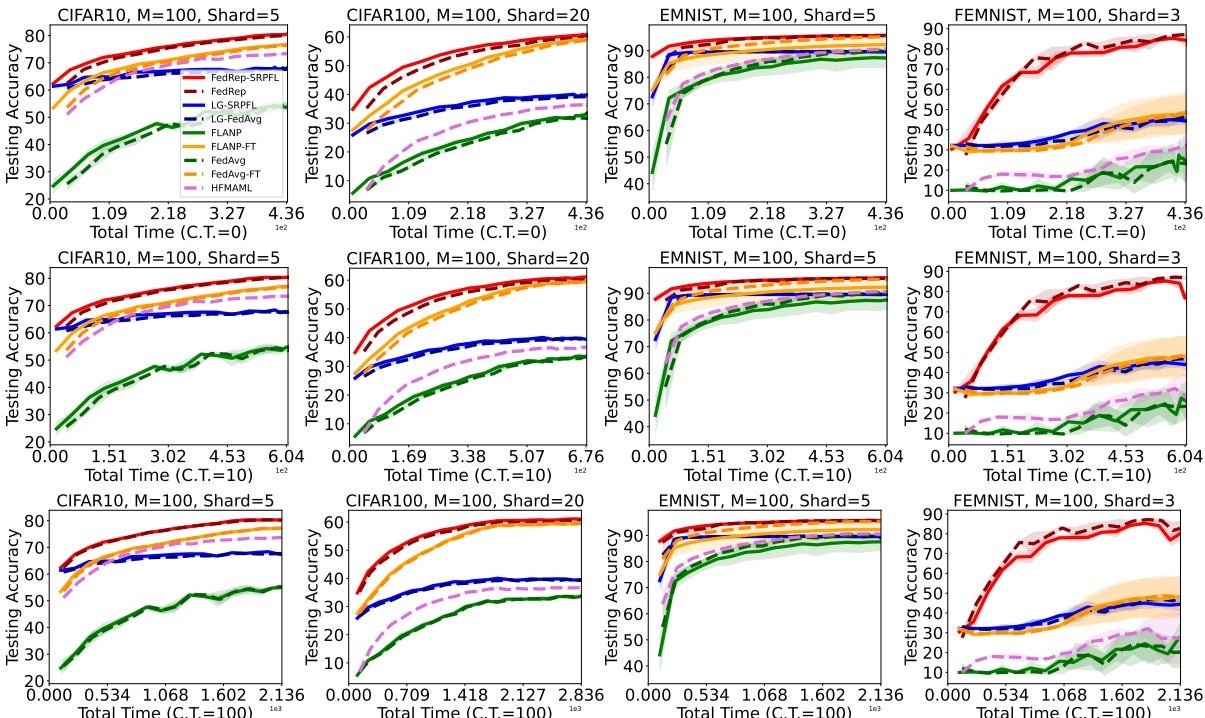

Figure 4: Numerical results on CIFAR10, CIFAR100, EMNIST, FEMNIST with partial participation ($N = M/5$) in the fixed computational speeds setting. 'Shard' denotes the number of classes per client. 'C.T.' denotes the communication cost per round.

We note that the second term on the r.h.s. is an artifact of the noisy measurements. Utilizing geometric series properties we can deduce that in the limit, contraction (11) converges to $\alpha/\sqrt{2^r(1-\alpha)}(1-\sqrt{1-\alpha})$. This implies that the achievable error of our algorithm is lower bounded by $\alpha/\sqrt{\frac{N}{n_0}(1-\alpha)}(1-\sqrt{1-\alpha})$, since the total number of stages is at most $r = \log(N/n_0)$.

To illustrate the theoretical benefits of `SRPFL` we compare Algorithm 2 to `FedRep`. One can distill `FedRep` from Algorithm 2 by disregarding the doubling scheme and instead at each round, randomly sampling $N$ nodes to participate. For fair comparison between the two methods we set the target error small enough so that the contribution of all $N$ nodes is necessary. Specifically, we express the error $\epsilon$ as

$$\epsilon = \hat{c}\frac{\alpha}{\sqrt{\frac{N}{n_0}(1-\alpha)}(1-\sqrt{1-\alpha})}, \quad \text{with} \quad \sqrt{2} > \hat{c} > 1. \tag{12}$$

Intuitively, one should expect `FedRep-SRPFL` to vastly outperform straggler-prone `FedRep` as $\hat{c}$ approaches $\sqrt{2}$ (large error), since in this case the biggest chunk of the workload is completed before `FedRep-SRPFL` utilizes the slower half of the clients. In contrast, `FedRep` experiences heavy delays throughout the whole training process due to the inclusion of stragglers at every round. On the contrary, as $\hat{c}$ approaches 1 (small error), the amount of rounds spent by `FedRep-SRPFL` utilizing $N$ clients increases. In this case one should expect the speedup achieved by `FedRep-SRPFL` in early stages, to eventually become obsolete. Theorem 4.9 provides a rigorous exposition of these insights.

**Theorem 4.9.** *Suppose that at each stage the client's computational times are i.i.d. random variables drawn from the exponential distribution with parameter $\lambda$. Further, suppose that the expected communication cost per round is $\mathcal{C} = \frac{c}{\lambda}$, for some constant c. Finally, consider the target error $\epsilon$ given in* (12). *Then, we have*
$\frac{\mathbb{E}[T_{SRPFL}]}{\mathbb{E}[T_{FedRep}]} = \mathcal{O}\left(\frac{\log\left(\frac{1}{\hat{c}-1}\right)}{\log(N)+\log\left(\frac{1}{\hat{c}-1}\right)}\right).$

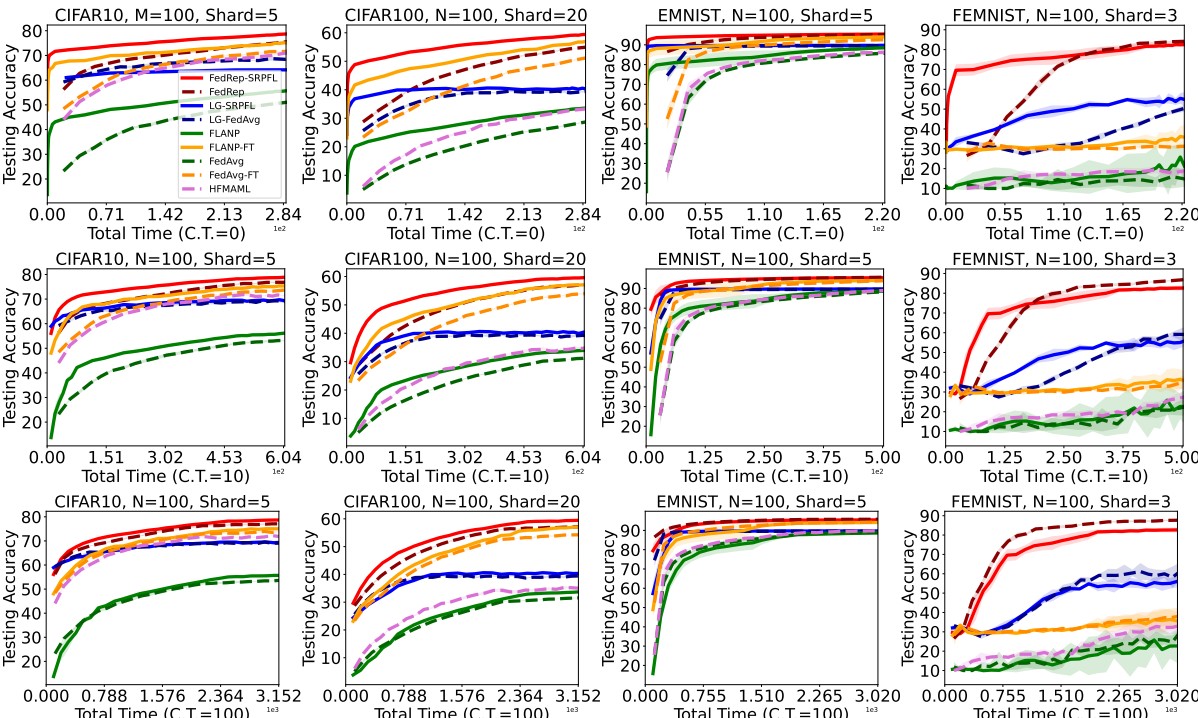

Figure 5: Numerical results on CIFAR10, CIFAR100, EMNIST, FEMNIST with full participation ($M = N$) in the random (dynamic) computation speeds setting. 'Shard' denotes the number of classes per client. 'C.T.' denotes the communication cost per round.

Theorem 4.9 establishes $\mathcal{O}\left(\log N\right)$ speedup for our method compared to its straggler prone benchmark. This result holds when speeds are drawn once at the beginning of the process as well as when new speeds are drawn at every round thus rendering our method versatile in a broad class of settings.

*Remark* 4.10. Our analysis is crucially intertwined with the representation learning framework where the presence of a shared, global representation serves as common ground across data-heterogeneous clients allowing us to show that intermediate solutions constitute good starting points and substantial progress is achieved between stages. Despite `FedAvg` being a general-purpose algorithm not designed for representation learning, it was recently shown to recover the ground-truth representation in the case of multi-task linear regression (Collins et al., 2022), thus casting `FedAvg` a potential candidate subroutine for our method.

*Remark* 4.11. The initialization phase of Algorithm 2 requires a one-time exchange of information between the clients and the server. Although this process reveals only the sum of outer products of local samples, it can be further fortified using differential privacy techniques, such as the ones in (Jain et al., 2021; Shen et al., 2022).

# 5 Experiments

In our empirical study we consider the classification tasks for the CIFAR10, CIFAR100, EMNIST, FEMNIST and Sent140 datasets. We conduct experiments under the full and partial participation scheme with different computation speed distributions comparing the performance of our proposed method against other state-of-the-art benchmarks. Due to space limitation, in this section we present the results for the image classification tasks in the full and partial participation regime with fixed and dynamic computation speeds. Additional results and extensive discussion can be found in Appendix C.

**Baselines**. The first benchmarks under consideration are `FedRep` (Collins et al., 2021) and Local-Global FedAvg (`LG-FedAvg`) (Liang et al., 2020). These federated methods utilize a mixture of global and local models to derive personalized solutions with small global loss. Coupling these algorithms with our proposed doubling scheme gives rise to `FedRep-SRPFL` and `LG-SRPFL`, respectively, which are our proposed algorithms

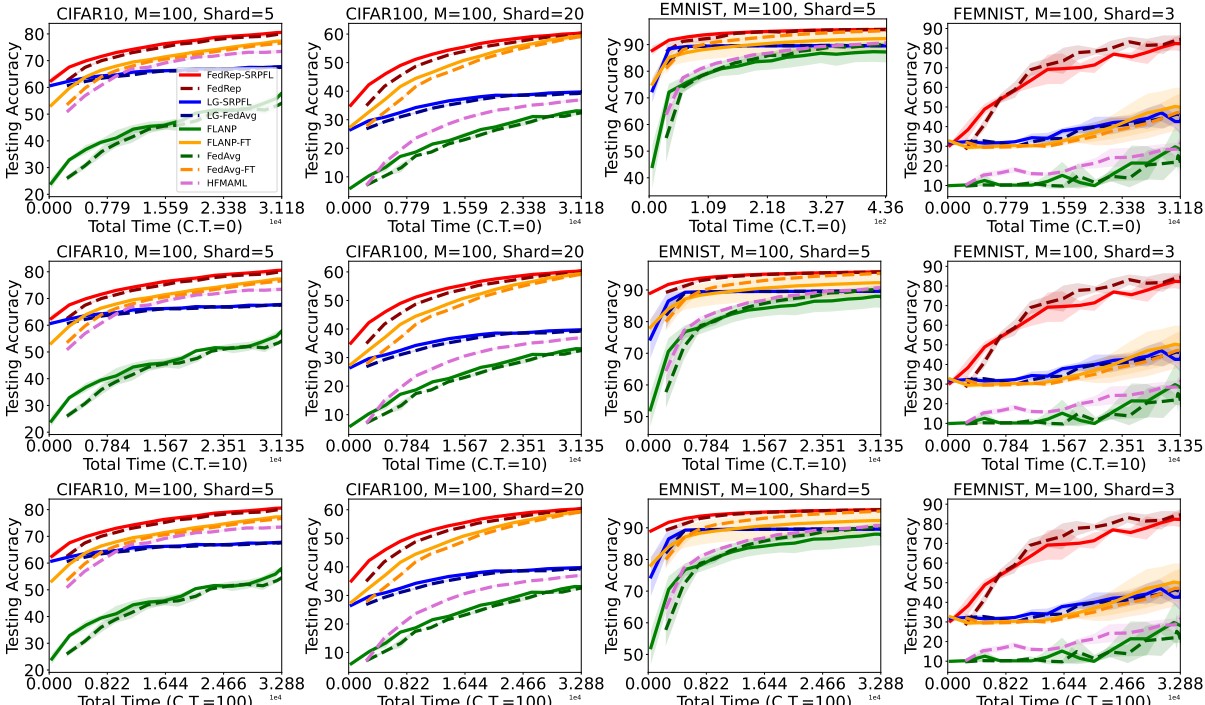

Figure 6: Numerical results on CIFAR10, CIFAR100, EMNIST, FEMNIST with partial participation ($M = N$) in the random (dynamic) computation speeds setting. 'Shard' denotes the number of classes per client. 'C.T.' denotes the communication cost per round.

for the numerical experiments. We further compare our methods with `FLANP` (Reisizadeh et al., 2022) and `FedAvg` (McMahan et al., 2017) with and without fine-tuning. Note that the fine-tuning variants are also considered as they often lead to better performance in data-heterogeneous settings (Wang et al., 2019; Yu et al., 2020). Our final baseline is the `HF-MAML` algorithm from (Fallah et al., 2020a) which is a Model Agnostic Meta Learning-based method producing high quality personalized solutions in data-heterogeneous settings.

**Data allocation.** To ensure that our data allocation is heterogeneous we randomly split the data points among the clients in a way that each client can only observe a specific subset of classes. For instance, in the CIFAR10 dataset where there are in total 10 different classes of data points, each client is only assigned data from 5 different classes, which we refer to as Shards; see first column of Figure 3. We also make sure that the test set for each client is consistent with the samples they have access to at training time, e.g., if client $i$ only observes samples with labels $1, 4, 5$ during training, then at test time they are only asked to classify samples from the same classes. Further details and different allocation schemes are presented in Appendix C.

**Simulation of System Heterogeneity.** We consider two different types of client speed configurations:
*Fixed computation speeds.* In this configuration we sample a value for each client from the exponential distribution with parameter $\lambda$, once at the beginning of the training process. These personalized values capture the computational time of every client and remain fixed throughout the training procedure. In addition to the computational time, our methods suffer a fixed communication cost at every round. In Figures 3 to 6 each row depicts the effects of different values of communication cost on the convergence of the algorithms under consideration (C.T. $= 0, 10, 100$[1]).
*Dynamic computation speeds.* In this configuration every client samples at every round their processing times from a personalized exponential distribution that remains fixed throughout the process. Specifically, at the beginning of the training process we sample for each client $i$ a parameter $\lambda_i$ from the uniform distribution over $[1/M, 1]$. Subsequently, at every round we sample a new computational time for each client $i$ from the exponential distribution with parameter $\lambda_i$. Similarly to the former setting an additional, fixed communication

---

[1]For reference the computational times are sampled with $\lambda$=1.

cost is incurred at every round contributing to the overall running time. As we illustrate in Figure 5 (full participation) and 6 (partial participation - %20), the experimental results under this configuration are qualitatively similar to the ones presented in Figure 3 (full participation) and Figure 4(partial participation - %20) for fixed computation speeds, with SRPFL providing substantial speedup over straggler prone variants.

**Results and Discussions.** From the numerical results in Figures 3 to 6 we distill the following takeaways: 1) Coupling SRPFL with FedRep exhibits consistently superior performance across different datasets and communication time regimes compared to all proposed baselines. 2) Applying SRPFL to personalized FL solvers (FedRep and LG-FedAvg) significantly enhances their efficiency. 3) The speedup achieved by our meta-algorithm is more significant for smaller values of communication time. Concretely, we observe that the gap between FedRep-SRPFL and FedRep as well as LG-SRPFL and LG-FedAvg diminishes as the communication cost increases (plots in the same column C.T.$= 0, 10, 100$ ). This is unsurprising since our method improves the computational cost of the training and thus when the communication cost dominates the overall running time, the benefits of SRPFL are less apparent. 4) FedRep-SRPFL vastly outperforms the fine-tuning variant of the previously proposed FLANP, especially in regimes with high data heterogeneity (FEMNIST).

# 6 Conclusion

In this paper, we proposed SRPFL, a straggler resilient FL meta-algorithm with near-optimal sample complexity and provable logarithmic speedup guarantees in regimes with data and system heterogeneity. Our method leverages ideas from representation learning theory to compute a global representation model along with local client heads, thus deriving personalized solutions for all clients. In SRPFL the participating clients are selected in an adaptive manner. In early stages fast nodes are prioritized and progressively slower nodes are included in the training process, therefore mitigating the effects of stragglers without compromising the quality of the solutions. Our numerical results illustrated the benefits of SRPFL when coupled with different personalized FL methods such as FedRep and LG-FedAvg. Furthermore, our experiments support our theoretical findings showcasing the superior performance of FedRep-SRPFL compared to state-of-the-art FL methods.

### Acknowledgments

The research of I. Tziotis and A. Mokhtari is supported in part by NSF Grants 2019844 and 2112471, ARO Grant W911NF2110226, the Machine Learning Lab (MLL) at UT Austin, and the Wireless Networking and Communications Group (WNCG) Industrial Affiliates Program. The research of Z. Shen and H. Hassani is supported by the NSF Institute for CORE Emerging Methods in Data Science (EnCORE) as well as The Institute for Learning-enabled Optimization at Scale (TILOS). The research of R. Pedarsani is supported by NSF awards 2003035 and 2236483.

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

# A    Appendix

Before we dive into the analysis we provide the following useful definition.

**Definition A.1.** For a random vector $\mathbf{x} \in \mathbb{R}^{d_1}$ and a fixed matrix $\mathbf{A} \in \mathbb{R}^{d_1 \times d_2}$, the vector $\mathbf{A}^\top \mathbf{x}$ is called $\|\mathbf{A}\|_2$-subgaussian if $\mathbf{y}^\top \mathbf{A}^\top \mathbf{x}$ is subgaussian with subgaussian norm $\mathcal{O}\left(\|\mathbf{A}_2\| \|\mathbf{y}\|_2\right)$ for all $\mathbf{y} \in \mathbb{R}^{d_2}$, i.e. $\mathbb{E}\left[\exp\left(\mathbf{y}^\top \mathbf{A}^\top \mathbf{x}\right)\right] \leq \exp\left(\|\mathbf{y}\|_2^2 \|\mathbf{A}\|_2^2 / 2\right)$.

We study the performance of SSRFRL with FedRep as the subroutine of choice. The first part of our analysis focuses on a single round $t$ and extends the analysis in Collins et al. (2021). We assume that there are $N$ clients in the network and at round $t$ a subset $\mathcal{I}^t$ of them participate in the learning procedure with cardinality $n \geq n_0 := \frac{2d}{\log(N) \cdot \bar{\sigma}_{\max,*}}$. Without loss of generality we assume that the clients are indexed from fastest to slowest thus the clients that participate in the learning process are all $i \in [n]$. Each client $i$, draws a batch of $m \geq c_0 \frac{(1+\sigma^2)k^3\kappa^4 \log(N)}{E_0^2}$ fresh, i.i.d. samples at every round. We denote by $\mathbf{X}_i^t \in \mathbb{R}^{d \times m}$ and $\mathbf{Y}_i^t \in \mathbb{R}^m$ the matrix of samples and the labels for client $i$, such that the rows of $\mathbf{X}_i^t$ are samples $\{\mathbf{x}_i^1, ..., \mathbf{x}_i^m\}$. By $\mathbf{Z}_i^t \in \mathbb{R}^m$ we denote the noise in the measurements of client $i$, with $z_{i,j} \sim \mathcal{N}(0, \sigma^2)$. Let $\hat{\mathbf{B}}^* \in \mathbb{R}^{d \times k}$ and $\mathbf{W}^* \in \mathbb{R}^{N \times k}$ stand for the optimal representation and the concatenation of optimal heads respectively. The hat denotes that a matrix is orthonormal i.e. its columns form an orthonormal set. Similarly, $\hat{\mathbf{B}}^t \in \mathbb{R}^{d \times k}$ and $\mathbf{W}^t \in \mathbb{R}^{N \times k}$ denote the global representation and the concatenation of the heads at round $t$. $\mathbf{w}_i^*$'s and $\mathbf{w}_i^t$'s denote the optimal heads and the heads at round $t$ which constitute the rows of $\mathbf{W}^*$ and $\mathbf{W}^t$ respectively. Furthermore we define $\bar{\sigma}_{\min,*} := \min_{\mathcal{I} \in [N], |\mathcal{I}| = n', n' \leq N} \sigma_{\min} \frac{1}{\sqrt{n'}} \mathbf{W}_{\mathcal{I}}^*$ and $\bar{\sigma}_{\max,*} := \max_{\mathcal{I} \in [N], |\mathcal{I}| = n', n' \leq N} \sigma_{\max} \frac{1}{\sqrt{n'}} \mathbf{W}_{\mathcal{I}}^*$, where $\mathbf{W}_{\mathcal{I}}$ is formed by taking the rows of $\mathbf{W}$ indexed by $\mathcal{I}$. That is $\bar{\sigma}_{\max,*}$ and $\bar{\sigma}_{\min,*}$ are the maximum and minimum singular values of any submatrix $\mathbf{W}_{\mathcal{I}}^*$ that can be obtained throughout the course of our algorithm. Notice that by assumption 4.6 each row of $\mathbf{W}^*$ has norm $\sqrt{k}$, so $\frac{1}{n'}$ acts as a normalization factor such that $\left\|\frac{1}{n'} \mathbf{W}_{\mathcal{I}}^*\right\|_F = \sqrt{k}$. Finally we define $\kappa = \bar{\sigma}_{\max,*} / \bar{\sigma}_{\min,*}$. Since we focus on a single round the time index can be dropped for simplicity. Further, henceforth we drop the subscripts $\mathcal{I}^t$ on $\mathbf{W}^t$.

First we derive the update scheme our algorithm follows. Notice that the empirical objective function given in (6) can be expressed via matrices $\mathbf{X}_i$ and $\mathbf{Y}_i$,

$$L_N(\mathbf{B}, \mathbf{W}) = \frac{1}{2mn} \sum_{i=1}^n \left(\mathbf{Y}_i - \mathbf{X}_i \hat{\mathbf{B}} \mathbf{w}_i\right)^2 \tag{13}$$

Further, computing the gradients we derive

$$\frac{1}{2mn} \sum_{i=1}^n \nabla_{\hat{\mathbf{B}}} \left(\mathbf{Y}_i - \mathbf{X}_i \hat{\mathbf{B}} \mathbf{w}_i\right)^2 = \frac{1}{mn} \sum_{i=1}^n \mathbf{X}_i^\top \left(\mathbf{X}_i \hat{\mathbf{B}} \mathbf{w}_i - \mathbf{Y}_i\right) \mathbf{w}_i^\top, \tag{14}$$

$$\frac{1}{2mn} \nabla_{\mathbf{w}_i} \sum_{j=1}^n \left(\mathbf{Y}_j - \mathbf{X}_j \hat{\mathbf{B}} \mathbf{w}_j\right)^2 = \frac{1}{mn} \hat{\mathbf{B}}^\top \mathbf{X}_i^\top \left(\mathbf{X}_i \hat{\mathbf{B}} \mathbf{w}_i - \mathbf{Y}_i\right) \tag{15}$$

and since $\left(\frac{1}{m}\hat{\mathbf{B}}^\top \mathbf{X}_i^\top \mathbf{X}_i \hat{\mathbf{B}}\right)$ is invertible with high probability by Lemma A.4, solving for the minimizer gives us

$$\mathbf{w}_i^+ = \left(\frac{1}{m}\hat{\mathbf{B}}^\top \mathbf{X}_i^\top \mathbf{X}_i \hat{\mathbf{B}}\right)^{-1} \frac{1}{m}\hat{\mathbf{B}}^\top \mathbf{X}_i^\top \mathbf{Y}_i \tag{16}$$

Thus our update scheme with stepsize $\eta$ is

$$\forall i \in [n] \quad \mathbf{w}_i^+ = \left(\frac{1}{m}\hat{\mathbf{B}}^\top \mathbf{X}_i^\top \mathbf{X}_i \hat{\mathbf{B}}\right)^{-1} \frac{1}{m}\hat{\mathbf{B}}^\top \mathbf{X}_i^\top \mathbf{Y}_i \tag{17}$$

$$\mathbf{B}^+ = \hat{\mathbf{B}} - \frac{\eta}{mn} \sum_{i=1}^n \mathbf{X}_i^\top \left(\mathbf{X}_i \hat{\mathbf{B}} \mathbf{w}_i^+ - \mathbf{Y}_i\right) \mathbf{w}_i^{+\top} \tag{18}$$

$$\hat{\mathbf{B}}^+, \mathbf{R}^+ = QR(\mathbf{B}^+) \tag{19}$$

where $QR$ denotes the QR decomposition and $\mathbf{Y}_i = \mathbf{X}_i \hat{\mathbf{B}}^* \mathbf{w}_i^* + \mathbf{Z}_i$.

**Lemma A.2.** *For every client $i$ the update for $\mathbf{w}_i$ can be expressed as follows :*

$$\mathbf{w}_i^+ = \hat{\mathbf{B}}^\top \hat{\mathbf{B}}^* \mathbf{w}_i^* + \mathbf{F}_i + \mathbf{G}_i, \tag{20}$$

*where $\mathbf{F}_i$ and $\mathbf{G}_i$ are defined in equations (24) and (25), respectively.*

*Proof.* Further expanding (17) we can write

$$\mathbf{w}_i^+ = \left(\frac{1}{m}\hat{\mathbf{B}}^\top \mathbf{X}_i^\top \mathbf{X}_i \hat{\mathbf{B}}\right)^{-1} \frac{1}{m}\hat{\mathbf{B}}^\top \mathbf{X}_i^\top \mathbf{X}_i \hat{\mathbf{B}}^* \mathbf{w}_i^* + \left(\frac{1}{m}\hat{\mathbf{B}}^\top \mathbf{X}_i^\top \mathbf{X}_i \hat{\mathbf{B}}\right)^{-1} \frac{1}{m}\hat{\mathbf{B}}^\top \mathbf{X}_i^\top \mathbf{Z}_i \tag{21}$$

$$= \hat{\mathbf{B}}^\top \hat{\mathbf{B}}^* \mathbf{w}_i^* + \left(\frac{1}{m}\hat{\mathbf{B}}^\top \mathbf{X}_i^\top \mathbf{X}_i \hat{\mathbf{B}}\right)^{-1} \left(\frac{1}{m}\hat{\mathbf{B}}^\top \mathbf{X}_i^\top \mathbf{X}_i \hat{\mathbf{B}}^* - \frac{1}{m}\hat{\mathbf{B}}^\top \mathbf{X}_i^\top \mathbf{X}_i \hat{\mathbf{B}} \hat{\mathbf{B}}^\top \hat{\mathbf{B}}^*\right) \mathbf{w}_i^*$$

$$+ \left(\frac{1}{m}\hat{\mathbf{B}}^\top \mathbf{X}_i^\top \mathbf{X}_i \hat{\mathbf{B}}\right)^{-1} \frac{1}{m}\hat{\mathbf{B}}^\top \mathbf{X}_i^\top \mathbf{Z}_i \tag{22}$$

$$= \hat{\mathbf{B}}^\top \hat{\mathbf{B}}^* \mathbf{w}_i^* + \mathbf{F}_i + \mathbf{G}_i \tag{23}$$

where we define

$$\mathbf{F}_i := \left(\frac{1}{m}\hat{\mathbf{B}}^\top \mathbf{X}_i^\top \mathbf{X}_i \hat{\mathbf{B}}\right)^{-1} \left(\frac{1}{m}\hat{\mathbf{B}}^\top \mathbf{X}_i^\top \mathbf{X}_i \hat{\mathbf{B}}^* - \frac{1}{m}\hat{\mathbf{B}}^\top \mathbf{X}_i^\top \mathbf{X}_i \hat{\mathbf{B}} \hat{\mathbf{B}}^\top \hat{\mathbf{B}}^*\right) \mathbf{w}_i^*, \tag{24}$$

$$\mathbf{G}_i := \left(\frac{1}{m}\hat{\mathbf{B}}^\top \mathbf{X}_i^\top \mathbf{X}_i \hat{\mathbf{B}}\right)^{-1} \frac{1}{m}\hat{\mathbf{B}}^\top \mathbf{X}_i^\top \mathbf{Z}_i \tag{25}$$

$\square$

We further have the following immediate corollary.

**Corollary A.3.** *Let $\mathbf{W}^+, \mathbf{F}$ and $\mathbf{G}$ be the matrices with rows the concatenation of $\mathbf{w}_i^+, \mathbf{F}_i$ and $\mathbf{G}_i$, respectively. Then*

$$\mathbf{W}^+ = \mathbf{W}^* \hat{\mathbf{B}}^* \hat{\mathbf{B}} + \mathbf{F} + \mathbf{G} \tag{26}$$

Our first goal is to control the norm of $\mathbf{w}_i^+$. In order to achieve that we provide lemmas that bound the norms of $\mathbf{F}_i$ and $\mathbf{G}_i$ extending the analysis in Collins et al. (2021) and Jain et al. (2013).

**Lemma A.4.** *Let $\delta = c\frac{k^{3/2}\sqrt{\log(N)}}{\sqrt{m}}$ for some absolute constant $c$, then with probability at least $1 - \exp\left(-115k^3 \log(N)\right)$*

$$\forall i \in [n], \quad \sigma_{\min}\left(\frac{1}{m}\hat{\mathbf{B}}^\top \mathbf{X}_i^\top \mathbf{X}_i \hat{\mathbf{B}}\right) \geq 1 - \delta \tag{27}$$

*It follows that with the same probability*

$$\forall i \in [n], \quad \sigma_{\max}\left(\frac{1}{m}\hat{\mathbf{B}}^\top \mathbf{X}_i^\top \mathbf{X}_i \hat{\mathbf{B}}\right)^{-1} \leq \frac{1}{1-\delta} \tag{28}$$

*Proof.* First notice that we can rewrite

$$\frac{1}{m}\hat{\mathbf{B}}^\top \mathbf{X}_i^\top \mathbf{X}_i \hat{\mathbf{B}} = \sum_{j=1}^m \frac{1}{\sqrt{m}}\hat{\mathbf{B}}^\top \mathbf{x}_i^j \left(\frac{1}{\sqrt{m}}\hat{\mathbf{B}}^\top \mathbf{x}_i^j\right)^\top \tag{29}$$

For all $i \in [n], j \in [m]$ we define $\mathbf{v}_i^j := \frac{1}{\sqrt{m}}\hat{\mathbf{B}}^\top \mathbf{x}_i^j$ such that each $\mathbf{v}_i^j$ is an i.i.d. $\|\frac{1}{\sqrt{m}}\hat{\mathbf{B}}\|_2$-subgaussian random variable (please see the definition of $\|\mathbf{A}\|_2$-subgaussian in Definition A.1) and thus by equation (4.22)(Theorem 4.6.1) in Vershynin (2018) we obtain the following bound for any $m \geq k, l \geq 0$

$$\sigma_{\min}\left(\frac{1}{m}\hat{\mathbf{B}}^\top \mathbf{X}_i^\top \mathbf{X}_i \hat{\mathbf{B}}\right) \geq 1 - c_1\left(\sqrt{\frac{k}{m}} + \frac{l}{\sqrt{m}}\right), \tag{30}$$

with probability at least $1 - \exp\left(-l^2\right)$ and $c_1$ some absolute constant. We set $l = 12k^{3/2}\log(N)\sqrt{k}$ and $\delta_1 = \frac{12c_1 k^{3/2}\sqrt{\log(N)}}{\sqrt{m}}$ and the above bound becomes

$$\sigma_{\min}\left(\frac{1}{m}\hat{\mathbf{B}}^\top \mathbf{X}_i^\top \mathbf{X}_i \hat{\mathbf{B}}\right) \geq 1 - \delta_1, \tag{31}$$

with probability at least $1 - \exp\left(-k\left(12k\sqrt{\log(N)} - 1\right)^2\right)$

Further notice that

$$\exp\left(-k\left(12k\sqrt{\log(N)} - 1\right)^2\right) = \exp\left(k\left(-144k^2\log(N) + 24k\log(N) - 1\right)\right) \tag{32}$$

$$\leq \exp\left(-120k^3\log(N)\right) \tag{33}$$

Thus taking Union Bound over $i \in [n]$ we have that for all $i \in [n]$

$$\sigma_{\min}\left(\frac{1}{m}\hat{\mathbf{B}}^\top \mathbf{X}_i^\top \mathbf{X}_i \hat{\mathbf{B}}\right) \geq 1 - \delta_1 \tag{34}$$

with probability at least

$$1 - n\exp\left(-120k^3\log(N)\right) \geq 1 - \exp\left(-115k^3\log(N)\right) \tag{35}$$

Choosing $c$ sufficiently large derives the statement of the lemma. □

**Lemma A.5.** *Let* $\mathbf{H}_i := \left(\frac{1}{\sqrt{m}}\hat{\mathbf{B}}^\top \mathbf{X}_i^\top\right)\frac{1}{\sqrt{m}}\mathbf{X}_i\left(\hat{\mathbf{B}}\hat{\mathbf{B}}^\top - \mathbf{I}_d\right)\hat{\mathbf{B}}^*$ *and* $\delta := c\frac{k^{3/2}\sqrt{\log(N)}}{\sqrt{m}}$, *for an absolute constant* $c$. *Then with probability at least* $1 - \exp\left(-115k^2\log(N)\right)$ *we have*

$$\sum_{i=1}^n \|\mathbf{H}_i \mathbf{w}_i^*\|_2^2 \leq \delta^2 \|\mathbf{W}^*\|_2^2 \, dist^2\left(\hat{\mathbf{B}}, \hat{\mathbf{B}}^*\right) \tag{36}$$

*Proof.* In order to argue about the quantity $\mathbf{H}_i = \left(\frac{1}{\sqrt{m}}\hat{\mathbf{B}}^\top \mathbf{X}_i^\top\right)\frac{1}{\sqrt{m}}\mathbf{X}_i\left(\hat{\mathbf{B}}\hat{\mathbf{B}}^\top - \mathbf{I}_d\right)\hat{\mathbf{B}}^*$ we define matrix $\mathbf{U} := \frac{1}{\sqrt{m}}\mathbf{X}_i\left(\hat{\mathbf{B}}\hat{\mathbf{B}}^\top - \mathbf{I}_d\right)\hat{\mathbf{B}}^*$ such that its $j$-th row, $\mathbf{u}_j = \frac{1}{\sqrt{m}}\hat{\mathbf{B}}^{*\top}\left(\hat{\mathbf{B}}\hat{\mathbf{B}}^\top - \mathbf{I}_d\right)\mathbf{x}_i^j$ is subgaussian with norm at most $\frac{1}{\sqrt{m}}\hat{\mathbf{B}}^{*\top}\left(\hat{\mathbf{B}}\hat{\mathbf{B}}^\top - \mathbf{I}_d\right)$. Similarly we define $\mathbf{V} = \frac{1}{\sqrt{m}}\hat{\mathbf{B}}^\top \mathbf{X}_i^\top$ such that its $j$-th row $\mathbf{v}_j = \frac{1}{\sqrt{m}}\hat{\mathbf{B}}\mathbf{x}_i^j$ has norm at most $\frac{1}{\sqrt{m}}\hat{\mathbf{B}}$. We are now ready to use a concentration argument similar to Proposition (4.4.5) in . Let $\mathcal{S}^{k-1}$ denote the unit sphere in $k$ dimensions and $\mathcal{N}_k$ the $1/4$-net of cardinality $9^k$. From equation (4.13) in Vershynin (2018) we have

$$\left\|\left(\hat{\mathbf{B}}^\top \mathbf{X}_i^\top\right)\mathbf{X}_i\left(\hat{\mathbf{B}}\hat{\mathbf{B}}^\top - \mathbf{I}_d\right)\hat{\mathbf{B}}^*\right\|_2 = \left\|\mathbf{U}^\top \mathbf{V}\right\|_2 \leq 2\max_{\mathbf{p},\mathbf{y}\in\mathcal{N}_k}\mathbf{p}^\top\left(\sum_{j=1}^m \mathbf{u}_j \mathbf{v}_j^\top\right)\mathbf{y} \tag{37}$$

$$= 2\max_{\mathbf{p},\mathbf{y}\in\mathcal{N}_k}\sum_{j=1}^m \langle\mathbf{p},\mathbf{u}_j\rangle\langle\mathbf{v}_j,\mathbf{y}\rangle \tag{38}$$

By definition $\langle\mathbf{p},\mathbf{u}_j\rangle$ and $\langle\mathbf{v}_j,\mathbf{y}\rangle$ are subgaussians with norms $\frac{1}{\sqrt{m}}\left\|\hat{\mathbf{B}}^{*\top}\left(\hat{\mathbf{B}}\hat{\mathbf{B}}^\top - \mathbf{I}_d\right)\right\|_2 = \frac{1}{\sqrt{m}}\text{dist}\left(\hat{\mathbf{B}},\hat{\mathbf{B}}^*\right)$ and $\frac{1}{\sqrt{m}}\|\hat{\mathbf{B}}\|_2 = \frac{1}{\sqrt{m}}$ respectively and thus for all $j \in [m]$ the product $\langle\mathbf{p},\mathbf{u}_j\rangle\langle\mathbf{v}_j,\mathbf{y}\rangle$ is subexponential with norm at most $\frac{C'}{m}\text{dist}\left(\hat{\mathbf{B}},\hat{\mathbf{B}}^*\right)$, for some constant $C'$. Note that

$$\mathbb{E}\left[\langle\mathbf{p},\mathbf{u}_j\rangle\langle\mathbf{v},\mathbf{y}\rangle\right] = \mathbf{p}^\top\left(\hat{\mathbf{B}}^{*\top}\left(\mathbf{I}_d - \hat{\mathbf{B}}\hat{\mathbf{B}}^\top\right)\hat{\mathbf{B}}\right)\mathbf{y} = 0 \tag{39}$$

and thus we can use Bernstein's inequality to bound the sum of $m$ zero mean subexponential random variables, for any fixed pair $\mathbf{p}, \mathbf{y} \in \mathcal{N}_k$:

$$\mathbb{P}\left(\sum_{i=1}^{m} \langle \mathbf{p}, \mathbf{u}_j \rangle \langle \mathbf{v}_j, \mathbf{y} \rangle \geq s\right) \leq \exp\left(-c_2 \min\left\{\frac{s^2 m^2}{\operatorname{dist}^2\left(\hat{\mathbf{B}}, \hat{\mathbf{B}}^*\right)}, \frac{sm}{\operatorname{dist}\left(\hat{\mathbf{B}}, \hat{\mathbf{B}}^*\right)}\right\}\right) \tag{40}$$

$$\leq \exp\left(-c_2 m \min\left\{\frac{s^2}{\operatorname{dist}^2\left(\hat{\mathbf{B}}, \hat{\mathbf{B}}^*\right)}, \frac{s}{\operatorname{dist}\left(\hat{\mathbf{B}}, \hat{\mathbf{B}}^*\right)}\right\}\right) \tag{41}$$

for constant $c_2$. Thus taking Union Bound over all the point in the net we derive

$$\mathbb{P}\left(\forall \mathbf{p}, \mathbf{y} \in \mathcal{N}_k, \quad 2\sum_{j=1}^{m} \langle \mathbf{p}, \mathbf{u}_j \rangle \langle \mathbf{v}_j, \mathbf{y} \rangle \geq 2s\right) \leq 9^{2k} \exp\left(-c_2 m \min\left\{\frac{s^2}{\operatorname{dist}^2\left(\hat{\mathbf{B}}, \hat{\mathbf{B}}^*\right)}, \frac{s}{\operatorname{dist}\left(\hat{\mathbf{B}}, \hat{\mathbf{B}}^*\right)}\right\}\right) \tag{42}$$

Since $m > Ck^2 \log(N)$, by setting $s = \operatorname{dist}\left(\hat{\mathbf{B}}, \hat{\mathbf{B}}^*\right)\sqrt{\frac{Ck^2 \log(N)}{4m}}$ and (38) we obtain

$$\mathbb{P}\left(\frac{1}{m}\left\|\left(\hat{\mathbf{B}}^\top \mathbf{X}_i^\top\right)\mathbf{X}_i\left(\hat{\mathbf{B}}\hat{\mathbf{B}}^\top - \mathbf{I}_d\right)\hat{\mathbf{B}}^*\right\|_2 \geq \operatorname{dist}\left(\hat{\mathbf{B}}, \hat{\mathbf{B}}^*\right)\sqrt{\frac{Ck^2 \log(N)}{m}}\right) \leq 9^{2k} \exp\left(-c_2 m \frac{s^2}{\operatorname{dist}^2\left(\hat{\mathbf{B}}, \hat{\mathbf{B}}^*\right)}\right) \tag{43}$$

$$\leq 9^{2k} \exp\left(-C \cdot c_2 m k^2 \log(N)\right) \tag{44}$$

$$\leq \exp\left(-120 k^2 \log(N)\right) \tag{45}$$

for sufficiently large $C$. Using Union Bound again over all participating clients we get

$$\mathbb{P}\left(\forall i \in [n] \quad \|\mathbf{H}_i\|_2 \leq \operatorname{dist}\left(\hat{\mathbf{B}}, \hat{\mathbf{B}}^*\right)\sqrt{\frac{Ck^2 \log(N)}{m}}\right) \geq 1 - n \exp\left(-120 k^2 \log(N)\right) \tag{46}$$

$$\geq 1 - \exp\left(-115 k^2 \log(N)\right) \tag{47}$$

The above also implies

$$\mathbb{P}\left(\frac{1}{n}\sum_{i=1}^{n}\|\mathbf{H}_i\|_2^2 \leq C \operatorname{dist}^2\left(\hat{\mathbf{B}}, \hat{\mathbf{B}}^*\right)\frac{k^2 \log(N)}{m}\right) \geq 1 - \exp\left(-115 k^2 \log(N)\right) \tag{48}$$

$$\mathbb{P}\left(\frac{k}{n}\|\mathbf{W}^*\|_2^2 \sum_{i=1}^{n}\|\mathbf{H}_i\|_2^2 \leq C\|\mathbf{W}^*\|_2^2 \operatorname{dist}^2\left(\hat{\mathbf{B}}, \hat{\mathbf{B}}^*\right)\frac{k^3 \log(N)}{m}\right) \geq 1 - \exp\left(-115 k^2 \log(N)\right) \tag{49}$$

Finally notice that

$$\sum_{i=1}^{n}\|\mathbf{H}_i \mathbf{w}_i^*\|_2^2 \leq \sum_{i=1}^{n}\|\mathbf{H}_i\|_2^2 k \leq \frac{\|\mathbf{W}^*\|_F^2}{n}\sum_{i=1}^{n}\|\mathbf{H}_i\|_2^2 \leq \frac{k}{n}\|\mathbf{W}^*\|_2^2 \sum_{i=1}^{n}\|\mathbf{H}_i\|_2^2, \tag{50}$$

where we used Assumption (4.6) and the fact that the rank of $\mathbf{W}^*$ is $k$. Combining this with (49) and choosing sufficiently large $c$ we derive the result. $\qquad\square$

Building on the previous lemmas we can now bound the norm of $\mathbf{F}_i$.

**Lemma A.6.** *Let $\delta := c\frac{k^{3/2}\sqrt{\log(N)}}{\sqrt{m}}$ for some absolute constant $c$ and for all $i \in [n]$ let $\mathbf{F}_i$ given by (24). Further let matrix $\mathbf{F} \in \mathbb{R}^{n \times k}$ such that its rows are the concatenation of $\mathbf{F}_i$'s. Then with probability at least $1 - \exp\left(-110k^2 \log(N)\right)$ we have*

$$\forall i \in [n] \quad \|\mathbf{F}_i\|_2 \leq \frac{\delta}{1-\delta} dist\left(\hat{\mathbf{B}}, \hat{\mathbf{B}}^*\right) \|\mathbf{w}_i^*\|_2, \tag{51}$$

$$\|\mathbf{F}\|_F \leq \frac{\delta}{1-\delta} dist\left(\hat{\mathbf{B}}, \hat{\mathbf{B}}^*\right) \|\mathbf{W}^*\|_2 \tag{52}$$

*Proof.*

$$\|\mathbf{F}_i\|_2^2 \leq \left\|\left(\frac{1}{m}\hat{\mathbf{B}}^\top \mathbf{X}_i^\top \mathbf{X}_i \hat{\mathbf{B}}\right)^{-1}\right\|_2^2 \|\mathbf{H}_i\|_2^2 \|\mathbf{w}_i^*\|_2^2 \tag{53}$$

$$\leq \frac{\delta^2}{(1-\delta)^2} \cdot \text{dist}^2\left(\hat{\mathbf{B}}, \hat{\mathbf{B}}^*\right) \|\mathbf{w}_i^*\|_2^2 \tag{54}$$

which holds for all $i \in [n]$ with probability at least $1 - \exp\left(-110k^2 \log(N)\right)$ by using Union Bound on the failure probability of (28) and (47). Similarly, we have

$$\|\mathbf{F}\|_F^2 = \sum_{i=1}^n \|\mathbf{F}_i\|_2^2 \leq \sum_{i=1}^m \left\|\left(\frac{1}{m}\hat{\mathbf{B}}^\top \mathbf{X}_i^\top \mathbf{X}_i \hat{\mathbf{B}}\right)^{-1}\right\|_2^2 \|\mathbf{H}_i \mathbf{w}_i^*\|_2^2 \tag{55}$$

$$\leq \frac{1}{(1-\delta)^2} \sum_{i=1}^m \|\mathbf{H}_i \mathbf{w}_i^*\|_2^2 \tag{56}$$

$$\leq \frac{\delta^2}{(1-\delta)^2} \cdot \text{dist}^2\left(\hat{\mathbf{B}}, \hat{\mathbf{B}}^*\right) \|\mathbf{W}^*\|_2^2 \tag{57}$$

which holds with probability at least $1 - \exp\left(-110k^2 \log(N)\right)$ taking Union Bound on the failure probability on (28) and (36). $\qquad\square$

We now turn our attention on deriving a bound for $\|\mathbf{G}_i\|_2$.

**Lemma A.7.** *Let $\delta := c\frac{k^{3/2}\sqrt{\log(N)}}{\sqrt{m}}$ for some absolute constant $c$ and for all $i \in [n]$ let $\mathbf{G}_i$ given by (25). Further let matrix $\mathbf{G} \in \mathbb{R}^{n \times k}$ such that its rows are the concatenation of $\mathbf{G}_i$'s. Then with probability at least $1 - \exp\left(-110k^2 \log(N)\right)$ we have*

$$\forall i \in [n] \quad \|\mathbf{G}_i\|_2 \leq \frac{\delta}{1-\delta}\sigma^2, \tag{58}$$

$$\|\mathbf{G}\|_F \leq \frac{\delta}{1-\delta}\sqrt{n}\sigma^2 \tag{59}$$

*Proof.* Notice that we can write

$$\mathbf{G}_i = \left(\frac{1}{m}\hat{\mathbf{B}}^\top \mathbf{X}_i^\top \mathbf{X}_i \hat{\mathbf{B}}\right)^{-1} \frac{1}{m}\hat{\mathbf{B}}^\top \mathbf{X}_i^\top \mathbf{Z}_i = \left(\frac{1}{m}\hat{\mathbf{B}}^\top \mathbf{X}_i^\top \mathbf{X}_i \hat{\mathbf{B}}\right)^{-1} \frac{1}{m}\sum_{i=1}^m z_i^j \hat{\mathbf{B}}^\top \mathbf{x}_i^j, \tag{60}$$

and since $z_i^j \sim \mathcal{N}\left(0, \sigma^2\right)$ we can conclude that for all $i, j$, $z_i^j \hat{\mathbf{B}}^\top \mathbf{x}_i^j$ is an i.i.d. zero mean subexponential with norm at most $C_2'\sigma^2\|\hat{B}\|_2 = C_2'\sigma^2$ for some constant $C_2$. Once again we denote by $\mathcal{S}^{k-1}$ the unit sphere in $k$ dimensions and by $\mathcal{N}_k$ the $1/4$-net with cardinality $9^k$. Using Bernstein's inequality and Union Bound over all the points on the net we follow the derivations from Lemma A.5 to get

$$\mathbb{P}\left(\left\|\frac{1}{m}\sum_{i=1}^m z_i^j \hat{\mathbf{B}}^\top \mathbf{x}_i^j\right\|_2 \geq 2s\right) \leq 9^{k+1}\exp\left(-c_3 m \min\left\{\frac{s^2}{\sigma^4}, \frac{s}{\sigma^2}\right\}\right) \tag{61}$$

Since $m > C_2 k^2 \log(N)$, by setting $s = \sigma^2 \sqrt{\frac{C_2 k^2 \log(N)}{4m}}$ we derive

$$\mathbb{P}\left(\left\|\frac{1}{m}\sum_{i=1}^{m} z_i^j \hat{\mathbf{B}}^\top \mathbf{x}_i^j\right\|_2 \geq \sigma^2 \sqrt{\frac{C_2 k^2 \log(N)}{m}}\right) \leq 9^{k+1} \exp\left(-c_3 m \frac{s^2}{\sigma^4}\right) \tag{62}$$

$$\leq 9^{k+1} \exp\left(-C_2 \cdot c_3 k^2 \log(N)\right) \tag{63}$$

$$\leq \exp\left(-115 k^2 \log(N)\right) \tag{64}$$

for sufficiently large $C_2$. Choosing $c$ large enough and taking Union Bound over all $i \in [n]$ we can obtain

$$\mathbb{P}\left(\forall i \in [n] \quad \left\|\frac{1}{m}\sum_{i=1}^{m} z_i^j \hat{\mathbf{B}}^\top \mathbf{x}_i^j\right\|_2 \leq \sigma^2 \delta\right) \leq 1 - n\exp\left(-115 k^2 \log(N)\right) \tag{65}$$

$$\leq 1 - \exp\left(-113 k^2 \log(N)\right) \tag{66}$$

$$\tag{67}$$

Finally taking Union Bound over the failure probabilities of (28) and (67) we get

$$\forall i \in [n] \quad \|\mathbf{G}_i\|_2 \leq \left\|\left(\frac{1}{m}\hat{\mathbf{B}}^\top \mathbf{X}_i^\top \mathbf{X}_i \hat{\mathbf{B}}\right)^{-1}\right\|_2 \left\|\frac{1}{m}\sum_{i=1}^{m} z_i^j \hat{\mathbf{B}}^\top \mathbf{x}_i^j\right\|_2 \leq \frac{\delta}{1-\delta}\sigma^2 \tag{68}$$

with probability at least $1 - n\exp\left(-110 k^2 \log(N)\right)$. It follows that with the same probability

$$\|\mathbf{G}\|_F^2 = \sum_{i=1}^{n} \|\mathbf{G}_i\|_2^2 \leq n\left(\frac{\delta}{1-\delta}\right)^2 \sigma^4 \tag{69}$$

$$\square$$

For all $i \in [n]$ we define $\mathbf{q}_i := \hat{\mathbf{B}}\mathbf{w}_i^+ - \hat{\mathbf{B}}^*\mathbf{w}_i^*$. The following lemma provides upper bounds on the norms of $\mathbf{w}_i^+$ and $\mathbf{q}_i$

**Lemma A.8.** *Let $\delta := c\frac{k^{3/2}\sqrt{\log(N)}}{\sqrt{m}}$ for some absolute constant $c$ and $\hat{\delta} = \delta/(1-\delta)$. Then with probability at least $1 - \exp\left(-105 k^2 \log(N)\right)$ we have*

$$\forall i \in [n] \quad \left\|\mathbf{w}_i^+\right\|_2 \leq 2\sqrt{k} + \sigma^2 \hat{\delta} \tag{70}$$

*Further with probability at least $1 - \exp\left(-105 k^2 \log(N)\right)$ we have*

$$\forall i \in [n] \quad \|\mathbf{q}_i\|_2 \leq 2\sqrt{k} \cdot dist\left(\hat{\mathbf{B}}, \hat{\mathbf{B}}^*\right) + \sigma^2 \hat{\delta} \tag{71}$$

*Proof.*

$$\left\|\mathbf{w}_i^+\right\|_2 \leq \left\|\hat{\mathbf{B}}^\top\right\|_2 \left\|\hat{\mathbf{B}}^*\right\|_2 \|\mathbf{w}_i^*\|_2 + \|\mathbf{F}_i\|_2 + \|\mathbf{G}_i\|_2 \tag{72}$$

$$\leq \|\mathbf{w}_i^*\|_2 + \hat{\delta}\|\mathbf{w}_i^*\|_2 \cdot \text{dist}\left(\hat{\mathbf{B}}, \hat{\mathbf{B}}^*\right) + \hat{\delta}\sigma^2 \tag{73}$$

$$\leq 2\sqrt{k} + \hat{\delta}\sigma^2 \tag{74}$$

where the first inequality comes from (20) and the third from Assumption 4.6. For the second inequality we take Union Bound over the failure probability of (51) and (58) and thus the above result holds with probability at least $1 - \exp\left(-107 k^2 \log(N)\right)$. Taking Union Bound for all $i \in [n]$ we get that with probability at least $1 - \exp\left(-105 k^2 \log(N)\right)$

$$\forall i \in [n] \quad \left\|\mathbf{w}_i^+\right\|_2 \leq 2\sqrt{k} + \sigma^2 \hat{\delta} \tag{75}$$

and the first result of the lemma follows. For the second part we have

$$\|\mathbf{q}_i\|_2 = \left\|\hat{\mathbf{B}}\mathbf{w}_i^+ - \hat{\mathbf{B}}^*\mathbf{w}_i^*\right\|_2 \leq \left\|\hat{\mathbf{B}}\hat{\mathbf{B}}^\top\hat{\mathbf{B}}^*\mathbf{w}_i^* + \hat{\mathbf{B}}\mathbf{F}_i + \hat{\mathbf{B}}\mathbf{G}_i - \hat{\mathbf{B}}^*\mathbf{w}_i^*\right\|_2 \tag{76}$$

$$\leq \left\|\left(\hat{\mathbf{B}}\hat{\mathbf{B}}^\top - \mathbf{I}_d\right)\hat{\mathbf{B}}^*\mathbf{w}_i^*\right\|_2 + \left\|\hat{\mathbf{B}}\mathbf{F}_i\right\|_2 + \left\|\hat{\mathbf{B}}\mathbf{G}_i\right\|_2 \tag{77}$$

$$\leq \left\|\hat{\mathbf{B}}_\perp\hat{\mathbf{B}}^*\right\|_2 \|\mathbf{w}_i^*\|_2 + \|\mathbf{F}_i\|_2 + \|\mathbf{G}_i\|_2 \tag{78}$$

$$\leq \mathrm{dist}\left(\hat{\mathbf{B}}, \hat{\mathbf{B}}^*\right)\|\mathbf{w}_i^*\|_2 + \mathrm{dist}\left(\hat{\mathbf{B}}, \hat{\mathbf{B}}^*\right)\hat{\delta}\|\mathbf{w}_i^*\|_2 + \sigma^2\hat{\delta} \tag{79}$$

$$\leq \mathrm{dist}\left(\hat{\mathbf{B}}, \hat{\mathbf{B}}^*\right)2\sqrt{k} + \sigma^2\hat{\delta} \tag{80}$$

where the first inequality comes from (20). For the forth inequality we take Union Bound over the failure probability of (51) and (58) and thus the above result holds with probability at least $1 - \exp\left(-107k^2\log(N)\right)$. Taking Union Bound for all $i \in [n]$ we get that with probability at least $1 - \exp\left(-105k^2\log(N)\right)$

$$\forall i \in [n] \quad \|\mathbf{q}_i\|_2 \leq 2\sqrt{k}\cdot\mathrm{dist}\left(\hat{\mathbf{B}}, \hat{\mathbf{B}}^*\right) + \sigma^2\hat{\delta} \tag{81}$$

$$\square$$

**Lemma A.9.** *Let* $\delta := c\frac{k^{3/2}\sqrt{\log(N)}}{\sqrt{m}}$ *for some absolute constant $c$ and $\hat{\delta} = \delta/(1-\delta)$. Then with probability at least $1 - \exp\left(-105d\right) - \exp\left(-105k^2\log(N)\right)$ we have*

$$\left\|\frac{1}{mn}\sum_{i=1}^n \mathbf{X}_i^\top\mathbf{Z}_i\mathbf{w}_i^{+\top}\right\|_2 \leq c\cdot\frac{\sigma^2\left(\sqrt{k} + \hat{\delta}\sigma^2\right)\sqrt{d}}{\sqrt{mn}} \tag{82}$$

*Proof.* Let $\mathcal{S}^{d-1}, \mathcal{S}^{k-1}$ denote the unit spheres in $d$ and $k$ dimensions and $\mathcal{N}_d, \mathcal{N}_k$ the 1/4-nets of cardinality $9^d$ and $9^k$, respectively. By equation 4.13 in Vershynin (2018) we have

$$\left\|\frac{1}{mn}\sum_{i=1}^n \mathbf{X}_i^\top\mathbf{Z}_i\mathbf{w}_i^{+\top}\right\|_2 \leq 2\max_{\mathbf{p}\in\mathcal{N}_d, \mathbf{y}\in\mathcal{N}_k}\mathbf{p}^\top\left(\sum_{i=1}^n\frac{1}{mn}\mathbf{X}_i^\top\mathbf{Z}_i\mathbf{w}_i^{+\top}\right)\mathbf{y} \tag{83}$$

$$=\leq 2\max_{\mathbf{p}\in\mathcal{N}_d, \mathbf{y}\in\mathcal{N}_k}\mathbf{p}^\top\left(\sum_{i=1}^n\sum_{j=1}^m\frac{z_i^j}{mn}\mathbf{x}_i^j\mathbf{w}_i^{+\top}\right)\mathbf{y} \tag{84}$$

$$=\leq 2\max_{\mathbf{p}\in\mathcal{N}_d, \mathbf{y}\in\mathcal{N}_k}\sum_{i=1}^n\sum_{j=1}^m\left(\frac{z_i^j}{mn}\left\langle\mathbf{x}_i^j, \mathbf{p}\right\rangle\left\langle\mathbf{w}_i^+, \mathbf{y}\right\rangle\right) \tag{85}$$

Notice that for any fixed $\mathbf{p}, \mathbf{y}$ and $\forall i \in [n], j \in [m]$ the random variables $\frac{z_i^j}{mn}\left\langle\mathbf{x}_i^j, \mathbf{p}\right\rangle\left\langle\mathbf{w}_i^+, \mathbf{y}\right\rangle$ are i.i.d. zero mean subexponentials with norm at most $\frac{C_3'\sigma^2\|\mathbf{w}_i^+\|_2}{mn}$, for some absolute constant $C_3'$. Conditioning on the event

$$\mathcal{E}_1 := \bigcap_{i=1}^n\left\{\left\|\mathbf{w}_i^+\right\|_2 \leq 2\sqrt{k} + \hat{\delta}\sigma^2\right\}, \tag{86}$$

which holds with probability at least $1 - \exp\left(-105k^2\log(N)\right)$ by Lemma A.8, we can invoke Bernstein's inequality to get

$$\mathbb{P}\left(\sum_{i=1}^n\sum_{j=1}^m\frac{z_i^j}{mn}\left\langle\mathbf{x}_i^j, \mathbf{p}\right\rangle\left\langle\mathbf{w}_i^+, \mathbf{y}\right\rangle \geq s\bigg|\mathcal{E}_1\right) \leq \exp\left(-c_4mn\min\left\{\frac{s^2}{\sigma^4\left(2\sqrt{k} + \sigma^2\hat{\delta}\right)^2}, \frac{s}{\sigma^2\left(2\sqrt{k} + \sigma^2\hat{\delta}\right)}\right\}\right) \tag{87}$$

Since $m > \frac{d \cdot C_3}{n_0} \geq \frac{d \cdot C_3}{n}$ by setting $s = \frac{\sigma^2 \left(2\sqrt{k} + \hat{\delta}\sigma^2\right)\sqrt{d \cdot C_3}}{8\sqrt{mn}}$ the above quantity simplifies as follows

$$\mathbb{P}\left(\sum_{i=1}^{n}\sum_{j=1}^{m}\frac{z_i^j}{mn}\left\langle \mathbf{x}_i^j, \mathbf{p}\right\rangle\left\langle \mathbf{w}_i^+, \mathbf{y}\right\rangle \geq \frac{\sigma^2\left(2\sqrt{k} + \hat{\delta}\sigma^2\right)\sqrt{d \cdot C_3}}{\sqrt{mn}}\middle|\mathcal{E}_1\right) \leq \exp\left(-c_4 mn \frac{s^2}{\sigma^4\left(2\sqrt{k} + \sigma^2\hat{\delta}\right)^2}\right) \quad (88)$$

$$\leq \exp\left(-C_3 \cdot c_4 \cdot d\right) \quad (89)$$

$$\leq \exp\left(-110d\right) \quad (90)$$

for $C_3$ large enough. Taking Union Bound over all points $\mathbf{p}, \mathbf{y}$ on the $\mathcal{N}_d, \mathcal{N}_k$ and using (85) we derive

$$\mathbb{P}\left(\left\|\frac{1}{mn}\sum_{i=1}^{n}\mathbf{X}_i^\top \mathbf{Z}_i \mathbf{w}_i^{+\top}\right\|_2 \geq \sqrt{C_3}\frac{\sigma^2\left(\sqrt{k} + \hat{\delta}\sigma^2\right)\sqrt{d}}{\sqrt{mn}}\middle|\mathcal{E}_1\right) \leq 9^{d+k}\exp\left(-110d\right) \quad (91)$$

$$\leq \exp\left(-105d\right) \quad (92)$$

and removing the conditioning on $\mathcal{E}_1$ we get

$$\mathbb{P}\left(\left\|\frac{1}{mn}\sum_{i=1}^{n}\mathbf{X}_i^\top \mathbf{Z}_i \mathbf{w}_i^{+\top}\right\|_2 \geq \sqrt{C_3}\frac{\sigma^2\left(\sqrt{k} + \hat{\delta}\sigma^2\right)\sqrt{d}}{\sqrt{mn}}\right) \leq \exp\left(-105d\right) + \mathbb{P}\left(\mathcal{E}_1^C\right) \quad (93)$$

$$\leq \exp\left(-105d\right) + \exp\left(-105k^2\log(N)\right) \quad (94)$$

Choosing $c$ large enough and taking the complementary event derives the result. $\qquad\square$

**Lemma A.10.** *Let* $\delta := c\frac{k^{3/2}\sqrt{\log(N)}}{\sqrt{m}}$ *for some absolute constant* $c$ *and* $\hat{\delta} = \delta/(1-\delta)$. *Then with probability at least* $1 - \exp\left(-100d\right) - \exp\left(-100k^2\log(N)\right)$ *we have*

$$\left\|\frac{1}{n}\sum_{i=1}^{n}\left(\frac{1}{m}\mathbf{X}_i^\top \mathbf{X}_i\left(\hat{\mathbf{B}}\mathbf{w}_i^+ - \hat{\mathbf{B}}^*\mathbf{w}_i^*\right) - \left(\hat{\mathbf{B}}\mathbf{w}_i^+ - \hat{\mathbf{B}}^*\mathbf{w}_i^*\right)\right)\mathbf{w}_i^{+\top}\right\|_2 \leq c \cdot \frac{\sqrt{d}\left(dist\left(\hat{\mathbf{B}}, \hat{\mathbf{B}}^*\right)k + \sqrt{k}\hat{\delta}\sigma^2 + \left(\hat{\delta}\sigma^2\right)^2\right)}{\sqrt{mn}} \quad (95)$$

*Proof.* Let us define the event

$$\mathcal{E}_2 := \bigcap_{i=1}^{n}\left\{\left\|\mathbf{w}_i^+\right\|_2 \leq 2\sqrt{k} + \hat{\delta}\sigma^2 \quad \bigcap \quad \left\|\mathbf{q}_i\right\|_2 \leq dist\left(\hat{\mathbf{B}}, \hat{\mathbf{B}}^*\right)2\sqrt{k} + \hat{\delta}\sigma^2\right\}, \quad (96)$$

which happens with probability at least $1 - \exp\left(-100k^2\log(N)\right)$ by Union Bound and Lemma A.8. For the rest of this proof we work conditioning on event $\mathcal{E}_2$. Recall that $q_i := \hat{\mathbf{B}}\mathbf{w}_i^+ - \hat{\mathbf{B}}^*\mathbf{w}_i^*$ and thus we can write

$$\frac{1}{n}\sum_{i=1}^{n}\left(\frac{1}{m}\mathbf{X}_i^\top \mathbf{X}_i\left(\hat{\mathbf{B}}\mathbf{w}_i^+ - \hat{\mathbf{B}}^*\mathbf{w}_i^*\right) - \left(\hat{\mathbf{B}}\mathbf{w}_i^+ - \hat{\mathbf{B}}^*\mathbf{w}_i^*\right)\right)\mathbf{w}_i^{+\top} = \frac{1}{n}\left(\sum_{i=1}^{n}\mathbf{X}_i^\top \mathbf{X}_i \mathbf{q}_i \mathbf{w}_i^{+\top} - \sum_{i=1}^{n}\mathbf{q}_i \mathbf{w}_i^{+\top}\right) \quad (97)$$

$$= \frac{1}{n}\left(\frac{1}{m}\sum_{i=1}^{n}\sum_{j=1}^{m}\left\langle \mathbf{x}_i^j, \mathbf{q}_i\right\rangle \mathbf{x}_i^j \mathbf{w}_i^{+\top} - \sum_{i=1}^{n}\mathbf{q}_i \mathbf{w}_i^{+\top}\right) \quad (98)$$

Let $\mathcal{S}^{d-1}, \mathcal{S}^{k-1}$ denote the unit spheres in $d$ and $k$ dimensions and $\mathcal{N}_d, \mathcal{N}_k$ the $1/4$-nets of cardinality $9^d$ and $9^k$, respectively. By equation 4.13 in Vershynin (2018) we have

$$\left\| \frac{1}{n} \sum_{i=1}^{n} \sum_{j=1}^{m} \frac{1}{m} \left\langle \mathbf{x}_i^j, \mathbf{q}_i \right\rangle \mathbf{x}_i^j \mathbf{w}_i^{+\top} - \frac{1}{n} \sum_{i=1}^{n} \mathbf{q}_i \mathbf{w}_i^{+\top} \right\|_2 \leq 2 \max_{\mathbf{p} \in \mathcal{N}_d, \mathbf{y} \in \mathcal{N}_k} \frac{1}{n} \mathbf{p}^\top \left( \sum_{i=1}^{n} \sum_{j=1}^{m} \frac{1}{m} \left\langle \mathbf{x}_i^j, \mathbf{q}_i \right\rangle \mathbf{x}_i^j \mathbf{w}_i^{+\top} - \frac{1}{n} \sum_{i=1}^{n} \mathbf{q}_i \mathbf{w}_i^{+\top} \right) \mathbf{y}$$
$$(99)$$

$$= 2 \max_{\mathbf{p} \in \mathcal{N}_d, \mathbf{y} \in \mathcal{N}_k} \frac{1}{mn} \sum_{i=1}^{n} \sum_{j=1}^{m} \left( \left\langle \mathbf{x}_i^j, \mathbf{q}_i \right\rangle \left\langle \mathbf{p}, \mathbf{x}_i^j \right\rangle \left\langle \mathbf{w}_i^+, \mathbf{y} \right\rangle - \left\langle \mathbf{p}, \mathbf{q}_i \right\rangle \left\langle \mathbf{w}_i^+, \mathbf{y} \right\rangle \right)$$
$$(100)$$

Notice that for any fixed $\mathbf{p}, \mathbf{y}$ the products $\left\langle \mathbf{x}_i^j, \mathbf{q}_i \right\rangle$ are i.i.d. subgaussians with norm at most $\tilde{c}_1 \|\mathbf{q}_i\|$ and $\left\langle \mathbf{p}, \mathbf{x}_i^j \right\rangle$ are i.i.d. subgaussians with norm at most $\tilde{c}_2 \|\mathbf{p}\|_2 = \tilde{c}_2$. Hence under the event $\mathcal{E}_2$ the product $\frac{1}{mn} \left\langle \mathbf{x}_i^j, \mathbf{q}_i \right\rangle \left\langle \mathbf{p}, \mathbf{x}_i^j \right\rangle \left\langle \mathbf{w}_i^+, \mathbf{y} \right\rangle$ are subexponentials with norm at most $\frac{C_4'}{mn} \left( \text{dist}\left( \hat{\mathbf{B}}, \hat{\mathbf{B}}^* \right) k + \sqrt{k} \hat{\delta} \sigma^2 + \left( \hat{\delta} \sigma^2 \right)^2 \right)$, for some constant $C_4'$. Also note that

$$\mathbb{E}\left[ \left\langle \mathbf{x}_i^j, \mathbf{q}_i \right\rangle \left\langle \mathbf{p}, \mathbf{x}_i^j \right\rangle \left\langle \mathbf{w}_i^+, \mathbf{y} \right\rangle - \left\langle \mathbf{p}, \mathbf{q}_i \right\rangle \left\langle \mathbf{w}_i^+, \mathbf{y} \right\rangle \right] = 0 \tag{101}$$

and thus applying Bernstein's inequality we get

$$\mathbb{P}\left( \frac{1}{mn} \sum_{i=1}^{n} \sum_{j=1}^{m} \left\langle \mathbf{x}_i^j, \mathbf{q}_i \right\rangle \left\langle \mathbf{p}, \mathbf{x}_i^j \right\rangle \left\langle \mathbf{w}_i^+, \mathbf{y} \right\rangle - \frac{1}{n} \sum_{i=1}^{n} \left\langle \mathbf{p}, \mathbf{q}_i \right\rangle \left\langle \mathbf{w}_i^+, \mathbf{y} \right\rangle \geq s \,\middle|\, \mathcal{E}_2 \right)$$

$$\leq \exp\left( -c_5 \cdot mn \min\left\{ \frac{s^2}{\left( \text{dist}\left( \hat{\mathbf{B}}, \hat{\mathbf{B}}^* \right) k + \sqrt{k} \hat{\delta} \sigma^2 + \left( \hat{\delta} \sigma^2 \right)^2 \right)^2}, \frac{s}{\left( \text{dist}\left( \hat{\mathbf{B}}, \hat{\mathbf{B}}^* \right) k + \sqrt{k} \hat{\delta} \sigma^2 + \left( \hat{\delta} \sigma^2 \right)^2 \right)} \right\} \right)$$
$$(102)$$

Since $m > \frac{d \cdot C_4}{n_0} \geq \frac{d \cdot C_4}{n}$ by setting $s = \frac{\sqrt{C_4 \cdot d} \left( \text{dist}(\hat{\mathbf{B}}, \hat{\mathbf{B}}^*) k + \sqrt{k} \hat{\delta} \sigma^2 + \left( \hat{\delta} \sigma^2 \right)^2 \right)}{2\sqrt{mn}}$ and taking Union Bound over all $\mathbf{p} \in \mathcal{N}_d, \mathbf{y} \in \mathcal{N}_k$ we derive

$$\mathbb{P}\left( \left\| \frac{1}{n} \sum_{i=1}^{n} \left( \frac{1}{m} \mathbf{X}_i^\top \mathbf{X}_i \left( \hat{\mathbf{B}} \mathbf{w}_i^+ - \hat{\mathbf{B}}^* \mathbf{w}_i^* \right) - \left( \hat{\mathbf{B}} \mathbf{w}_i^+ - \hat{\mathbf{B}}^* \mathbf{w}_i^* \right) \right) \mathbf{w}_i^{+\top} \right\|_2 \geq \frac{\sqrt{C_4 \cdot d} \left( \text{dist}\left( \hat{\mathbf{B}}, \hat{\mathbf{B}}^* \right) k + \sqrt{k} \hat{\delta} \sigma^2 + \left( \hat{\delta} \sigma^2 \right)^2 \right)}{\sqrt{mn}} \,\middle|\, \mathcal{E}_2 \right)$$

$$\leq 9^{d+k} \exp\left( \frac{-c_5 \cdot mn s^2}{\left( \text{dist}\left( \hat{\mathbf{B}}, \hat{\mathbf{B}}^* \right) k + \sqrt{k} \hat{\delta} \sigma^2 + \left( \hat{\delta} \sigma^2 \right)^2 \right)^2} \right)$$

$$\leq 9^{d+k} \exp\left( -C_4 \cdot c_5 \cdot d \right) \tag{103}$$

$$\leq 9^{d+k} \exp\left( -120d \right) \tag{104}$$

$$\leq \exp\left( -100d \right) \tag{105}$$

choosing a large enough constant $C_4$. Recall that $\mathbb{P}\left(\mathcal{E}_2^C\right) \leq \exp\left(-100k^2\log(N)\right)$. Hence by removing the conditioning on $\mathcal{E}_2$ we get that with probability at least $1 - \exp\left(-100d\right) - \exp\left(-100k^2\log(N)\right)$

$$\left\| \frac{1}{n}\sum_{i=1}^{n}\left(\frac{1}{m}\mathbf{X}_i^\top\mathbf{X}_i\left(\hat{\mathbf{B}}\mathbf{w}_i^+ - \hat{\mathbf{B}}^*\mathbf{w}_i^*\right) - \left(\hat{\mathbf{B}}\mathbf{w}_i^+ - \hat{\mathbf{B}}^*\mathbf{w}_i^*\right)\right)\mathbf{w}_i^{+\top}\right\|_2 \leq c\cdot\frac{\sqrt{d}\left(\text{dist}\left(\hat{\mathbf{B}},\hat{\mathbf{B}}^*\right)k + \sqrt{k}\hat{\delta}\sigma^2 + \left(\hat{\delta}\sigma^2\right)^2\right)}{\sqrt{mn}} \tag{106}$$

for sufficiently large $c$. $\qquad\square$

Having set all the building blocks we now proceed to the proof of Theorem 4.7.

**Theorem 4.7.** *Let Assumptions 4.3-4.6 hold. Further, let the following inequalities hold for the number of participating nodes and the batch size respectively, $n \geq n_0$ and $m \geq c_0\frac{(1+\sigma^2)k^3\kappa^4}{E_0^2}\max\left\{\log(N), d/n_0\right\}$, for some absolute constant $c_0$. Then* `FedRep-SRPFL` *with stepsize $\eta \leq \frac{1}{8\bar{\sigma}_{\max,*}^2}$, satisfies the following contraction inequality:*

$$dist\left(\mathbf{B}^{t+1},\mathbf{B}^*\right) \leq dist\left(\mathbf{B}^t,\mathbf{B}^*\right)\sqrt{1-a} + \frac{a}{\sqrt{\frac{n}{n_0}\left(1-a\right)}}, \tag{10}$$

*w.p. at least $1 - T\cdot\exp\left(-90\min\left\{d,k^2\log(N)\right\}\right)$, where $a = \frac{1}{2}\eta E_0\bar{\sigma}_{\min,*}^2 \leq \frac{1}{4}$.*

*Proof.* First let us recall the definition of $\delta := c\frac{k^{3/2}\sqrt{\log(N)}}{\sqrt{m}}$ for some absolute constant $c$ and $\hat{\delta} = \delta/(1-\delta)$. Further notice that for our choice of $m$ and sufficiently large $c_0$ we have the following useful inequality

$$\hat{\delta} = \frac{\delta}{1-\delta} \leq 2\delta \leq \frac{E_0}{20\cdot\kappa^2}\cdot\frac{1}{1+\sigma^2} \leq \frac{1}{20} \tag{107}$$

From the update scheme of our algorithm (18) we have

$$\mathbf{B}^+ = \hat{\mathbf{B}} - \frac{\eta}{mn}\left(\sum_{i=1}^{n}\mathbf{X}_i^\top\mathbf{X}_i\hat{\mathbf{B}}\mathbf{w}_i^+\mathbf{w}_i^{+\top} - \sum_{i=1}^{n}\mathbf{X}_i^\top\mathbf{X}_i\hat{\mathbf{B}}^*\mathbf{w}_i^*\mathbf{w}_i^{+\top} - \sum_{i=1}^{n}\mathbf{X}_i^\top\mathbf{Z}_i\mathbf{w}_i^{+\top}\right) \tag{108}$$

$$= \hat{\mathbf{B}} - \frac{\eta}{n}\left(\sum_{i=1}^{n}\left(\frac{1}{m}\mathbf{X}_i^\top\mathbf{X}_i\left(\hat{\mathbf{B}}\mathbf{w}_i^+ - \hat{\mathbf{B}}^*\mathbf{w}_i^*\right) - \left(\hat{\mathbf{B}}\mathbf{w}_i^+ - \hat{\mathbf{B}}^*\mathbf{w}_i^*\right)\right)\mathbf{w}_i^{+\top}\right)$$

$$- \frac{\eta}{n}\sum_{i=1}^{n}\left(\hat{\mathbf{B}}\mathbf{w}_i^+ - \hat{\mathbf{B}}^*\mathbf{w}_i^*\right)\mathbf{w}_i^{+\top} + \frac{\eta}{n}\sum_{i=1}^{n}\frac{1}{m}\mathbf{X}_i^\top\mathbf{Z}_i\mathbf{w}_i^{+\top} \tag{109}$$

where we added and subtracted terms. Multiplying both sides by $\hat{\mathbf{B}}_\perp^{*\top}$ we get

$$\hat{\mathbf{B}}_\perp^{*\top}\mathbf{B}^+ = \hat{\mathbf{B}}_\perp^{*\top}\hat{\mathbf{B}} - \frac{\eta}{n}\hat{\mathbf{B}}_\perp^{*\top}\left(\sum_{i=1}^{n}\left(\frac{1}{m}\mathbf{X}_i^\top\mathbf{X}_i\left(\hat{\mathbf{B}}\mathbf{w}_i^+ - \hat{\mathbf{B}}^*\mathbf{w}_i^*\right) - \left(\hat{\mathbf{B}}\mathbf{w}_i^+ - \hat{\mathbf{B}}^*\mathbf{w}_i^*\right)\right)\mathbf{w}_i^{+\top}\right)$$

$$- \frac{\eta}{n}\sum_{i=1}^{n}\left(\hat{\mathbf{B}}_\perp^{*\top}\hat{\mathbf{B}}\mathbf{w}_i^+ - \hat{\mathbf{B}}_\perp^{*\top}\hat{\mathbf{B}}^*\mathbf{w}_i^*\right)\mathbf{w}_i^{+\top} + \frac{\eta}{n}\hat{\mathbf{B}}_\perp^{*\top}\sum_{i=1}^{n}\frac{1}{m}\mathbf{X}_i^\top\mathbf{Z}_i\mathbf{w}_i^{+\top} \tag{110}$$

$$= \hat{\mathbf{B}}_\perp^{*\top}\hat{\mathbf{B}}\left(\mathbf{I}_k - \frac{\eta}{n}\sum_{i=1}^{n}\mathbf{w}_i^+\mathbf{w}_i^{+\top}\right) + \frac{\eta}{n}\hat{\mathbf{B}}_\perp^{*\top}\sum_{i=1}^{n}\frac{1}{m}\mathbf{X}_i^\top\mathbf{Z}_i\mathbf{w}_i^{+\top}$$

$$- \frac{\eta}{n}\hat{\mathbf{B}}_\perp^{*\top}\left(\sum_{i=1}^{n}\left(\frac{1}{m}\mathbf{X}_i^\top\mathbf{X}_i\left(\hat{\mathbf{B}}\mathbf{w}_i^+ - \hat{\mathbf{B}}^*\mathbf{w}_i^*\right) - \left(\hat{\mathbf{B}}\mathbf{w}_i^+ - \hat{\mathbf{B}}^*\mathbf{w}_i^*\right)\right)\mathbf{w}_i^{+\top}\right) \tag{111}$$

where the second equality holds since $\hat{\mathbf{B}}_\perp^{*\top}\hat{\mathbf{B}}^* = 0$. Recall that from the $QR$ decomposition of $\mathbf{B}^+$ we have $\mathbf{B}^+ = \hat{\mathbf{B}}^+\mathbf{R}^+$. Hence multiplying by $(\mathbf{R}^+)^{-1}$ and taking both sides the norm we derive

$$\text{dist}\left(\hat{\mathbf{B}}^+, \hat{\mathbf{B}}^*\right) \leq \left\|\hat{\mathbf{B}}_\perp^{*\top}\hat{\mathbf{B}}\left(\mathbf{I}_k - \frac{\eta}{n}\sum_{i=1}^n \mathbf{w}_i^+\mathbf{w}_i^{+\top}\right)\right\|_2 \left\|(\mathbf{R}^+)^{-1}\right\|_2 + \left\|\frac{\eta}{n}\hat{\mathbf{B}}_\perp^{*\top}\sum_{i=1}^n \frac{1}{m}\mathbf{X}_i^\top\mathbf{Z}_i\mathbf{w}_i^{+\top}\right\|_2 \left\|(\mathbf{R}^+)^{-1}\right\|_2$$
$$+ \left\|\frac{\eta}{n}\hat{\mathbf{B}}_\perp^{*\top}\left(\sum_{i=1}^n \left(\frac{1}{m}\mathbf{X}_i^\top\mathbf{X}_i\left(\hat{\mathbf{B}}\mathbf{w}_i^+ - \hat{\mathbf{B}}^*\mathbf{w}_i^*\right) - \left(\hat{\mathbf{B}}\mathbf{w}_i^+ - \hat{\mathbf{B}}^*\mathbf{w}_i^*\right)\right)\mathbf{w}_i^{+\top}\right)\right\|_2 \left\|(\mathbf{R}^+)^{-1}\right\|_2 \tag{112}$$

Let us define

$$A_1 := \text{dist}\left(\hat{\mathbf{B}}, \hat{\mathbf{B}}^*\right)\left\|\mathbf{I}_k - \frac{\eta}{n}\sum_{i=1}^n \mathbf{w}_i^+\mathbf{w}_i^{+\top}\right\|_2 \tag{113}$$

$$A_2 := \left\|\frac{\eta}{n}\hat{\mathbf{B}}_\perp^{*\top}\sum_{i=1}^n \frac{1}{m}\mathbf{X}_i^\top\mathbf{Z}_i\mathbf{w}_i^{+\top}\right\|_2 \tag{114}$$

$$A_3 := \left\|\frac{\eta}{n}\hat{\mathbf{B}}_\perp^{*\top}\left(\sum_{i=1}^n \left(\frac{1}{m}\mathbf{X}_i^\top\mathbf{X}_i\left(\hat{\mathbf{B}}\mathbf{w}_i^+ - \hat{\mathbf{B}}^*\mathbf{w}_i^*\right) - \left(\hat{\mathbf{B}}\mathbf{w}_i^+ - \hat{\mathbf{B}}^*\mathbf{w}_i^*\right)\right)\mathbf{w}_i^{+\top}\right)\right\|_2 \tag{115}$$

so that the following inequality holds

$$\text{dist}\left(\hat{\mathbf{B}}^+, \hat{\mathbf{B}}^*\right) \leq (A_1 + A_2 + A_3)\left\|(\mathbf{R}^+)^{-1}\right\|_2 \tag{116}$$

For the rest of the proof we will work conditioning on the intersection of the events

$$\mathcal{E}_2 := \bigcap_{i=1}^n \left\{\left\|\mathbf{w}_i^+\right\|_2 \leq 2\sqrt{k} + \hat{\delta}\sigma^2 \quad \bigcap \quad \|\mathbf{q}_i\|_2 \leq \text{dist}\left(\hat{\mathbf{B}}, \hat{\mathbf{B}}^*\right) 2\sqrt{k} + \hat{\delta}\sigma^2\right\} \tag{117}$$

$$\mathcal{E}_3 := \left\{\|\mathbf{F}\|_F \leq \text{dist}\left(\hat{\mathbf{B}}, \hat{\mathbf{B}}^*\right)\hat{\delta}\left\|\mathbf{W}^*\right\|_2 \quad \bigcap \quad \|\mathbf{G}\|_F \leq \hat{\delta}\sqrt{n}\sigma^2\right\} \tag{118}$$

$$\mathcal{E}_4 := \left\{\left\|\frac{1}{mn}\sum_{i=1}^n \mathbf{X}_i^\top\mathbf{Z}_i\mathbf{w}_i^{+\top}\right\|_2 \leq c \cdot \frac{\sigma^2\left(\sqrt{k} + \hat{\delta}\sigma^2\right)\sqrt{d}}{\sqrt{mn}}\right\} \tag{119}$$

$$\mathcal{E}_5 := \left\{\left\|\frac{1}{n}\sum_{i=1}^n \left(\frac{1}{m}\mathbf{X}_i^\top\mathbf{X}_i\left(\hat{\mathbf{B}}\mathbf{w}_i^+ - \hat{\mathbf{B}}^*\mathbf{w}_i^*\right) - \left(\hat{\mathbf{B}}\mathbf{w}_i^+ - \hat{\mathbf{B}}^*\mathbf{w}_i^*\right)\right)\mathbf{w}_i^{+\top}\right\|_2 \leq \frac{c\sqrt{d}\left(\text{dist}\left(\hat{\mathbf{B}}, \hat{\mathbf{B}}^*\right)k + \sqrt{k}\hat{\delta}\sigma^2 + \left(\hat{\delta}\sigma^2\right)^2\right)}{\sqrt{mn}}\right\} \tag{120}$$

which happens with probability at least $1 - \exp\left(-90d\right) - \exp\left(-90k^2\log(N)\right)$ by Union Bound on the failure probability of (52), (59), (82), (95) and (96).

We will now provide bounds for each of the terms of interest in (116), starting from $A_1$. Notice that by (26) we have

$$\lambda_{\max}\left(\mathbf{W}^+\mathbf{W}^+\right) = \left\|\mathbf{W}^+\right\|_2^2 = \left\|\mathbf{W}^*\hat{\mathbf{B}}^*\hat{\mathbf{B}} + \mathbf{F} + \mathbf{G}\right\|_2^2 \tag{121}$$

$$\leq 2\left\|\mathbf{W}^*\right\|_2^2 + 4\left\|\mathbf{F}\right\|_2^2 + 4\left\|\mathbf{G}\right\|_2^2 \tag{122}$$

$$\leq 2\left\|\mathbf{W}^*\right\|_2^2 + \text{dist}\left(\hat{\mathbf{B}}, \hat{\mathbf{B}}^*\right)4\hat{\delta}^2\left\|\mathbf{W}^*\right\|_2^2 + 4\hat{\delta}^2\sigma^4 n \tag{123}$$

$$\leq 4\left(\left\|\mathbf{W}^*\right\|_2^2 + n\right) \tag{124}$$

$$\leq 4n\left(\bar{\sigma}_{\max,*}^2 + 1\right) \tag{125}$$

where in the last inequality we use the fact that $\|\mathbf{W}^*\|_2 = \sqrt{n} \cdot \bar{\sigma}_{\max,*}$. Since $\eta < \left(\bar{\sigma}_{\max,*}^2 + 1\right)^{-1}$ the matrix $\mathbf{I}_k - \frac{\eta}{n}\mathbf{W}^{+\top}\mathbf{W}^+$ is positive definite. Thus we have

$$\left\|\mathbf{I}_k - \frac{\eta}{n}\mathbf{W}^{+\top}\mathbf{W}^+\right\|_2 \leq 1 - \frac{\eta}{n}\lambda_{\min}\left(\mathbf{W}^{+\top}\mathbf{W}^+\right) \tag{126}$$

$$\leq 1 - \frac{\eta}{n}\lambda_{\min}\left(\left(\mathbf{W}^*\hat{\mathbf{B}}^*\hat{\mathbf{B}} + \mathbf{F} + \mathbf{G}\right)^\top \left(\mathbf{W}^*\hat{\mathbf{B}}^*\hat{\mathbf{B}} + \mathbf{F} + \mathbf{G}\right)\right) \tag{127}$$

$$\leq 1 - \frac{\eta}{n}\left(\sigma_{\min}^2\left(\mathbf{W}^*\hat{\mathbf{B}}^*\hat{\mathbf{B}}\right) - \sigma_{\min}^2(\mathbf{F}) - \sigma_{\min}^2(\mathbf{G})\right)$$
$$+ \frac{2\eta}{n}\left(\sigma_{\max}\left(\mathbf{F}^\top\mathbf{W}^*\hat{\mathbf{B}}^{*\top}\hat{\mathbf{B}}\right) + \sigma_{\max}\left(\mathbf{F}^\top\mathbf{G}\right) + \sigma_{\max}\left(\mathbf{G}^\top\mathbf{W}^*\hat{\mathbf{B}}^{*\top}\hat{\mathbf{B}}\right)\right) \tag{128}$$

$$\leq 1 - \frac{\eta}{n}\left(\sigma_{\min}^2\left(\mathbf{W}^*\right)\sigma_{\min}^2\left(\hat{\mathbf{B}}^{*\top}\hat{\mathbf{B}}\right) + 2\left\|\hat{\mathbf{B}}^{*\top}\hat{\mathbf{B}}\right\|_2\left(\sigma_{\max}\left(\mathbf{F}^\top\mathbf{W}^*\right) + \sigma_{\max}\left(\mathbf{G}^\top\mathbf{W}^*\right)\right)\right)$$
$$+ \frac{2\eta}{n}\sigma_{\max}(\mathbf{F})\sigma_{\max}(\mathbf{G}) \tag{129}$$

$$\leq 1 - \eta \cdot \bar{\sigma}_{\min,*}^2 \cdot \sigma_{\min}^2\left(\hat{\mathbf{B}}^{*\top}\hat{\mathbf{B}}\right) + \frac{2\eta}{n}\left(\|\mathbf{F}\|_2 + \|\mathbf{G}\|_2\right)\|\mathbf{W}^*\|_2 + \frac{2\eta}{n}\left(\|\mathbf{F}\|_2 \cdot \|\mathbf{G}\|_2\right) \tag{130}$$

where we used that the norms of $\hat{\mathbf{B}}^*$ and $\hat{\mathbf{B}}$ are 1 since the matrices are orthonormal and $\bar{\sigma}_{\min,*} \leq \sigma_{\min}(\mathbf{W}^*)$. Recall that we operate under $\mathcal{E}_3$ and thus we can further write

$$\left\|\mathbf{I}_k - \frac{\eta}{n}\mathbf{W}^{+\top}\mathbf{W}^+\right\|_2 \leq 1 - \eta \cdot \bar{\sigma}_{\min,*}^2 \cdot \sigma_{\min}^2\left(\hat{\mathbf{B}}^{*\top}\hat{\mathbf{B}}\right) + \frac{2\eta}{n}\left(\text{dist}\left(\hat{\mathbf{B}},\hat{\mathbf{B}}^*\right)\hat{\delta}\|\mathbf{W}^*\|_2 + \sqrt{n}\hat{\delta}\sigma^2\right)\|\mathbf{W}^*\|_2$$
$$+ \frac{2\eta}{n}\left(\text{dist}\left(\hat{\mathbf{B}},\hat{\mathbf{B}}^*\right)\hat{\delta}^2\sigma^2\sqrt{n}\|\mathbf{W}^*\|_2\right) \tag{131}$$

$$\leq 1 - \eta \cdot \bar{\sigma}_{\min,*}^2 \cdot \sigma_{\min}^2\left(\hat{\mathbf{B}}^{*\top}\hat{\mathbf{B}}\right) + 2\eta\left(\hat{\delta}\frac{\|\mathbf{W}^*\|_2^2}{n} + \hat{\delta}\sigma^2\frac{\|\mathbf{W}^*\|_2}{\sqrt{n}} + \hat{\delta}^2\sigma^4\frac{\|\mathbf{W}^*\|}{\sqrt{n}}\right) \tag{132}$$

$$\leq 1 - \eta \cdot \bar{\sigma}_{\min,*}^2 \cdot \sigma_{\min}^2\left(\hat{\mathbf{B}}^{*\top}\hat{\mathbf{B}}\right) + 3\eta\left(\frac{E_0\bar{\sigma}_{\min,*}^2}{20\bar{\sigma}_{\max,*}^2} \cdot \bar{\sigma}_{\max,*}^2\right) \tag{133}$$

$$\leq 1 - \eta \cdot \bar{\sigma}_{\min,*}^2 \cdot \sigma_{\min}^2\left(\hat{\mathbf{B}}^{*\top}\hat{\mathbf{B}}\right) + \frac{1}{6}\eta E_0\bar{\sigma}_{\min,*}^2 \tag{134}$$

where we upper bound dist$\left(\hat{\mathbf{B}},\hat{\mathbf{B}}^*\right)$ by 1, $\hat{\delta} \leq \frac{E_0}{20 \cdot \kappa^2} \cdot \frac{1}{1+\sigma^2}$ and use Assumption 4.4 in the third inequality. Further by the definition of $E_0 := 1 - \text{dist}^2\left(\hat{\mathbf{B}}^0,\hat{\mathbf{B}}^*\right) \leq \sigma_{\min}^2\left(\hat{\mathbf{B}}^{*\top},\hat{\mathbf{B}}\right)$ we have

$$\left\|\mathbf{I}_k - \frac{\eta}{n}\mathbf{W}^{+\top}\mathbf{W}^+\right\|_2 \leq 1 - \eta E_0\bar{\sigma}_{\min,*}^2 + \frac{1}{6}\eta E_0\bar{\sigma}_{\min,*}^2 \tag{135}$$

and it follows immediately that

$$A_1 \leq \text{dist}\left(\hat{\mathbf{B}},\hat{\mathbf{B}}^*\right)\left(1 - \eta E_0\bar{\sigma}_{\min,*}^2 + \frac{1}{6}\eta E_0\bar{\sigma}_{\min,*}^2\right) \tag{136}$$

Further since we operate under $\mathcal{E}_4$ (119) and $\left\|\hat{\mathbf{B}}_\perp^*\right\|_2 = 1$ we have

$$A_2 \leq \eta c\sigma^2\left(\sqrt{k}+1\right)\frac{\sqrt{d}}{\sqrt{mn}} \tag{137}$$

and since we operate under $\mathcal{E}_5$ (120) we obtain

$$A_3 \leq \eta c\left(\text{dist}\left(\hat{\mathbf{B}},\hat{\mathbf{B}}^*\right)k + \sqrt{k}+1\right)\frac{\sqrt{d}}{\sqrt{mn}} \tag{138}$$

Combining (116), (136), (137) and (138) we get

$$\text{dist}\left(\hat{\mathbf{B}}^+, \hat{\mathbf{B}}^*\right) \leq \text{dist}\left(\hat{\mathbf{B}}, \hat{\mathbf{B}}^*\right)\left(1 - \frac{5}{6}\eta E_0 \bar{\sigma}_{\min,*}^2 + \eta c k \frac{\sqrt{d}}{\sqrt{mn}}\right) \cdot \left\|(\mathbf{R}^+)^{-1}\right\|_2$$
$$+ \eta c\left(\sqrt{k} + 1\right)\left(\sigma^2 + 1\right)\frac{\sqrt{d}}{\sqrt{mn}} \cdot \left\|(\mathbf{R}^+)^{-1}\right\|_2 \tag{139}$$

The last part of the proof focuses on bounding $\left\|(\mathbf{R}^+)^{-1}\right\|_2$.

Let us define

$$\mathbf{S} := \sum_{i=1}^n \frac{1}{m}\mathbf{X}_i^\top \mathbf{X}_i \left(\hat{\mathbf{B}}\mathbf{w}_i^+ - \hat{\mathbf{B}}^*\mathbf{w}_i^*\right)\mathbf{w}_i^{+\top} \tag{140}$$

$$\mathbf{E} := \sum_{i=1}^n \frac{1}{m}\mathbf{X}_i^\top \mathbf{Z}_i \mathbf{w}_i^{+\top} \tag{141}$$

and hence (108) takes the form

$$\mathbf{B}^+ = \hat{\mathbf{B}} - \frac{\eta}{n}\mathbf{S} + \frac{\eta}{n}\mathbf{E} \tag{142}$$

and also

$$\mathbf{B}^{+\top}\mathbf{B}^+ = \hat{\mathbf{B}}^\top\hat{\mathbf{B}} - \frac{\eta}{n}\left(\hat{\mathbf{B}}^\top\mathbf{S} + \mathbf{S}^\top\hat{\mathbf{B}}\right) + \frac{\eta}{n}\left(\hat{\mathbf{B}}^\top\mathbf{E} + \mathbf{E}^\top\hat{\mathbf{B}}\right) + \frac{\eta^2}{n^2}\mathbf{S}^\top\mathbf{S} - \frac{\eta^2}{n^2}\left(\mathbf{E}^\top\mathbf{S} + \mathbf{S}^\top\mathbf{E}\right) + \frac{\eta^2}{n^2}\mathbf{E}^\top\mathbf{E} \tag{143}$$

$$= \mathbf{I}_k - \frac{\eta}{n}\left(\hat{\mathbf{B}}^\top\mathbf{S} + \mathbf{S}^\top\hat{\mathbf{B}}\right) + \frac{\eta}{n}\left(\hat{\mathbf{B}}^\top\mathbf{E} + \mathbf{E}^\top\hat{\mathbf{B}}\right) - \frac{\eta^2}{n^2}\left(\mathbf{E}^\top\mathbf{S} + \mathbf{S}^\top\mathbf{E}\right) + \frac{\eta^2}{n^2}\mathbf{E}^\top\mathbf{E} + \frac{\eta^2}{n^2}\mathbf{S}^\top\mathbf{S} \tag{144}$$

By Weyl's inequality and since $\mathbf{R}^{+\top}\mathbf{R}^+ = \hat{\mathbf{B}}^{+\top}\hat{\mathbf{B}}^+$ we derive

$$\sigma_{\min}^2\left(\mathbf{R}^+\right) \geq 1 - \frac{\eta}{n}\lambda_{\max}\left(\hat{\mathbf{B}}^\top\mathbf{S} + \mathbf{S}^\top\hat{\mathbf{B}}\right) - \frac{\eta}{n}\lambda_{\max}\left(\hat{\mathbf{B}}^\top\mathbf{E} + \mathbf{E}^\top\hat{\mathbf{B}}\right) - \frac{\eta^2}{n^2}\lambda_{\max}\left(\mathbf{E}^\top\mathbf{S} + \mathbf{S}^\top\mathbf{E}\right) \tag{145}$$

Let us further define

$$R_1 := \frac{\eta}{n}\lambda_{\max}\left(\hat{\mathbf{B}}^\top\mathbf{S} + \mathbf{S}^\top\hat{\mathbf{B}}\right) \tag{146}$$

$$R_2 := \frac{\eta^2}{n^2}\lambda_{\max}\left(\mathbf{E}^\top\mathbf{S} + \mathbf{S}^\top\mathbf{E}\right) \tag{147}$$

$$R_3 := \frac{\eta}{n}\lambda_{\max}\left(\hat{\mathbf{B}}^\top\mathbf{E} + \mathbf{E}^\top\hat{\mathbf{B}}\right) \tag{148}$$

So that we can succinctly rewrite the above inequality as follows

$$\sigma_{\min}^2\left(\mathbf{R}^+\right) \geq 1 - R_1 - R_2 - R_3 \tag{149}$$

We work to bound separately each of the three terms.

$$R_1 = \frac{2\eta}{n}\max_{\mathbf{p}:\|\mathbf{p}\|_2=1}\mathbf{p}^\top\hat{\mathbf{B}}^\top\mathbf{S}\mathbf{p} \tag{150}$$

$$= \max_{\mathbf{p}:\|\mathbf{p}\|_2=1}\frac{2\eta}{n}\mathbf{p}^\top\hat{\mathbf{B}}^\top\left[\left(\sum_{i=1}^n\left(\frac{1}{m}\mathbf{X}_i^\top\mathbf{X}_i\left(\hat{\mathbf{B}}\mathbf{w}_i^+ - \hat{\mathbf{B}}^*\mathbf{w}_i^*\right) - \left(\hat{\mathbf{B}}\mathbf{w}_i^+ - \hat{\mathbf{B}}^*\mathbf{w}_i^*\right)\right)\mathbf{w}_i^{+\top}\right)\right]\mathbf{p}$$
$$+ \max_{\mathbf{p}:\|\mathbf{p}\|_2=1}\frac{2\eta}{n}\mathbf{p}^\top\hat{\mathbf{B}}^\top\left[\sum_{i=1}^n\left(\hat{\mathbf{B}}\mathbf{w}_i^+ - \hat{\mathbf{B}}^*\mathbf{w}_i^*\right)\mathbf{w}_i^{+\top}\right]\mathbf{p} \tag{151}$$

and since we operate under $\mathcal{E}_5$ (120) the above simplifies to

$$R_1 \leq 2\eta \left\|\hat{\mathbf{B}}\right\|_2 c \left(\text{dist}\left(\hat{\mathbf{B}}, \hat{\mathbf{B}}^*\right) k + \sqrt{k} + 1\right) \frac{\sqrt{d}}{\sqrt{mn}} + \max_{\mathbf{p}: \|\mathbf{p}\|_2 = 1} \frac{2\eta}{n} \mathbf{p}^\top \hat{\mathbf{B}}^\top \left[\sum_{i=1}^n \left(\hat{\mathbf{B}} \mathbf{w}_i^+ - \hat{\mathbf{B}}^* \mathbf{w}_i^*\right) \mathbf{w}_i^{+\top}\right] \mathbf{p} \quad (152)$$

$$\leq 3\eta c \left(\text{dist}\left(\hat{\mathbf{B}}, \hat{\mathbf{B}}^*\right) k + \sqrt{k}\right) \frac{\sqrt{d}}{\sqrt{mn}} + \max_{\mathbf{p}: \|\mathbf{p}\|_2 = 1} \frac{2\eta}{n} \mathbf{p}^\top \hat{\mathbf{B}}^\top \left[\sum_{i=1}^n \left(\hat{\mathbf{B}} \mathbf{w}_i^+ - \hat{\mathbf{B}}^* \mathbf{w}_i^*\right) \mathbf{w}_i^{+\top}\right] \mathbf{p} \quad (153)$$

We focus on the second term and using (20) we get

$$\frac{2\eta}{n} \mathbf{p}^\top \hat{\mathbf{B}}^\top \left[\sum_{i=1}^n \left(\hat{\mathbf{B}} \mathbf{w}_i^+ - \hat{\mathbf{B}}^* \mathbf{w}_i^*\right) \mathbf{w}_i^{+\top}\right] \mathbf{p} = \frac{2\eta}{n} \cdot \text{tr}\left[\sum_{i=1}^n \left(\hat{\mathbf{B}} \mathbf{w}_i^+ - \hat{\mathbf{B}}^* \mathbf{w}_i^*\right) \mathbf{w}_i^{+\top} \mathbf{p} \mathbf{p}^\top \hat{\mathbf{B}}^\top\right] \quad (154)$$

$$= \frac{2\eta}{n} \cdot \text{tr}\left[\sum_{i=1}^n \left(\hat{\mathbf{B}} \mathbf{w}_i^+ - \hat{\mathbf{B}}^* \mathbf{w}_i^*\right) \left(\hat{\mathbf{B}}^\top \hat{\mathbf{B}}^* \mathbf{w}_i^* + \mathbf{F}_i + \mathbf{G}_i\right)^\top \mathbf{p} \mathbf{p}^\top \hat{\mathbf{B}}^\top\right] \quad (155)$$

We bound each term separately and to this end we define

$$T_1 := \frac{2\eta}{n} \cdot \text{tr}\left[\sum_{i=1}^n \left(\hat{\mathbf{B}} \mathbf{w}_i^+ - \hat{\mathbf{B}}^* \mathbf{w}_i^*\right) \mathbf{w}_i^{*\top} \hat{\mathbf{B}}^{*\top} \hat{\mathbf{B}} \mathbf{p} \mathbf{p}^\top \hat{\mathbf{B}}^\top\right] \quad (156)$$

$$T_2 := \frac{2\eta}{n} \cdot \text{tr}\left[\sum_{i=1}^n \left(\hat{\mathbf{B}} \mathbf{w}_i^+ - \hat{\mathbf{B}}^* \mathbf{w}_i^*\right) \mathbf{F}_i^\top \mathbf{p} \mathbf{p}^\top \hat{\mathbf{B}}^\top\right] \quad (157)$$

$$T_3 := \frac{2\eta}{n} \cdot \text{tr}\left[\sum_{i=1}^n \left(\hat{\mathbf{B}} \mathbf{w}_i^+ - \hat{\mathbf{B}}^* \mathbf{w}_i^*\right) \mathbf{G}_i^\top \mathbf{p} \mathbf{p}^\top \hat{\mathbf{B}}^\top\right] \quad (158)$$

such that (155) can be expressed as

$$\frac{2\eta}{n} \mathbf{p}^\top \hat{\mathbf{B}}^\top \left[\sum_{i=1}^n \left(\hat{\mathbf{B}} \mathbf{w}_i^+ - \hat{\mathbf{B}}^* \mathbf{w}_i^*\right) \mathbf{w}_i^{+\top}\right] \mathbf{p} = T_1 + T_2 + T_3 \quad (159)$$

Further expanding $T_1$ we have

$$T_1 = \frac{2\eta}{n} \text{tr}\left[\sum_{i=1}^n \left(\hat{\mathbf{B}} \hat{\mathbf{B}}^\top \hat{\mathbf{B}}^* \mathbf{w}_i^* \mathbf{w}_i^{*\top} + \hat{\mathbf{B}} \mathbf{F}_i \mathbf{w}_i^{*\top} + \hat{\mathbf{B}} \mathbf{G}_i \mathbf{w}_i^{*\top} - \hat{\mathbf{B}}^* \mathbf{w}_i^* \mathbf{w}_i^{*\top}\right) \hat{\mathbf{B}}^{*\top} \hat{\mathbf{B}} \mathbf{p} \mathbf{p}^\top \hat{\mathbf{B}}^\top\right] \quad (160)$$

$$= \frac{2\eta}{n} \text{tr}\left[\hat{\mathbf{B}}^\top \left(\hat{\mathbf{B}} \hat{\mathbf{B}}^\top - \mathbf{I}_d\right) \sum_{i=1}^n \left(\hat{\mathbf{B}}^{*\top} \mathbf{w}_i^* \mathbf{w}_i^{*\top}\right) \hat{\mathbf{B}}^{*\top} \hat{\mathbf{B}} \mathbf{p} \mathbf{p}^\top\right] + \frac{2\eta}{n} \text{tr}\left[\sum_{i=1}^n \left(\hat{\mathbf{B}} \mathbf{F}_i \mathbf{w}_i^{*\top}\right) \hat{\mathbf{B}}^{*\top} \hat{\mathbf{B}} \mathbf{p} \mathbf{p}^\top \hat{\mathbf{B}}^\top\right]$$

$$+ \frac{2\eta}{n} \text{tr}\left[\sum_{i=1}^n \left(\hat{\mathbf{B}} \mathbf{G}_i \mathbf{w}_i^{*\top}\right) \hat{\mathbf{B}}^{*\top} \hat{\mathbf{B}} \mathbf{p} \mathbf{p}^\top \hat{\mathbf{B}}^\top\right] \quad (161)$$

$$= \frac{2\eta}{n} \text{tr}\left[\hat{\mathbf{B}}^\top \hat{\mathbf{B}} \left(\mathbf{F}^\top + \mathbf{G}^\top\right) \mathbf{W}^* \hat{\mathbf{B}}^{*\top} \hat{\mathbf{B}} \mathbf{p} \mathbf{p}^\top\right] \quad (162)$$

$$\leq \frac{2\eta}{n} \left(\|\mathbf{F}\|_F + \|\mathbf{G}\|_F\right) \|\mathbf{W}^*\|_2 \quad (163)$$

where in the first equality we expand $w_i^+$ via (20) and in the third equality we use that $\hat{\mathbf{B}}^\top \left(\hat{\mathbf{B}} \hat{\mathbf{B}}^\top - \mathbf{I}_d\right) = 0$ and $\mathbf{F}^\top = \sum_{i=1}^n \mathbf{F}_i \mathbf{w}_i^{*\top}$, $\mathbf{G}^\top = \sum_{i=1}^n \mathbf{G}_i \mathbf{w}_i^{*\top}$. The inequality is obtained by noticing that the norms of the orthonormal $\hat{\mathbf{B}}, \hat{\mathbf{B}}^*$ is one and also $\left\|\mathbf{p} \mathbf{p}^\top\right\|_2 \leq 1$. Conditioning on $\mathcal{E}_3$ (118) we can further simplify as follows

$$T_1 \leq \frac{2\eta}{n} \left(\text{dist}\left(\hat{\mathbf{B}}, \hat{\mathbf{B}}^*\right) \hat{\delta} \|\mathbf{W}^*\|_2^2 + \hat{\delta} \sigma^2 \sqrt{n} \|\mathbf{W}\|_2\right) \quad (164)$$

$$\leq \frac{1}{10} \eta E_0 \bar{\sigma}_{\min,*}^2 \quad (165)$$

We now turn our attention to $T_2$

$$T_2 = \frac{2\eta}{n} \cdot \text{tr}\left[\sum_{i=1}^{n}\left(\hat{\mathbf{B}}\hat{\mathbf{B}}^{\top}\hat{\mathbf{B}}^{*}\mathbf{w}_i^{*} + \hat{\mathbf{B}}\mathbf{F}_i + \hat{\mathbf{B}}\mathbf{G}_i - \hat{\mathbf{B}}^{*}\mathbf{w}_i^{*}\right)\mathbf{F}_i^{\top}\mathbf{pp}^{\top}\hat{\mathbf{B}}^{\top}\right] \tag{166}$$

$$= \frac{2\eta}{n}\text{tr}\left[\hat{\mathbf{B}}^{\top}\left(\hat{\mathbf{B}}\hat{\mathbf{B}}^{\top} - \mathbf{I}_d\right)\sum_{i=1}^{n}\left(\hat{\mathbf{B}}^{*}\mathbf{w}_i^{*}\right)\mathbf{F}_i^{\top}\mathbf{pp}^{\top}\right] + \frac{2\eta}{n}\text{tr}\left[\hat{\mathbf{B}}^{\top}\hat{\mathbf{B}}\sum_{i=1}^{n}\mathbf{F}_i\mathbf{F}_i^{\top}\mathbf{pp}^{\top}\right] + \frac{2\eta}{n}\text{tr}\left[\hat{\mathbf{B}}^{\top}\hat{\mathbf{B}}\sum_{i=1}^{n}\mathbf{G}_i\mathbf{F}_i^{\top}\mathbf{pp}^{\top}\right] \tag{167}$$

$$= \frac{2\eta}{n}\text{tr}\left[\hat{\mathbf{B}}^{\top}\hat{\mathbf{B}}\left(\mathbf{F}^{\top}\mathbf{F} + \mathbf{G}^{\top}\mathbf{F}\right)\mathbf{pp}^{\top}\right] \tag{168}$$

$$\leq \frac{2\eta}{n}\left(\|\mathbf{F}\|_F^2 + \|\mathbf{G}\|_F\|\mathbf{F}\|_F\right) \tag{169}$$

where in the third equality we used that $\hat{\mathbf{B}}^{\top}\left(\hat{\mathbf{B}}\hat{\mathbf{B}}^{\top} - \mathbf{I}_d\right) = 0$ and in the forth that the norms of the orthonormal matrices is 1 as well as the norm of $\mathbf{pp}^{\top}$. Following the same calculations for $T_3$ we get

$$T_3 \leq \frac{2\eta}{n}\left(\|\mathbf{G}\|_F^2 + \|\mathbf{G}\|_F\|\mathbf{F}\|_F\right) \tag{170}$$

and thus summing the two terms we get the following

$$T_2 + T_3 \leq \frac{2\eta}{n}\left(\|\mathbf{F}\|_F + \|\mathbf{G}\|_F\right)^2 \tag{171}$$

Again conditioning on $\mathcal{E}_3$ (118) we derive

$$T_2 + T_3 \leq \frac{2\eta}{n}\left(\hat{\delta}\left(\|\mathbf{W}^{*}\|_2 + \sqrt{n}\sigma^2\right)\right)^2 \tag{172}$$

$$\leq 2\eta\hat{\delta}^2\left(\bar{\sigma}_{\max,*}^2 + \sigma^4\right) \tag{173}$$

$$\leq \frac{1}{10}\eta E_0\bar{\sigma}_{\min,*}^2 \tag{174}$$

Hence combining (153), (165) and (174) we get a bound for $R_1$

$$R_1 \leq 3\eta c\left(k + \sqrt{k}\right)\frac{\sqrt{d}}{\sqrt{mn}} + \frac{1}{5}\eta E_0\bar{\sigma}_{\min,*}^2 \tag{175}$$

$$\leq 6\eta \cdot c \cdot k\frac{\sqrt{d}}{\sqrt{mn}} + \frac{1}{5}\eta E_0\bar{\sigma}_{\min,*}^2 \tag{176}$$

We work in similar fashion to derive the bound on $R_2$

$$R_2 = \frac{\eta^2}{n^2}\lambda_{\max}\left(\mathbf{E}^{\top}\mathbf{S} + \mathbf{S}^{\top}\mathbf{E}\right) \tag{177}$$

$$= \frac{2\eta^2}{n^2}\max_{\mathbf{p}:\|\mathbf{p}\|_2=1}\mathbf{p}^{\top}\mathbf{S}^{\top}\mathbf{Ep} \tag{178}$$

$$\leq \frac{2\eta^2}{n^2}\max_{\mathbf{p}:\|\mathbf{p}\|_2=1}\mathbf{p}^{\top}\left[\sum_{i=1}^{n}\left(\frac{1}{m}\mathbf{X}_i^{\top}\mathbf{X}_i\left(\hat{\mathbf{B}}\mathbf{w}_i^{+} - \hat{\mathbf{B}}^{*}\mathbf{w}_i^{*}\right) - \left(\hat{\mathbf{B}}\mathbf{w}_i^{+} - \hat{\mathbf{B}}^{*}\mathbf{w}_i^{*}\right)\right)\mathbf{w}_i^{+\top}\right]\sum_{i=1}^{n}\left(\frac{1}{m}\mathbf{X}_i^{\top}\mathbf{Z}_i\mathbf{w}_i^{+\top}\right)\mathbf{p}$$

$$+ \frac{2\eta^2}{n^2}\max_{\mathbf{p}:\|\mathbf{p}\|_2=1}\mathbf{p}^{\top}\left[\sum_{i=1}^{n}\left(\hat{\mathbf{B}}\mathbf{w}_i^{+} - \hat{\mathbf{B}}^{*}\mathbf{w}_i^{*}\right)\mathbf{w}_i^{+\top}\right]\sum_{i=1}^{n}\left(\frac{1}{m}\mathbf{X}_i^{\top}\mathbf{Z}_i\mathbf{w}_i^{+\top}\right)\mathbf{p} \tag{179}$$

$$\leq 2\eta^2\left\|\frac{1}{n}\sum_{i=1}^{n}\left(\frac{1}{m}\mathbf{X}_i^{\top}\mathbf{X}_i\left(\hat{\mathbf{B}}\mathbf{w}_i^{+} - \hat{\mathbf{B}}^{*}\mathbf{w}_i^{*}\right) - \left(\hat{\mathbf{B}}\mathbf{w}_i^{+} - \hat{\mathbf{B}}^{*}\mathbf{w}_i^{*}\right)\right)\mathbf{w}_i^{+\top}\right\|_2 \cdot \left\|\frac{1}{mn}\sum_{i=1}^{n}\left(\mathbf{X}_i^{\top}\mathbf{Z}_i\mathbf{w}_i^{+\top}\right)\right\|_2$$

$$+ \frac{2\eta^2}{n}\left\|\sum_{i=1}^{n}\left(\hat{\mathbf{B}}\mathbf{w}_i^{+} - \hat{\mathbf{B}}^{*}\mathbf{w}_i^{*}\right)\mathbf{w}_i^{+\top}\right\|_2 \cdot \left\|\frac{1}{mn}\sum_{i=1}^{n}\left(\mathbf{X}_i^{\top}\mathbf{Z}_i\mathbf{w}_i^{+\top}\right)\right\|_2 \tag{180}$$

$$\tag{181}$$

Since we work conditioning on the event $\mathcal{E}_4 \bigcap \mathcal{E}_5$ we further derive

$$R_2 \leq 2\eta^2 \left( c \frac{\sqrt{d} \left( \text{dist}\left(\hat{\mathbf{B}}, \hat{\mathbf{B}}^*\right) k + \sqrt{k}\hat{\delta}\sigma^2 + \left(\hat{\delta}\sigma^2\right)^2 \right)}{\sqrt{mn}} \right) \cdot \left( c \frac{\sigma^2 \left(\sqrt{k} + \hat{\delta}\sigma^2\right)\sqrt{d}}{\sqrt{mn}} \right)$$

$$+ \frac{2\eta^2}{n} \left\| \sum_{i=1}^n \left( \hat{\mathbf{B}}\mathbf{w}_i^+ - \hat{\mathbf{B}}^*\mathbf{w}_i^* \right) \mathbf{w}_i^{+\top} \right\|_2 \cdot \left( c \frac{\sigma^2 \left(\sqrt{k} + \hat{\delta}\sigma^2\right)\sqrt{d}}{\sqrt{mn}} \right) \tag{182}$$

$$\leq 3\eta^2 \left( ck\frac{\sqrt{d}}{\sqrt{mn}} \right) \left( c\sqrt{k}\sigma^2 \frac{\sqrt{d}}{\sqrt{mn}} \right) + \frac{2\eta^2}{n} \sum_{i=1}^n \|\mathbf{q}_i\|_2 \|\mathbf{w}_i^+\|_2 \left( c\sqrt{k}\sigma^2 \frac{\sqrt{d}}{\sqrt{mn}} \right) \tag{183}$$

And since we also condition on $\mathcal{E}_2$ (117) we finally get

$$R_2 \leq 3\eta^2 c^2 k^{\frac{3}{2}} \sigma^2 \frac{d}{mn} + \frac{2\eta^2}{n} \sum_{i=1}^n \left( 2\sqrt{k} + \hat{\delta}\sigma^2 \right)^2 \left( c\sqrt{k}\sigma^2 \frac{\sqrt{d}}{\sqrt{mn}} \right) \tag{184}$$

$$\leq 3\eta^2 c^2 k^{\frac{3}{2}} \sigma^2 \frac{d}{mn} + 9\eta^2 \left( ck^{\frac{3}{2}}\sigma^2 \frac{\sqrt{d}}{\sqrt{mn}} \right) \tag{185}$$

The last term we need to bound is $R_3$

$$R_3 = \frac{\eta}{n} \lambda_{\max} \left( \mathbf{E}^\top \hat{\mathbf{B}} + \hat{\mathbf{B}}^\top \mathbf{E} \right) \tag{186}$$

$$= \frac{2\eta^2}{n^2} \max_{\mathbf{p}:\|\mathbf{p}\|_2 = 1} \mathbf{p}^\top \hat{\mathbf{B}}^\top \mathbf{E} \mathbf{p} \tag{187}$$

$$\leq \frac{2\eta}{n} \left\| \hat{\mathbf{B}} \right\|_2 \left\| \sum_{i=1}^n \frac{1}{m} \mathbf{X}_i^\top \mathbf{Z}_i \mathbf{w}_i^{+\top} \right\|_2 \tag{188}$$

$$\leq 2\eta c \cdot \frac{\sigma^2 \left(\sqrt{k} + \hat{\delta}\sigma^2\right)\sqrt{d}}{\sqrt{mn}} \tag{189}$$

$$\leq 3\eta c \sqrt{k}\sigma^2 \frac{\sqrt{d}}{\sqrt{mn}} \tag{190}$$

Combining (149) with (176), (185) and (190) we derive

$$\sigma_{\min}^2 \left( \mathbf{R}^+ \right) \geq 1 - 6\eta \cdot c \cdot k \frac{\sqrt{d}}{\sqrt{mn}} - \frac{1}{5}\eta E_0 \bar{\sigma}_{\min,*}^2 - 3\eta^2 c^2 k^{\frac{3}{2}} \sigma^2 \frac{d}{mn} - 9\eta^2 \left( ck^{\frac{3}{2}}\sigma^2 \frac{\sqrt{d}}{\sqrt{mn}} \right) - 3\eta c\sqrt{k}\sigma^2 \frac{\sqrt{d}}{\sqrt{mn}} \tag{191}$$

$$\geq 1 - 14\eta ck^{\frac{3}{2}}\sigma^2 \frac{\sqrt{d}}{\sqrt{mn}} - 3\eta^2 c^2 k^{\frac{3}{2}} \sigma^2 \frac{d}{mn} - \frac{1}{5}\eta E_0 \bar{\sigma}_{\min,*}^2 \tag{192}$$

$$\geq 1 - 15\eta ck^{\frac{3}{2}}\sigma^2 \frac{\sqrt{d}}{\sqrt{mn}} - \frac{1}{5}\eta E_0 \bar{\sigma}_{\min,*}^2 \tag{193}$$

where the last inequality holds since $\sqrt{mn} \geq c\sqrt{d}$. We can now combine (139) and (193) to obtain the contraction inequality

$$\text{dist}\left(\hat{\mathbf{B}}^+, \hat{\mathbf{B}}^*\right) \leq \text{dist}\left(\hat{\mathbf{B}}, \hat{\mathbf{B}}^*\right) \left( 1 - \frac{5}{6}\eta E_0 \bar{\sigma}_{\min,*}^2 + \eta ck \frac{\sqrt{d}}{\sqrt{mn}} \right) \cdot \left\| \left( 1 - 15\eta ck^{\frac{3}{2}}\sigma^2 \frac{\sqrt{d}}{\sqrt{mn}} - \frac{1}{5}\eta E_0 \bar{\sigma}_{\min,*}^2 \right)^{-1/2} \right\|_2$$

$$+ \eta c \left( \sqrt{k} + 1 \right) \left( \sigma^2 + 1 \right) \frac{\sqrt{d}}{\sqrt{mn}} \cdot \left\| \left( 1 - 15\eta ck^{\frac{3}{2}}\sigma^2 \frac{\sqrt{d}}{\sqrt{mn}} - \frac{1}{5}\eta E_0 \bar{\sigma}_{\min,*}^2 \right)^{-1/2} \right\|_2 \tag{194}$$

We divide and multiply by $n_0$ and using our bounds on $m$ and $n$ the previous inequality further simplifies,

$$
\text{dist}\left(\hat{\mathbf{B}}^+, \hat{\mathbf{B}}^*\right) \leq \text{dist}\left(\hat{\mathbf{B}}, \hat{\mathbf{B}}^*\right)\left(1 - \frac{5}{6}\eta E_0 \bar{\sigma}_{\min,*}^2 + \eta ck \frac{\sqrt{d}}{\sqrt{mn_0 \cdot \frac{n}{n_0}}}\right)\left(1 - 15\eta ck^{\frac{3}{2}}\sigma^2 \frac{\sqrt{d}}{\sqrt{mn_0 \cdot \frac{n}{n_0}}} - \frac{1}{5}\eta E_0 \bar{\sigma}_{\min,*}^2\right)^{-1/2}
$$
$$
+ \eta c\left(\sqrt{k}+1\right)\left(\sigma^2+1\right)\frac{\sqrt{d}}{\sqrt{mn_0 \cdot \frac{n}{n_0}}} \cdot \left(1 - 15\eta ck^{\frac{3}{2}}\sigma^2 \frac{\sqrt{d}}{\sqrt{mn_0 \cdot \frac{n}{n_0}}} - \frac{1}{5}\eta E_0 \bar{\sigma}_{\min,*}^2\right)^{-1/2}
\tag{195}
$$

$$
\leq \text{dist}\left(\hat{\mathbf{B}}, \hat{\mathbf{B}}^*\right)\left(1 - \frac{1}{2}\eta E_0 \bar{\sigma}_{\min,*}^2\right)\left(1 - \frac{1}{2}\eta E_0 \bar{\sigma}_{\min,*}^2\right)^{-1/2} + \left(\sqrt{\frac{n_0}{4n}}\eta E_0 \bar{\sigma}_{\min,*}^2\right)\left(1 - \frac{1}{2}\eta E_0 \bar{\sigma}_{\min,*}^2\right)^{-1/2}
\tag{196}
$$

$$
\leq \text{dist}\left(\hat{\mathbf{B}}, \hat{\mathbf{B}}^*\right)\sqrt{\left(1 - \frac{1}{2}\eta E_0 \bar{\sigma}_{\min,*}^2\right)} + \frac{\left(\frac{1}{2}\eta E_0 \bar{\sigma}_{\min,*}^2\right)}{\sqrt{\frac{n}{n_0}\left(1 - \frac{1}{2}\eta E_0 \bar{\sigma}_{\min,*}^2\right)}}
\tag{197}
$$

where in the second inequality we used that for our choices of $m$ and $n$ the following inequality holds $15\eta ck^{\frac{3}{2}}(1+\sigma^2)\frac{\sqrt{d}}{\sqrt{mn_0}} \leq \frac{1}{10}\eta E_0 \bar{\sigma}_{\min,*}^2$. Taking Union Bound over the total number of iterations $T$ we derive the result. $\qquad\square$

**Corollary A.11.** *Recall that our algorithm starts at stage $0$ with $n_0$ participating clients and doubles the number of participating clients at every subsequent stage. Thus, by slightly abusing notation, we can reformulate the contraction inequality of Theorem 4.7 at stage $r$ as follows*

$$
dist^+ \leq dist\sqrt{1-a} + \frac{a}{\sqrt{2^r(1-a)}} \quad with \quad a \leq \frac{1}{4}
\tag{198}
$$

# B Appendix

In the second part of our analysis we compute the expected 'Wall Clock Time' of our proposed method and compare it to the corresponding 'Wall Clock Time' of straggler-prone `FedRep`. We prove that when the computational speeds are drawn from the exponential distribution with parameter $\lambda$ and the communication cost is given by $\mathcal{C} = c\frac{1}{\lambda}$, (for some constant $c$), then `SRPFL` guarantees a logarithmic speedup. Recall that in Corollary A.11 we get the following simplified version of the contraction inequality

$$
\text{dist}^+ \leq \text{dist}\sqrt{1-a} + \frac{a}{\sqrt{2^r(1-a)}} \quad with \quad a \leq \frac{1}{4}.
\tag{199}
$$

For the rest of this section w.l.g. we assume that the clients are re-indexed at every stage so that the expected computation times maintain a decreasing ordering i.e. $\forall r \quad \mathbb{E}\left[\mathcal{T}_1^r\right] \leq \mathbb{E}\left[\mathcal{T}_2^r\right] \leq ..., \leq \mathbb{E}\left[\mathcal{T}_N^r\right]$. For simplicity henceforth we drop the stage index $r$. Notice that the decreasing ordering of the computation times in combination with (199) imply that `SRPFL` initially benefits by including only few fast nodes in the training procedure. However, as the distance diminishes the improvement guaranteed by the contraction inequality becomes negligible and thus our method benefits by including slower nodes, thus decreasing the second term of the r.h.s. of (199).

Let us denote by $X_i$ the maximum distance for which the optimal number of participating nodes (for `SRPFL`) is $n_0 \cdot 2^i$. This definition immediately implies that $X_0 = +\infty$. To compute each $X_i$ we turn our attention on measuring the progress per unit of time achieved by `SRPFL`, when $2^r \cdot n_0$ nodes are utilized. This ratio at stage $r$ can be expressed as

$$
\frac{\text{dist}^+ - \text{dist}\sqrt{1-a} - \frac{a}{\sqrt{2^r(1-a)}}}{\mathbb{E}\left[\mathcal{T}_{n_0 2^r}\right] + \mathcal{C}}.
\tag{200}
$$

Notice that by (199) the nominator captures the progress per round while the algorithm incurs $\mathbb{E}\left[\mathcal{T}_{n_0 2^r}\right]$ computation and $\mathcal{C}$ communication cost. Similarly the ration when $2^{r+1} \cdot n_0$ nodes are used is given by

$$\frac{\text{dist}^+ - \text{dist}\sqrt{1-a} - \frac{a}{\sqrt{2^{r+1}(1-a)}}}{\mathbb{E}\left[\mathcal{T}_{n_0 2^{r+1}}\right] + \mathcal{C}}. \tag{201}$$

Based on the above inequalities we can now compute the optimal doubling points (in terms of distance) and thus the values of $X_i$'s. Subsequently, we compute the number of iterations SRPFL spends in every stage.

**Lemma B.1.** *For all $i$ let $X_i$ denote the maximum distance for which the optimal number of nodes for SRPFL is $n_0 \cdot 2^i$. Then the following holds*

$$\forall i > 0 \quad X_i = \frac{a}{\sqrt{2^r(1-a)}1 - \sqrt{1-a})}\left(1 + \frac{\left(\mathbb{E}\left[\mathcal{T}_{n_0 2^r}\right] + \mathcal{C}\right)\left(1 - \frac{1}{\sqrt{2}}\right)}{\mathbb{E}\left[\mathcal{T}_{n_0 2^{r+1}}\right] - \mathbb{E}\left[\mathcal{T}_{n_0 2^r}\right]}\right) \tag{202}$$

$$X_0 = +\infty$$

*Further SRPFL spends at each stage $r$ at most $t^r$ communication rounds such that*

$$t^r \geq \frac{2\log\left(\frac{\sqrt{2}\left(\mathbb{E}\left[\mathcal{T}_{n_0 2^{r+1}}\right] - \mathbb{E}\left[\mathcal{T}_{n_0 2^r}\right]\right)}{\mathbb{E}\left[\mathcal{T}_{n_0 2^r}\right] - \mathbb{E}\left[\mathcal{T}_{n_0 2^{r-1}}\right]}\right)}{\log\left(\frac{1}{1-a}\right)}. \tag{203}$$

*Proof.* For each stage $r$ let us compute the point where the transitioning between $2^r \cdot n_0$ and $2^{r+1} \cdot n_0$ occurs. That is the distance at which SRPFL benefits by doubling the number of participation nodes to $2^{r+1}$. Thus equating the two ratios in (200) and (201) we get

$$\frac{X_{r+1} - X_{r+1}\sqrt{1-a} - \frac{a}{\sqrt{2^r(1-a)}}}{\mathbb{E}\left[\mathcal{T}_{n_0 2^r}\right] + \mathcal{C}} = \frac{X_{r+1} - X_{r+1}\sqrt{1-a} - \frac{a}{\sqrt{2^{r+1}(1-a)}}}{\mathbb{E}\left[\mathcal{T}_{n_0 2^{r+1}}\right] + \mathcal{C}} \tag{204}$$

$$X_{r+1}\left(\mathbb{E}\left[\mathcal{T}_{n_0 2^{r+1}}\right] - \mathbb{E}\left[\mathcal{T}_{n_0 2^r}\right]\right) = \frac{a}{\sqrt{2^r(1-a)}(1-\sqrt{1-a})}\left(\mathbb{E}\left[\mathcal{T}_{n_0 2^{r+1}}\right] - \frac{1}{\sqrt{2}}\mathbb{E}\left[\mathcal{T}_{n_0 2^r}\right] + \mathcal{C}\left(1 - \frac{1}{\sqrt{2}}\right)\right) \tag{205}$$

$$X_{r+1} = \frac{a}{\sqrt{2^r(1-a)}(1-\sqrt{1-a})}\left(1 + \frac{\left(\mathbb{E}\left[\mathcal{T}_{n_0 2^r}\right] + \mathcal{C}\right)\left(1 - \frac{1}{\sqrt{2}}\right)}{\mathbb{E}\left[\mathcal{T}_{n_0 2^{r+1}}\right] - \mathbb{E}\left[\mathcal{T}_{n_0 2^r}\right]}\right) \tag{206}$$

Let us now compute the number of rounds $t^r$ (henceforth denoted by $t$) required in stage $r$. That is the minimum number of iterations that SRPFL needs to decrease the distance from $X_r$ to $X_{r+1}$ using only the $n_0 2^r$ fastest participating nodes. Thus, starting off at $X_r$ and following (199) for $t$ rounds we have

$$X_r^t \leq X_r(\sqrt{1-a})^t + \sum_{i=0}^{t-1}\frac{a}{\sqrt{2^r(1-a)}}(\sqrt{1-a})^i \tag{207}$$

As stated above we want to find the minimum number of rounds such that we reach the next doubling point i.e. we want $t$ large enough such that

$$X_{r+1} \geq X_r(\sqrt{1-a})^t + \sum_{i=0}^{t-1}\frac{a}{\sqrt{2^r(1-a)}}(\sqrt{1-a})^i \tag{208}$$

$$\geq X_r(\sqrt{1-a})^t + \frac{a}{\sqrt{2^r(1-a)}}\cdot\frac{1-\sqrt{1-a}^t}{1-\sqrt{1-a}} \tag{209}$$

where in the last inequality we use geometric series properties. We proceed to solve for $t$ by rearranging and using (202) and the fact that $X_r > \frac{a}{\sqrt{2^r(1-a)}(1-\sqrt{1-a})}$,

$$(\sqrt{1-a})^t \leq \frac{\sqrt{2^r(1-a)}(1-\sqrt{1-a})X_{r+1}-a}{\sqrt{2^r(1-a)}(1-\sqrt{1-a})X_r-a} \tag{210}$$

$$\leq \frac{\frac{\left(\mathbb{E}[\mathcal{T}_{n_0 2^r}]+\mathcal{C}\right)\left(1-\frac{1}{\sqrt{2}}\right)}{\mathbb{E}[\mathcal{T}_{n_0 2^{r+1}}]-\mathbb{E}[\mathcal{T}_{n_0 2^r}]}}{\sqrt{2}\left(\frac{\left(\mathbb{E}\left[\mathcal{T}_{n_0 2^{r-1}}\right]+\mathcal{C}\right)\left(1-\frac{1}{\sqrt{2}}\right)}{\mathbb{E}[\mathcal{T}_{n_0 2^r}]-\mathbb{E}\left[\mathcal{T}_{n_0 2^{r-1}}\right]}+1-\frac{1}{\sqrt{2}}\right)} \tag{211}$$

$$\leq \frac{\mathbb{E}\left[\mathcal{T}_{n_0 2^r}\right]-\mathbb{E}\left[\mathcal{T}_{n_0 2^{r-1}}\right]}{\mathbb{E}\left[\mathcal{T}_{n_0 2^{r+1}}\right]-\mathbb{E}\left[\mathcal{T}_{n_0 2^r}\right]} \tag{212}$$

$$\tag{213}$$

taking the logarithm on both sides we derive the required amount of rounds

$$t \geq \frac{2\log\left(\frac{\sqrt{2}\left(\mathbb{E}[\mathcal{T}_{n_0 2^{r+1}}]-\mathbb{E}[\mathcal{T}_{n_0 2^r}]\right)}{\mathbb{E}[\mathcal{T}_{n_0 2^r}]-\mathbb{E}\left[\mathcal{T}_{n_0 2^{r-1}}\right]}\right)}{\log(\frac{1}{1-a})} \tag{214}$$

$$\tag{215}$$

$$\square$$

The following lemmas compute the 'Wall Clock Time' that `SRPFL` and `FedRep` require in order to achieve target accuracy $\epsilon$. As discussed in Section 4.1 for fair comparison we consider accuracy of the form

$$\epsilon = \hat{c}\frac{\alpha}{\sqrt{\frac{N}{n_0}(1-\alpha)}\left(1-\sqrt{1-\alpha}\right)}, \qquad with \qquad \sqrt{2} > \hat{c} > 1. \tag{216}$$

When $\hat{c}$ takes values close to $\sqrt{2}$ we expect `SRPFL` to vastly outperform `FedRep` and as $\hat{c}$ takes values close to 1 the performance gap diminishes.

**Lemma B.2.** *Suppose at each stage the client's computational times are i.i.d. random variables drawn from the exponential distribution with parameter $\lambda$. Further, suppose that the expected communication cost per round is $\mathcal{C} = c\frac{1}{\lambda}$, for some constant c. Finally, consider target accuracy $\epsilon$ given in (12). Then the expected 'Wall Clock Time' for `SRPFL` is upper bounded as follows*

$$\mathbb{E}\left[T_{SRPFL}\right] \leq \log N \left(\frac{6(c+1)+4\log(\frac{1}{\hat{c}-1})}{\log(\frac{1}{1-a})}\right)\frac{1}{\lambda} \tag{217}$$

*Proof.* First we upper bound the expected cost suffered by our method until the distance between the current representation and the optimal representation becomes smaller than $X_{\log(N/n_0)}$, i.e. the cost corresponding to the first $\log\left(\frac{N}{2n_0}\right)$ stages of `SRPFL` (denoted by $\mathbb{E}\left[T_{SRPFL}\right]$).

$$\mathbb{E}\left[T_{SRPFL}^1\right] = \sum_{i=1}^{\log(\frac{N}{2n_0})} t^i \left(\mathbb{E}\left[\mathcal{T}_{n_0 2^i}\right] + \mathcal{C}\right) \tag{218}$$

$$\leq \sum_{i=1}^{\log(\frac{N}{2n_0})} 2\left(\mathbb{E}\left[\mathcal{T}_{n_0 2^i}\right] + \mathcal{C}\right) \cdot \frac{\log\left(\frac{\sqrt{2}\left(\mathbb{E}\left[\mathcal{T}_{n_0 2^{i+1}}\right] - \mathbb{E}\left[\mathcal{T}_{n_0 2^i}\right]\right)}{\mathbb{E}\left[\mathcal{T}_{n_0 2^i}\right] - \mathbb{E}\left[\mathcal{T}_{n_0 2^{i-1}}\right]}\right)}{\log(\frac{1}{1-a})} \tag{219}$$

$$\leq \sum_{i=1}^{\log(\frac{N}{4n_0})} 2\left(\mathbb{E}\left[\mathcal{T}_{n_0 \cdot 2^i}\right] + \mathcal{C}\right) \frac{\log\left(\frac{\sqrt{2}\left(\mathbb{E}\left[\mathcal{T}_{\frac{N}{2}}\right] - \mathbb{E}\left[\mathcal{T}_{\frac{N}{4}}\right]\right)}{\mathbb{E}\left[\mathcal{T}_{\frac{N}{4}}\right] - \mathbb{E}\left[\mathcal{T}_{\frac{N}{8}}\right]}\right)}{\log(\frac{1}{1-a})}$$

$$+ 2(\mathbb{E}\left[\mathcal{T}_{N/2}\right] + \mathcal{C}) \frac{\log\left(\frac{\sqrt{2}\left(\mathbb{E}[\mathcal{T}_N] - \mathbb{E}\left[\mathcal{T}_{\frac{N}{2}}\right]\right)}{\mathbb{E}\left[\mathcal{T}_{\frac{N}{2}}\right] - \mathbb{E}\left[\mathcal{T}_{\frac{N}{4}}\right]}\right)}{\log(\frac{1}{1-a})}, \tag{220}$$

where we used 214. Since the computational times of the clients come from the exponential distribution it is straightforward to derive the following bounds

$$\mathbb{E}\left[\mathcal{T}_N\right] - \mathbb{E}\left[\mathcal{T}_{N/2}\right] = \frac{1}{\lambda} \sum_{i=1}^{N/2} \frac{1}{i} \leq \frac{1}{\lambda}\left(\ln(N/2) + 1\right) \leq \frac{1}{\lambda}\log(N) \tag{221}$$

$$\mathbb{E}\left[\mathcal{T}_{N/2}\right] - \mathbb{E}\left[\mathcal{T}_{N/4}\right] = \frac{1}{\lambda}\left(\frac{1}{N/2+1} + \frac{1}{N/2+2} + ... + \frac{1}{3N/4}\right) \geq \frac{1}{\lambda} \cdot \frac{N}{4} \cdot \frac{4}{3N} = \frac{1}{3\lambda} \tag{222}$$

$$\mathbb{E}\left[\mathcal{T}_{N/2}\right] - \mathbb{E}\left[\mathcal{T}_{N/4}\right] \leq \frac{1}{\lambda} \cdot \frac{N}{4} \cdot \frac{2}{N} = \frac{1}{2\lambda} \tag{223}$$

$$\mathbb{E}\left[\mathcal{T}_{N/4}\right] - \mathbb{E}\left[\mathcal{T}_{N/8}\right] = \frac{1}{\lambda}\left(\frac{1}{3N/4+1} + \frac{1}{3N/4+2} + ... + \frac{1}{7N/8}\right) \geq \frac{1}{\lambda} \cdot \frac{N}{8} \cdot \frac{4}{3N} = \frac{1}{6\lambda} \tag{224}$$

Making use of the above bounds the expression in 220 further simplifies to

$$\leq 2 \sum_{i=1}^{\log\left(\frac{N}{4n_0}\right)} \left[\left(\mathbb{E}\left[\mathcal{T}_{n_0 \cdot 2^i}\right] + \mathcal{C}\right) \frac{\log(3\sqrt{2})}{\log(\frac{1}{1-a})}\right] + 2\left(\mathbb{E}\left[\mathcal{T}_{N/2}\right] + \mathcal{C}\right) \frac{\log(3\sqrt{2}\log(N))}{\log(\frac{1}{1-a})} \tag{225}$$

$$\leq \frac{5}{\log(\frac{1}{1-a})} \left(\sum_{i=1}^{\log\left(\frac{N}{4n_0}\right)} \mathbb{E}\left[\mathcal{T}_{n_0 \cdot 2^i}\right] + \log\left(\frac{N}{2n_0}\right) \cdot \mathcal{C}\right) + 2\frac{\log(3\sqrt{2}\log(N))}{\log(\frac{1}{1-a})}\left(\mathbb{E}\left[\mathcal{T}_{N/2}\right] + \mathcal{C}\right) \tag{226}$$

Further, notice that

$$\mathbb{E}\left[\mathcal{T}_{N/2}\right] = \frac{1}{\lambda} \sum_{i=1}^{N/2} \frac{1}{N/2+i} \leq \frac{1}{\lambda}, \tag{227}$$

and similarly $\mathbb{E}\left[\mathcal{T}_{N/4}\right] \leq \frac{1}{\lambda} \cdot \frac{1}{3}$, $\mathbb{E}\left[\mathcal{T}_{N/8}\right] \leq \frac{1}{\lambda} \cdot \frac{1}{7}$, $\mathbb{E}\left[\mathcal{T}_{N/16}\right] \leq \frac{1}{\lambda} \cdot \frac{1}{15}$ and so on. Thus,

$$\sum_{i=1}^{\log\left(\frac{N}{4n_0}\right)} \mathbb{E}\left[\mathcal{T}_{n_0 \cdot 2^i}\right] = \mathbb{E}\left[2n_0\right] + \mathbb{E}\left[4n_0\right] + ... + \mathbb{E}\left[N/4\right] \leq \frac{1}{\lambda} \sum_{i=1}^{\infty} \frac{1}{2^i} \leq \frac{1}{\lambda} \tag{228}$$

Combining the bounds from 227 and 228 and substituting $\mathcal{C} = c\frac{1}{\lambda}$ in expression 226 we derive the following bound

$$\mathbb{E}\left[T_{SRPFL}^1\right] \leq \frac{5}{\log\left(\frac{1}{1-a}\right)} \left(c\log(N/2n_0) + 1\right)\frac{1}{\lambda} + \frac{2\log(3\sqrt{2}\log(N))}{\log\left(\frac{1}{1-a}\right)(c+1)}\frac{1}{\lambda} \tag{229}$$

$$\leq \log(N/n_0) \frac{6(c+1)}{\log\left(\frac{1}{1-a}\right)} \cdot \frac{1}{\lambda} \tag{230}$$

Having derived an upper bound on the cost suffered by `SRPFL` on the first $\log\left(\frac{N}{2n_0}\right)$ stages we now turn our attention on bounding the cost incurred from $X_{\log(\frac{N}{n_0})}$ until the target accuracy $\epsilon$ is achieved (denoted by $\mathbb{E}\left[T_{SRPFL}^2\right]$). Recall that

$$X_{\log(N/n_0)} = \frac{a}{\sqrt{\frac{N}{2n_0}(1-a)}(1-\sqrt{1-a})}\left(1 + \frac{(\mathbb{E}\left[\mathcal{T}_{N/2}\right]+\mathcal{C})(1-\frac{1}{\sqrt{2}})}{\mathbb{E}\left[\mathcal{T}_N\right]-\mathbb{E}\left[\mathcal{T}_{N/2}\right]}\right) \tag{231}$$

and further during the last stage of `SRPFL`, $N$ clients are utilized deriving the following form in the contractions inequality from (199)

$$\text{dist}^+ \le \text{dist}\sqrt{1-a} + \frac{a}{\sqrt{\frac{N}{n_0}(1-a)}} \quad with \quad a \le \frac{1}{4}. \tag{232}$$

We first compute the number of rounds required in this second phase of the algorithm. Starting with distance $X_{\log(N/n_0)}$ and following the contraction in (232) for $t$ rounds, we derive current distance at most

$$X_{\log\left(\frac{N}{n_0}\right)} \cdot (\sqrt{1-a})^t + \sum_{i=0}^{t-1} \frac{a}{\sqrt{\frac{N}{n_0}(1-a)}}(\sqrt{1-a})^i \tag{233}$$

$$= X_{\log\left(\frac{N}{n_0}\right)} \cdot (\sqrt{1-a})^t + \frac{a}{\sqrt{\frac{N}{n_0}(1-a)}}\frac{1-\sqrt{1-a}^t}{1-\sqrt{1-a}} \tag{234}$$

$$= \frac{a}{\sqrt{\frac{N}{n_0}(1-a)}(1-\sqrt{1-a})}\left(\sqrt{2}\left(1 + \frac{(\mathbb{E}\left[\mathcal{T}_{N/2}\right]+\mathcal{C})(1-\frac{1}{\sqrt{2}})}{\mathbb{E}\left[\mathcal{T}_N\right]-\mathbb{E}\left[\mathcal{T}_{N/2}\right]}\right)(\sqrt{1-a}^t) + (1-\sqrt{1-a}^t)\right) \tag{235}$$

where in the first equality we use geometric series properties and in the second we substitute according to (231). Using the fact that $\sqrt{2}(\mathbb{E}\left[\mathcal{T}^{N/2}\right]+\mathcal{C})(1-\frac{1}{\sqrt{2}}) \le \frac{1}{\lambda}(c+1)(\sqrt{2}-1)$ and $\mathbb{E}\left[\mathcal{T}_N\right] - \mathbb{E}\left[\mathcal{T}^{N/2}\right] \le \frac{1}{\lambda}\log N$ expression (235) is upper bounded by

$$\le \frac{a}{\sqrt{\frac{N}{n_0}(1-a)}(1-\sqrt{1-a})}\left(\left(\sqrt{2} + \frac{(c+1)(\sqrt{2}-1)}{\log N}\right)(\sqrt{1-a}^t) + (1-\sqrt{1-a}^t)\right) \tag{236}$$

$$\le \frac{a}{\sqrt{\frac{N}{n_0}(1-a)}(1-\sqrt{1-a})} + \frac{a}{\sqrt{\frac{N}{n_0}(1-a)}(1-\sqrt{1-a})} \cdot \sqrt{1-a}^t \tag{237}$$

The above implies that the number of rounds in the second phase is the smallest $t$ so that the target accuracy is achieved, i.e.,

$$\epsilon \ge \frac{a}{\sqrt{\frac{N}{n_0}(1-a)}(1-\sqrt{1-a})} + \frac{a}{\sqrt{\frac{N}{n_0}(1-a)}(1-\sqrt{1-a})} \cdot \sqrt{1-a}^t \tag{238}$$

Further recall that from (216) the accuracy can be expressed in terms of

$$\epsilon = \hat{c}\frac{\alpha}{\sqrt{\frac{N}{n_0}(1-\alpha)}\left(1-\sqrt{1-\alpha}\right)}, \qquad with \qquad \sqrt{2} > \hat{c} > 1. \tag{239}$$

Combining the above and solving for $t$ we derive the required number of rounds for the second phase

$$t \ge \frac{2\log(\frac{1}{\hat{c}-1})}{\log(\frac{1}{1-a})} \tag{240}$$

The expected cost during phase 2 can be computed as follows

$$\mathbb{E}\left[T_{SRPFL}^2\right] \leq (\mathbb{E}\left[\mathcal{T}_N\right] + \mathcal{C}) \left(\frac{2\log(\frac{1}{\hat{c}-1})}{\log(\frac{1}{1-a})} + 1\right) \tag{241}$$

$$\leq (\ln(N) + 1 + c)\frac{1}{\lambda}\left(\frac{2\log(\frac{1}{\hat{c}-1})}{\log(\frac{1}{1-a})} + 1\right) \tag{242}$$

$$\leq 4\log(N)\left(\frac{\log(\frac{1}{\hat{c}-1})}{\log(\frac{1}{1-a})}\right)\frac{1}{\lambda} \tag{243}$$

$$\leq \log(N) \cdot \frac{4}{\log(\frac{1}{1-a})}\log\left(\frac{1}{\hat{c}-1}\right)\frac{1}{\lambda} \tag{244}$$

Summing the two quantities of interest we can derive the promised upper bound on the 'Wall Clock Time' of `SRPFL`.

$$\mathbb{E}\left[T_{SRPFL}\right] = \mathbb{E}\left[T_{SRPFL}^1\right] + \mathbb{E}\left[T_{SRPFL}^2\right] \tag{245}$$

$$\leq \log(N/n_0)\frac{6(c+1)}{\log\left(\frac{1}{1-a}\right)} \cdot \frac{1}{\lambda} + \log(N) \cdot \frac{4}{\log(\frac{1}{1-a})}\log\left(\frac{1}{\hat{c}-1}\right)\frac{1}{\lambda} \tag{246}$$

$$\leq \log N \left(\frac{6(c+1) + 4\log(\frac{1}{\hat{c}-1})}{\log(\frac{1}{1-a})}\right)\frac{1}{\lambda} \tag{247}$$

$\square$

Having computed an upper bound on the expected 'Wall Clock Time' of `SRPFL` we proceed to compute an lower bound on the expected 'Wall Clock Time' of `FedRep`.

**Lemma B.3.** *Suppose at each stage the client's computational times are i.i.d. random variables drawn from the exponential distribution with parameter $\lambda$. Further, suppose that the expected communication cost per round is $\mathcal{C} = c\frac{1}{\lambda}$, for some constant $c$. Finally, consider target accuracy $\epsilon$ given in (12). Then the expected 'Wall Clock Time' for `FedRep` is lower bounded as follows*

$$\mathbb{E}\left[T_{FedRep}\right] \geq \log N \left(\frac{\log N + 2\log\left(\frac{1}{\hat{c}-1}\right)}{\log\left(\frac{1}{1-a}\right)}\right)\frac{1}{\lambda} \tag{248}$$

*Proof.* First we compute the number of rounds required by `FedRep` to achieve the target accuracy. Recall that `FedRep` utilizes $N$ clients at each round deriving the following form of the contractions inequality from (199)

$$\text{dist}^+ \leq \text{dist}\sqrt{1-a} + \frac{a}{\sqrt{\frac{N}{n_0}(1-a)}} \quad with \quad a \leq \frac{1}{4}. \tag{249}$$

Starting with distance equal 1 and following the contraction in (249) for $t$ rounds, we derive current distance at most

$$(\sqrt{1-a})^t + \sum_{i=0}^{t-1}\frac{a}{\sqrt{\frac{N}{n_0}(1-a)}}(\sqrt{1-a})^i = (\sqrt{1-a})^t + \frac{a}{\sqrt{\frac{N}{n_0}(1-a)}}\frac{1-\sqrt{1-a}^t}{1-\sqrt{1-a}}, \tag{250}$$

using the properties of geometric series. Further recall that from (216) the accuracy can be expressed as

$$\epsilon = \hat{c}\frac{\alpha}{\sqrt{\frac{N}{n_0}(1-\alpha)}\left(1-\sqrt{1-\alpha}\right)}, \qquad with \qquad \sqrt{2} > \hat{c} > 1. \tag{251}$$

The above imply that the number of rounds is going to be the smallest $t$ that guarantees that the target accuracy has been achieved that is

$$\hat{c}\frac{\alpha}{\sqrt{\frac{N}{n_0}\left(1-\alpha\right)}\left(1-\sqrt{1-\alpha}\right)} \geq (\sqrt{1-a})^t + \frac{a}{\sqrt{\frac{N}{n_0}(1-a)}}\frac{1-\sqrt{1-a}^t}{1-\sqrt{1-a}} \tag{252}$$

We use the fact that $\sqrt{\frac{N}{n_0}(1-a)}(1-\sqrt{1-a}) - a > 0$ for $a \leq 1/4$ and all reasonable values of $N$ to rearrange and solve for $t$. Thus, we derive

$$t \geq \frac{2\log\left(\frac{1}{\hat{c}-1}\right)}{\log\left(\frac{1}{1-a}\right)} + \frac{\log N}{\log\left(\frac{1}{1-a}\right)} \tag{253}$$

Multiplying the number of rounds with a lower bound on the expected cost incurred per round, results in the desired lower bound on the expected 'Wall Clock Time' suffered by `FedRep`:

$$\mathbb{E}\left[T_{FedRep}\right] \geq \left(\mathbb{E}\left[\mathcal{T}_N\right]+\mathcal{C}\right)\left(\frac{2\log\left(\frac{1}{\hat{c}-1}\right)}{\log\left(\frac{1}{1-a}\right)}\right) \tag{254}$$

$$\geq \log N\left(\frac{\log N + 2\log\left(\frac{1}{\hat{c}-1}\right)}{\log\left(\frac{1}{1-a}\right)}\right)\frac{1}{\lambda} \tag{255}$$

$$\square$$

Combining the results of Lemma B.2 and Lemma B.3 we obtain Theorem 4.9.

**Theorem 4.9.** *Suppose that at each stage the client's computational times are i.i.d. random variables drawn from the exponential distribution with parameter $\lambda$. Further, suppose that the expected communication cost per round is $\mathcal{C} = \frac{c}{\lambda}$, for some constant c. Finally, consider the target error $\epsilon$ given in (12). Then, we have*
$\frac{\mathbb{E}[T_{SRPFL}]}{\mathbb{E}[T_{FedRep}]} = \mathcal{O}\left(\frac{\log\left(\frac{1}{\hat{c}-1}\right)}{\log(N)+\log\left(\frac{1}{\hat{c}-1}\right)}\right).$

*Proof.*

$$\frac{\mathbb{E}\left[T_{SRPFL}\right]}{\mathbb{E}\left[T_{FedRep}\right]} \leq \frac{6(c+1)+4\log(\frac{1}{\hat{c}-1})}{\log N + 2\log\left(\frac{1}{\hat{c}-1}\right)} = \mathcal{O}\left(\frac{\log\left(\frac{1}{\hat{c}-1}\right)}{\log(N)+\log\left(\frac{1}{\hat{c}-1}\right)}\right) \tag{256}$$

*Remark* B.4. The initialization scheme in Algorithm 2 guarantees that $\text{dist}\left(\mathbf{B}^0, \mathbf{B}^*\right) \leq 1-c$, with probability at least $1-\mathcal{O}\left((mn)^{-100}\right)$, effectively without increasing the overall sample complexity. The formal statement and proof is identical to Theorem 3 in (Collins et al., 2021) and is omitted.

## C    More on Experiments

**Hyperparameters and choice of models.** We set the hyperparameters following the work of (Collins et al., 2021). Specifically, for the implementation of `FedRep` and `FedRep-SRPFL` we use SGD with momentum where the momentum parameter is set to 0.5 and the local learning rate to 0.1. Further, similarly to (Collins et al., 2021) we set the local learning rate to 0.1 for all other methods under consideration, which obtains optimal performance. We fix the batch size to 10 for all our implementations. The number of local epochs is set to 1 for CIFAR10 with $N = 100$ and to 5 for the rest of the datasets. In terms of the choice of the neural network model, for CIFAR10, we use LeNet-5 including two convolution layers with $(64, 64)$ channels and three fully connected layers where the numbers of hidden neurons are $(120, 64)$. The same structure is used for CIFAR100, but the numbers of channels in the convolution layers are increased to $(64, 128)$ and the numbers of hidden neurons are increased to $(256, 128)$. Additionally, a dropout layer with parameter 0.6 is added after the first two fully connected layers, which improves the testing accuracy. For EMNIST and

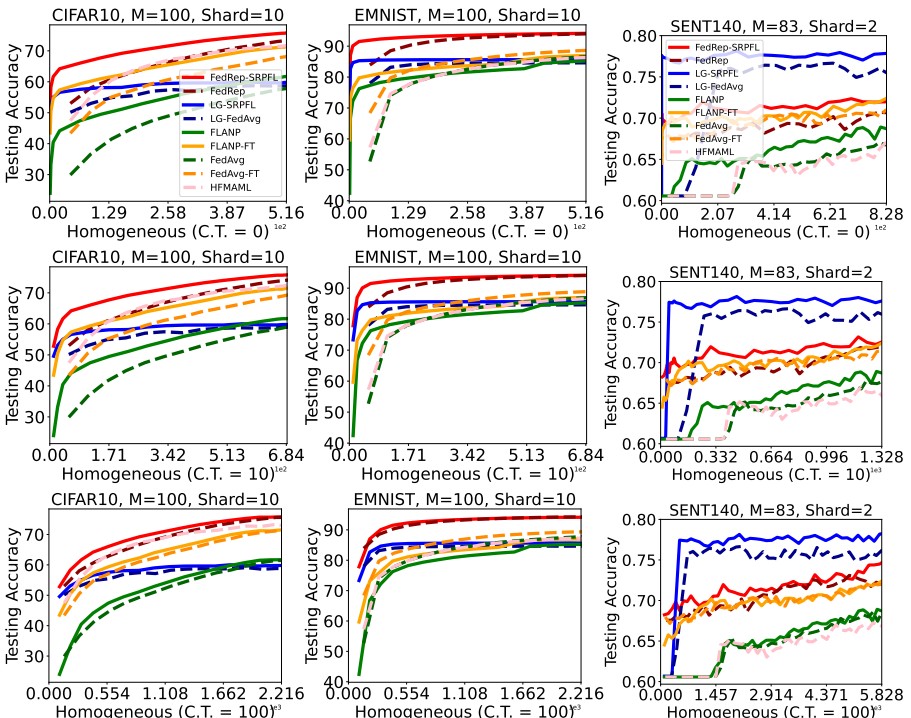

Figure 7: Numerical results on CIFAR10, EMNIST, Sent140 with full participation ($M = N$) in the fixed computation speeds setting. 'Shard' denotes the number of classes per client. 'C.T.' denotes the communication cost per round.

FEMNIST, we use MLP with three hidden layers with $(512, 256, 64)$ hidden neurons. For Sent140 we use two-layer bidirectional LSTM with dimension 256 and dropout rate 0.5 followed by a fully connected layer of dimensions 5 and a classification head. Further, we use the standard glove embedding of dimension 100 and vocabulary of size 10000.

For the needs of SRFRL, we split the neural network model into two parts, the customized head $h_i$ and the common representation $\phi$. In our experiments, we simply take the customized head to be the last hidden layer and the rest of the parameters are treated as the common representation. Note that LG-FedAvg and LG-FLANP have a different head/representation split scheme and the head is globally shared across all clients while a local version of the representation is maintained on every client. For all included datasets, i.e. CIFAR10, CIFAR100, EMNIST, FEMNIST and Sent140 the common head include the last two fully connected layers and the rest of the layers are treated as the representation part.

**Datasets.** We include five datasets in our empirical study: CIFAR10 which consists of 10 classes and a total number of 50,000 training data points, CIFAR100 which consists of 100 classes and the same amount of data points as CIFAR10, and EMNIST (balanced) which consists of 47 classes and 131,600 training data points. Note that in Figures 3-7, we use the first 10 classes from EMNIST following (Collins et al., 2021). For FEMNIST, we use the same setting as (Collins et al., 2021) with the exception that in Figures 3-6 we allocate to each client 150 data points and in Figure 6 we allocate to client $i$, $(100 + u_i)$ samples, with $u_i$ a uniformly distributed random variable in $[0, 50]$. The sentiment140 dataset contains $1,600,000$ tweets annotated negative or positive and they can be used to detect sentiment. For this dataset we perform pre-processing splitting the samples across clients both in a homogeneous as well as a heterogeneous manner. The number of allocated samples to clients follow the log-normal distribution. During our training procedure, we perform the data augmentation operations of standard random crop and horizontal flip on the first two datasets and and perform no pre-processing for the last one.

**Homogeneous Setting.** Apart from providing straggler-resilience benefits our method takes advantage of the shared representation model to simultaneously address the hurdle of data heterogeneity. In the less challenging

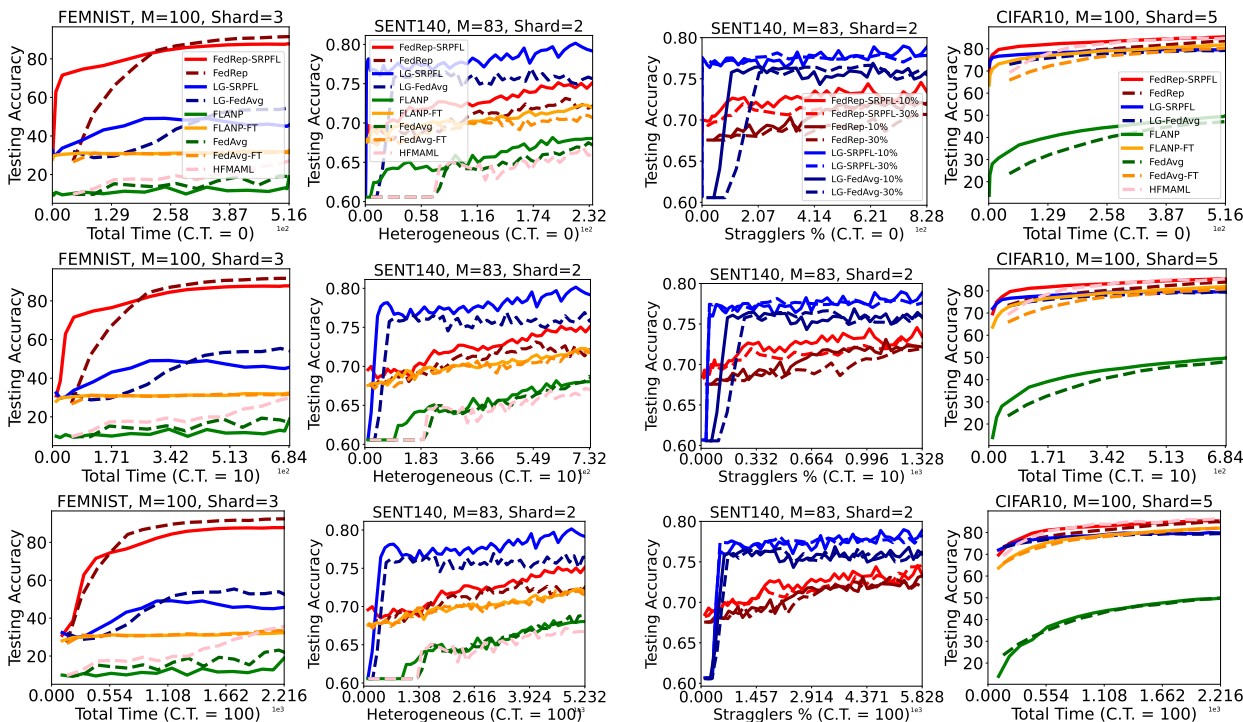

Figure 8: Numerical results on FEMNIST, Sent140, CIFAR10 in the fixed computation speeds setting. From left to right: Imbalanced/Imbalanced/Stragglers%/Correlated Heterogeneity. 'Shard' denotes the number of classes per client.'C.T.' denotes the communication cost per round.

homogeneous setting one would expect `FedRep-SRPFL` to lose some of its competitive advantage. However, our numerical results in Figure 7 indicate that our method exhibits a stable performance while maintaining an edge over the other baselines. We note that `LG-SRPFL` outperforms `FedRep-SRPFL` in the Sent140 dataset where `LG-FedAvg` appears to be better-suited than `FedRep`. Importantly, our doubling scheme continues to enhance both subroutines providing clear straggler-resilience benefits even in data homogeneous environments.

**Additional Regimes.** In Figure 8 we observe the extent at which the benefits of our meta-algorithm persist in various regimes. *Imbalanced Datasets.* In the two left columns we explore how the performance of our doubling scheme degrades for different levels of imbalanced local datasets. In the FEMNIST dataset we allocate to each client $i$, $(100 + u_i)$ local samples where $u_i$ is a random variable uniformly distributed in $[0, 50]$. In this setting the numbers of local samples are sufficiently close and as a result the benefits of our scheme are prevalent for all different values of communication cost. As we allow the clients to have more diverse numbers of local samples however, the speedup provided by our doubling scheme diminishes. Indeed this is the case in the Sent140 dataset where we allocate data to clients in a more dispropotional manner following the log-normal distribution. We point out that in settings with high diversity simply doubling the number of clients is not sufficient. Instead, to achieve optimal performance one needs to select consecutive participating sets of clients with cumulative number of samples that double from one stage to the next. *Different levels of stragglers.* We further consider the setting where clients are split into two categories. The first category consists of typically fast clients whose computational values come from the exponential distribution with $\lambda = 0.1$. The second category consists of clients who are typically straggling, i.e. sampling their computational cost from the exponential distribution with $\lambda = 10$. On the third column of Figure 8 we compare `FedRep`, `LG-FedAvg` and their straggler-resilient variants for different percentages of clients coming from the latter category. Unsurprisingly, the performance of the methods under consideration degrades as the percentage of stragglers increases however our doubling scheme to some extent mitigates this effect. *Correlated heterogeneity.* In this setting we assign data to clients depending on their speeds. More specifically,

we split clients into 3 groups based on the computational times and respectively we group samples into 3 distinct groups based on their labels. Subsequently, we allocate mutually exclusive (with respect to their labels) groups of data to specific groups of clients. The rightmost column of Figure 8 indicates that the benefits of our doubling scheme persist, although to a lesser degree as the correlation between data and system heterogeneity grows.

