# OpenReview forum: "Straggler-Resilient Personalized Federated Learning"
_TMLR — Accepted by TMLR_

### Review · Reviewer_wdpT · 2023-06-09

**Summary Of Contributions:**

The paper proposes a new framework to handle the data heterogeneity problem as well as system heterogeneity at same time. Specifically, to deal with the data heterogeneity problem, it adopt a framework that include a personalized head for every client and a shared embedding among all the clients. To deal with the system heterogeneity problem, instead of random sampling several candidates uniformly, based on the computation power, it select the top n_0 clients in every round. The theoretical analysis shows that in the linear representation case, the proposed SRPFL method could achieve O(log N) speed up compared with FedRep. The experiments done in several datasets with different system heterogeneity show the proposed method could achieve a faster convergence than other baselines.

**Audience:**

Yes

**Claims And Evidence:**

Yes

**Requested Changes:**

Typos:
dist(B^0, B*) in eq 7.

1. Move the experiments of Figure 5 or 6 in the appendix to the main paper.
2. Add experiments on the data-homogeneous settings.

**Strengths And Weaknesses:**

Pros:
1. The paper is well-written and easy to follow. The background introduction is detailed and the commons and differences of the proposed framework have been clearly illustrated.
2. The theoretical analysis shows the proposed method could achieve O(log N) improvement over FedRep although it is based on a linear representation cases.
3. The experiments on different level system heterogeneity show the potential of the proposed method to achieve a faster convergence over other baselines.

Cons:
1. The proposed setting is not that practical and affects its novelty. Although the paper mention it is possible to get the computation power of every client, I still think it is a very big assumption. The network communication could affect the overall aggregation as well so that the slowest client might not be the slowest always. Also, I think it is not trivial to achieve the rank every easily.
2. The theoretical analysis is done in the linear way that is not practical as well.
3. The experimental results should be included for the data-homogeneous settings to show it could still achieve a stable performance.

---

> ### Author Response · Authors · 2023-07-06
> **Rebuttal**
>
> $\textbf{Q}$. Typo in equation $7$.
>
> $\textbf{A}$. We thank the reviewer for their diligent study of our work and for pointing out this typo. We have updated our revisited manuscript accordingly.
>
> $\textbf{Q}$. Move the experiments of Figure $5$ or $6$ in the appendix to the main paper.
>
> $\textbf{A}$. We thank the reviewer for their comment. In our revised version of our paper we moved Figure $5$ and $6$ in the main body and updated the text in the experiments section accordingly.
>
> $\textbf{Q}$. The experimental results should be included for the data-homogeneous settings to show it could still achieve a stable performance.
>
> $\textbf{A}$.  We thank the reviewer for their comment. To address their concern we have included additional numerical results in the data homogeneous setting illustrated in Figure $7$ of our revisited paper. We note that apart from providing straggler-resilience benefits our method takes advantage of the shared representation model to simultaneously address the hurdle of data heterogeneity. In the less challenging homogeneous setting one would expects FedRep-SRPFL to lose some of its competitive advantage. However, our numerical results indicate that our method exhibits a stable performance while maintaining an edge over the other baselines. Importantly, our doubling scheme continues to enhance both subroutines providing clear straggler-resilience benefits even in data homogeneous environments.
>
> $\textbf{Q}$. Although the paper mention it is possible to get the computation power of every client, I still think it is a very big assumption.
>
> $\textbf{A}$. We note that our method does not require for the server to know the computational power of the clients. As we explain in Remark $3.1$ at every round the server utilizes the first $n$ models that it receives from the clients in order to aggregate them and produce a new representation model. Indeed, aggregating these $n$ models suffices for our method to exhibit straggler-resilient performance and enjoy theoretically proven speedup guarantees (Theorem $4.6$). We should also add that, our method does not require the $n$ fastest nodes among all $M$ nodes to share their information with the server. Instead, it requires the $n$ fastest nodes among the $N$ participating nodes (randomly selected and beyond the server's control) to send their updates to the server.

---

> ### Author Response · Authors · 2023-07-06
> **Rebuttal**
>
> $\textbf{Q}$. The network communication could affect the overall aggregation as well so that the slowest client might not be the slowest always.
>
> $\textbf{A}$. We note that there is a long line of work in the literature of federated learning that aims at reducing the communication cost per round either through quantization and compression (Seide et al., 2014; Smith et al., 2016; Alistarh et al., 2017) or sparcification (Jiang and Agrawal, 2018; Wangni et al., 2018; Basu et al., 2019). Our work is orthogonal to these schemes and can be utilized in combination with them to derive robust results. Indeed, in the extreme scenario where the communication cost vastly dominates the computation times the latter becomes irrelevant with respect to the total time of the algorithm and no client-participation scheme is able to achieve meaningful improvements. However, as illustrated in Figure $3$, even when the communication cost is $100$ times greater than the mean of the computation times $(C=100, \lambda =1)$, \texttt{SRPFL} achieves significant benefits compared to the straggler-prone variants on CIFAR100 and (slightly less) on CIFAR10. That said, combining our work with a communication efficient framework (with compression) is an interesting research direction to explore in order to magnify the effects of adaptive node participation.
>
> $\textbf{Q}$. The theoretical analysis is done in the linear way that is not practical as well.
>
> $\textbf{A}$. In the linear representation setting one aims to learn a low-dimensional subspace in which the ground-truth regressors for a collection of linear regression tasks lie. This model has been commonly used by numerous studies in the areas of meta-learning (Tripuraneni et al., 2021; Du et al., 2020; Sun et al., 2021), multitask learning (Maurer er al., 2016), model personalization (Jain et al., 2021; Shen et al., 2022) and representation learning (Collins et al., 2021; 2022).
>
> The theoretical findings we derive in this regime, are of independent interest but also provide insights and serve as a starting point towards exploring broader, non-linear settings. Furthermore, a series of numerical results suggest that our proposed doubling scheme can provide significant straggler-resilience benefits beyond the linear representation case, although rigorously providing theoretical guarantees is substantially more challenging and lies outside of the scope of this work.

---

> ### Author Response · Authors · 2023-07-06
> **Rebuttal**
>
> $\textbf{References}$
>
> Seide, Fu, Droppo, Li, and Yu (2014). "1-bit stochastic gradient descent and its application to data-parallel distributed training of speech dnns", Fifteenth Annual Conference of the International Speech Communication Association.
>
> Smith, Forte, Ma, Takac, Jordan, and Jaggi (2016). "Cocoa: A general framework for communication efficient distributed optimization", arXiv preprint, arXiv:1611.02189.
>
> Alistarh, Grubic, Li, Tomioka, and Vojnovic (2017). "Qsgd: Communication-efficient sgd via gradient quantization and encoding", Advances in Neural Information Processing Systems.
>
> Jiang and Agrawal (2018). “A linear speedup analysis of distributed deep learning with sparse and quantized communication”, NeurIPS.
>
> Wangni, Wang, Liu, and Zhang (2018). “Gradient sparsification for communication-efficient distributed optimization”, NeurIPS.
>
> Basu, Karakus and Diggavi (2019). “Qsparse-local-SGD: Distributed SGD with quantization, sparsification, and local computations”, arXiv preprint, arXiv:1901.04359.
>
> Tripuraneni, Jin, and Jordan (2021). "Provable meta-learning of linear representations", International Conference on Machine Learning.
>
> Du, Hu, Kakade, Lee, and Lei (2020). "Few-shot learning via learning the representation, provably", arXiv preprint, arXiv:2002.09434, 2020.
>
> Sun, Narang, Gulluk, Oymak, and Fazel (2021). "Towards sample efficient overparameterized meta-learning", Advances in Neural Information Processing Systems.
>
> Maurer, Pontil, and Romera-Paredes (2016). "The benefit of multitask representation learning", Journal of Machine Learning Research.
>
> Jain, Rush, Smith, Song, and Thakurta (2021). "Differentially private model personalization", Advances in Neural Information Processing Systems.
>
> Shen, Ye, Kang, Hassani, and Shokri (2022). "Share your representation only: Guaranteed improvement of the privacy-utility tradeoff in federated learning", The Eleventh International Conference on Learning Representations.
>
> Collins, Hassani, Mokhtari, and Shakkottai (2021). "Exploiting shared representations for personalized federated learning", International conference on machine learning.
>
> Collins, Hassani, Mokhtari, and Shakkottai (2022). "Fedavg with fine tuning: Local updates lead to representation learning", Advances in Neural Information Processing Systems.

---

### Review · Reviewer_5MxQ · 2023-06-09

**Summary Of Contributions:**

The paper proposes an algorithm for personalized federated learning that tackles data/system heterogeneity and is resilient to stragglers. Specifically, personalization is attained by learning a global representation followed by user-specific parameters. Straggler-resilience is achieved by adaptively and disproportionally selecting participating clients over time. Theoretical results show that the proposed algorithm can attain near-optimal sample complexity and logarithmic speedup in certain settings. Experiments on several datasets show that the algorithm can attain better performance than baselines.

**Audience:**

Yes

**Broader Impact Concerns:**

The proposed algorithm may raise unfairness issues for devices with large computational time.

**Claims And Evidence:**

Yes

**Requested Changes:**

1. More experiments need to be conducted to validate the proposed algorithm, including:
(a) settings when the computational times of clients are not uniformly distributed, e.g., when stragglers are always the same set of clients, when certain classes are not accessible to the fastest clients, etc.
(b) in addition to the overall accuracy, the performance of local clients also needs to be reported.
(c) comparison with more baselines, especially those tackling personalization and heterogeneity.

2. Because the idea of using shared representation mapping and client-specific head parameters has been leveraged in many prior works, it is essential to highlight the novelty of the proposed algorithm.

3. The theoretical results are only for a special case with linear representation, I wonder whether it can provide some insights for non-linear cases.

**Strengths And Weaknesses:**

Strengths:

1. The paper studies important issues in federated learning: it considers data/system heterogeneity and aims the issues of stragglers. It is easy to follow and well-organized.
2. The paper conducts a theoretical analysis of the proposed algorithm to show the speedup guarantee.


Weakness:
1. The idea of using shared representation and client-specific head parameters for personalized FL is not new, e.g., see (Jiang & Lin, 2023, URL: https://openreview.net/forum?id=3aBuJEza5sq), (Zhu et al., 2022, URL: https://openreview.net/pdf?id=E4EE_ohFGz). As far as I can tell, the main contribution seems to be the adaptive participating scheme, which I think also has some issues as detailed next.

2. To tackle the issue of stragglers, the proposed algorithm only selects a subset of the fastest clients to participate in each round. This may raise several issues:
(a) In practical federated learning systems, the participation of clients is usually out of the central server's control. This significantly limits the adoption of the proposed algorithm in practice.
(b) While the proposed algorithm geometrically increases the number of participating nodes, certain nodes such as stragglers may never participate. This can be problematic especially when the data distribution of stragglers differs from the fast nodes (as these clients never got the chance to update their local head). As also indicated by the authors, such a disproportional selection of nodes raises fairness and accuracy concerns.

3. In the experiments, the computational time of all clients is determined uniformly (i.e., time follows an exponential distribution with parameter uniformly sampled from [1/M,1]). This setup is not comprehensive enough as it intentionally avoids the issue of 2(b) mentioned above. Moreover, only overall accuracy on all clients is reported. It is necessary to compare local accuracy for personalized FL. More experiments should be conducted (see next section).

---

> ### Author Response · Authors · 2023-07-06
> **Rebuttal**
>
> $\textbf{Q}$. Because the idea of using shared representation mapping and client-specific head parameters has been leveraged in many prior works (Jiang and Lin, 2022; Zhu et al., 2021), it is essential to highlight the novelty of the proposed algorithm.
>
> $\textbf{A}$. We thank the reviewer for bringing these interesting works (Jiang and Lin, 2022; Zhu et al., 2021) to our attention which we have included in our revised manuscript. As the reviewer correctly points out, prior works have utilized representation learning to address data heterogeneity in Federated Learning (Liang et al., 2020; Collins et al., 2021; Jiang and Lin, 2022). Specifically, (Collins et al., 2021) showed that FedRep can handle clients with heterogeneous data. Nevertheless, the element of system heterogeneity eludes these works and no straggler-resilience guarantees are provided. Similarly, a long line of work focuses on handling system heterogeneity in Federated Learning (Xie et al., 2019; Reisizadeh et al., 2019; Reisizadeh et al., 2022); however, theoretical results of these works rely heavily on some notion of data homogeneity across clients. Our paper is inspired by the work of (Collins et al., 2021) and is among the first studies that present a framework that can handle both data and system heterogeneity simultaneously while deriving theoretically proven speed-up guarantees.
>
> To clarify our contribution we point out that although each of the two aforementioned hurdles (data heterogeneity and system heterogeneity) has been individually the subject of study in the literature, handling simultaneously both of them gives rise to new challenges which necessitate taking a novel approach that goes beyond the mere combination of prior works. More precisely, despite some similarities, our work differs substantially from (Collins et al., 2021) with regard to the framework and the analysis, and takes aim at the development of a straggler-resilient method with provably improved performance in the presence of system heterogeneity. Specifically, in (Collins et al., 2021) access to infinite noiseless samples is assumed, whereas in our work only a finite number of noisy samples are available. In this more challenging setting we prove Theorem $4.5$ which captures the level of achievable accuracy and the rate at which the representation model improves as a function of the participating clients. We note that no similar result is provided in (Collins et al., 2021) and their analysis differs from ours in multiple dimensions: (i) They consider a fixed number of nodes throughout their method and (ii) their algorithm has access to infinite noiseless samples. Additionally, our work proves that FedRep-SRPFL achieves logarithmic speedup over FedRep in the presence of data and system heterogeneity (illustrated in Theorem $4.6$), a result that lies entirely outside the scope of (Collins et al., 2021).
>
> Furthermore, as we mention in Remark $3.3$ a participation scheme similar to ours was proposed in (Reisizadeh et al., 2022), however their analysis follows a different approach and their results apply to significantly more restrictive regimes. Specifically, in (Reisizadeh et al., 2022) the analysis relies heavily on deriving a connection between the ERM solutions of consecutive stages. In order to control the statistical accuracy of the corresponding ERM problems (i) data homogeneity across all clients is necessary and further (ii) clients who participate in early stages are required to remain active and connected to the server in all subsequent stages, maintaining fixed computational speeds throughout the whole training process. Furthermore, (iii) the results of their analysis hold only for strongly convex
> loss functions and (iv) their stage termination criterion requires the knowledge of the strong convexity parameter. The above restrictions are detrimental in the FL regime and severely undermine the applicability of the resulting algorithm. In our work we follow a different approach controlling -in terms of principal angle distance- a quantity analogous to statistical accuracy, therefore directly connecting the common representation (and overall solution) at every stage to the ground-truth representation. This novel approach allows our algorithm to accommodate (i) data heterogeneity, (ii) clients with dynamically changing speeds or equivalently clients that are replaced by new ones at every round, and (iii) non-convex loss functions. Additionally, a major part of our technical contribution focuses on (iv) analytically deriving the optimal number of rounds per stage, thus producing a simple and efficient doubling scheme.
>
> We kindly refer the reviewer to Theorems $4.5$ and $4.6$ as well as the corresponding Appendices A and B for a more extended exposition of our theoretical contributions. Thanks for your comment.

---

> ### Author Response · Authors · 2023-07-06
> **Rebuttal**
>
> $\textbf{Q}$. To tackle the issue of stragglers, the proposed algorithm only selects a subset of the fastest clients to participate in each round. This may raise several issues: (a) In practical Federated Learning systems, the participation of clients is usually out of the central server's control. This significantly limits the adoption of the proposed algorithm in practice.
>
> $\textbf{A}$. Throughout our method, the server selects the $n$ fastest nodes (at the corresponding stage) from a subset of $N$ available nodes. It is important to note that the subset of $N$ available clients is randomly chosen from a network of $M$ total clients, as is typically the case in most Federated Learning setups, and is completely beyond the control of the server. Furthermore, as mentioned in Remark $3.2$, the set of available nodes may change at each stage or round without affecting the resulting theoretical guarantees of our method. This makes our approach robust and applicable in a wide range of Federated Learning scenarios.
>
> We should also add that, our method does not require the $n$ fastest nodes among all $M$ nodes to share their information with the server. Instead, it requires the $n$ fastest nodes among the $N$ participating nodes (randomly selected and beyond the server's control) to send their updates to the server. As mentioned in Remark $3.1$, this mechanism can be easily implemented in practice by ensuring that the first $n$ models received by the server belong to the fastest clients in the participating set for the round. Thus, our scheme is applicable and provides strong theoretical guarantees without requiring any additional central coordination.
>
> We hope that our response clarifies that our method does not require any additional coordination between the server and the clients, nor does it require knowledge of the clients' computational speeds. Thank you for your comment.
>
>
> $\textbf{Q}$. (b) While the proposed algorithm geometrically increases the number of participating nodes, certain nodes such as stragglers may never participate. This can be problematic especially when the data distribution of stragglers differs from the fast nodes (as these clients never got the chance to update their local head). As also indicated by the authors, such a disproportional selection of nodes raises fairness and accuracy concerns.
>
> $\textbf{A}$. This is a very good question. We thank the reviewer for raising this point. In fact, in our first draft, we should have highlighted that the issue that the reviewer is referring to will not affect our theoretical guarantees. Next, we explain why ignoring some of the clients would not impact our theoretical results.
>
> More precisely, a careful examination of Theorem $4.5$ suggests that as long as sufficient data heterogeneity is present within the participating set (captured by Assumption $4.3$), the server can recover the ground-truth representation that is shared across all clients. As the reviewer correctly points out, the server may learn this representation before some of the clients update their local heads or even participate in the learning procedure. Essentially, this implies that learning representation does not necessitate access to the data of all clients. Instead, at each iteration, it only requires access to data points from a subset of clients with diverse tasks, as guaranteed by Assumption $4.3$. While involving more clients would enhance sample efficiency and expedite convergence, omitting certain clients during the learning process would not necessarily have a detrimental impact on representation learning.
>
> One might raise concerns about fairness or accuracy when ignoring certain clients, but it is important to highlight that such concerns are unfounded. This is because, after obtaining the ground-truth representation, it is shared with all clients in the system. Even if a client, denoted as $i$, did not participate or was not selected in the representation learning process, it can still optimize its low-dimensional head $w_i \in \mathbb{R}^k$ using its local samples through a few local updates (given that $k$ is a small constant). Consequently, the disproportional participation of clients in our mechanism does not create fairness or accuracy issues. Instead, the derivation of the ground-truth model benefits both the clients that have already participated in the learning procedure and new clients who choose to join the federated system at a later time. Thanks for raising this point. We have added a remark regarding this point after Theorem $4.6$ of the revised paper.

---

> ### Author Response · Authors · 2023-07-06
> **Rebuttal**
>
> $\textbf{Q}$. More experiments need to be conducted to validate the proposed algorithm, including: (a) settings when the computational times of clients are not uniformly distributed, e.g., when stragglers are always the same set of clients, when certain classes are not accessible to the fastest clients, etc.
>
> $\textbf{A}$. In our experiments we consider two different types of client speed configuration: (i) Fixed Computation Speeds where computational times are sampled once at the beginning of the process and remain fixed throughout the training and (ii) Dynamic Computation Speeds where new computational times are sampled at every round. In the former configuration the set of stragglers indeed remains the same and our doubling scheme substantially improves the performance of FedRep as illustrated in Figures $3,4,7$ and $8$.
>
> We point out that in order to accurately recover the ground-truth, common representation, our analysis heavily relies on Assumption $4.3$ which guarantees that the participating clients have sufficiently heterogeneous data. Our setting is well-suited for this condition with the client speeds being independent of the local data available to them. This is a natural assumption since the computational power of each client crucially depends on their device characteristics (battery, CPU, etc.), whereas any connection to their local data is unclear. More specifically, in most real-world settings, it is reasonable to assume that the similarity between the tasks of two clients (indicating data homogeneity) is independent of the difference or similarity in their computational power (representing system heterogeneity).
>
> That said, when there is a strong correlation between data and system heterogeneity, Assumption $4.3$ may not hold, which can be seen as a potential limitation of our work and an interesting future direction to explore. We have added a remark regarding this point after Assumption $4.3$ of the revised paper. Furthermore, in our revisited paper we have included experiments on the CIFAR10 dataset that explore the behavior of our method is such a regime. Specifically, clients are split into groups based on their speeds and each group is given access to data with specific labels. Our numerical results, illustrated in Figure $8$, indicate that the benefits of our doubling scheme persist, although to a lesser degree as the correlation between data and system heterogeneity grows.
>
> $\textbf{Q}$. (b) in addition to the overall accuracy, the performance of local clients also needs to be reported.
>
> $\textbf{A}$. A careful examination of Theorem $4.5$ suggest that as long as sufficient data heterogeneity is present within the participating set (captured by Assumption $4.3$), the server can recover the ground-truth representation that is shared across all clients. As the reviewer correctly points out, the server may learn this representation before some of the clients update their local heads or even participate in the learning procedure. We emphasize that obtaining the ground-truth representation is sufficient to guarantee personalized solutions of high quality for the whole network. Indeed, once a client $i$ receives the optimal ground-truth representation from the server, they can easily optimize their low dimensional head $w_i \in \mathbb{R}^k$ utilizing their local samples via a few local updates (since $k$ is a small constant). As a result, the local accuracy of the clients is of marginal importance as it  emanates for free from the global representation.

---

> ### Author Response · Authors · 2023-07-06
> **Rebuttal**
>
> $\textbf{Q}$. (c) comparison with more baselines, especially those tackling personalization and heterogeneity.
>
> $\textbf{A}$.  We currently provide comparisons with $8$ baselines that take aim at tackling either data heterogeneity (personalization) or system heterogeneity. Specifically, FedRep, LG-FedAvg, HF-MAML and FedAvg with fine-tuning address data heterogeneity, FLANP addresses system heterogeneity and LG-SRPFL and FLANP with fine-tuning tackle simultaneously both data and system heterogeneity. We would be happy to include any additional baselines indicated by the reviewer. We note however that although each of the aforementioned hurdles (data heterogeneity and system heterogeneity) has been individually the subject of study in the literature, the framework where both are simultaneously prevalent is not extensively explored and most of the methods that have been proposed are heuristics without strong theoretical guarantees.
>
> $\textbf{Q}$. The theoretical results are only for a special case with linear representation, I wonder whether it can provide some insights for non-linear cases.
>
> $\textbf{A}$. In the linear representation setting one aims to learn a low-dimensional subspace in which the ground-truth regressors for a collection of linear regression tasks lie. This model has been commonly used by numerous studies in the areas of meta-learning (Tripuraneni et al., 2021; Du et al., 2020; Sun et al., 2021), multitask learning (Maurer er al., 2016), model personalization (Jain et al., 2021; Shen et al., 2022) and representation learning (Collins et al., 2021; 2022).
>
> The theoretical findings we derive in this regime, are of independent interest but also provide insights and serve as a starting point towards exploring broader, non-linear settings. Furthermore, a series of numerical results suggest that our proposed doubling scheme can provide significant straggler-resilience benefits beyond the linear representation case, although rigorously providing theoretical guarantees is substantially more challenging and lies outside of the scope of this work.

---

> ### Author Response · Authors · 2023-07-06
> **Rebuttal**
>
> $\textbf{References}$
>
> Jiang, and Lin (2022). "Test-time robust personalization for federated learning", arXiv preprint, arXiv:2205.10920.
>
> Zhu, Xu, Chen, Konecny, Hard, and Goldstein (2021). "Diurnal or nocturnal? federated learning of multi-branch networks from periodically shifting distributions", International Conference on Learning Representations.
>
> Liang, Liu, Ziyin, Allen, Auerbach, Brent, Salakhutdinov, and Morency (2020). "Think locally, act globally: Federated learning with local and global representations", arXiv preprint, arXiv:2001.01523.
>
> Collins, Hassani, Mokhtari, and Shakkottai (2021). "Exploiting shared representations for personalized federated learning", International conference on machine learning.
>
> Xie, Koyejo, and Gupta (2019). "Asynchronous federated optimization", arXiv preprint, arXiv:1903.03934.
>
> Reisizadeh, Prakash, Pedarsani, and Avestimehr (2019). "Coded computation over heterogeneous clusters", IEEE Transactions on Information Theory.
>
> Reisizadeh, Tziotis, Hassani, Mokhtari, and Pedarsani (2022). "Straggler-resilient federated learning: Leveraging the interplay between statistical accuracy and system heterogeneity", IEEE Journal on Selected Areas in Information Theory.
>
> Tripuraneni, Jin, and Jordan (2021). "Provable meta-learning of linear representations", International Conference on Machine Learning.
>
> Du, Hu, Kakade, Lee, and Lei (2020). "Few-shot learning via learning the representation, provably", arXiv preprint, arXiv:2002.09434, 2020.
>
> Sun, Narang, Gulluk, Oymak, and Fazel (2021). "Towards sample efficient overparameterized meta-learning", Advances in Neural Information Processing Systems.
>
> Maurer, Pontil, and Romera-Paredes (2016). "The benefit of multitask representation learning", Journal of Machine Learning Research.
>
> Jain, Rush, Smith, Song, and Thakurta (2021). "Differentially private model personalization", Advances in Neural Information Processing Systems.
>
> Shen, Ye, Kang, Hassani, and Shokri (2022). "Share your representation only: Guaranteed improvement of the privacy-utility tradeoff in federated learning", The Eleventh International Conference on Learning Representations.
>
> Collins, Hassani, Mokhtari, and Shakkottai (2022). "Fedavg with fine tuning: Local updates lead to representation learning", Advances in Neural Information Processing Systems.

---

### Review · Reviewer_rm5K · 2023-06-14

**Summary Of Contributions:**

This paper studies the problem of straggler nodes in the context of federated learning with partial model personalization. More specifically, it proposes a sampling mechanism based on doubling in which:

* At each round, the server sends a model to $N$ clients.
* The server only waits for the fastest $n \leq N$ clients to return an update.
* The parameter $n$ is doubled at a specified cadence.

The authors then study the application of this "doubling mechanism" to the FedRep algorithm (Collins et al., 2021), which does partial model personalization. The authors study this algorithm theoretically in a "linear representation" setting, and show that under various assumptions (including a kind of incoherence across clients, a "full-rank" assumption, and an exponential distribution runtime assumption), that this incurs a logarithmic speed-up in runtime.

Finally, the paper does some comparisons on image datasets (ie. (F)EMNIST and CIFAR-10(0) datasets) and the SENT140 dataset. These experiments show that in many settings (varying amounts of heterogeneity, and varying communication costs), this FedRep + doubling algorithm converges faster than other methods, especially FedRep without the mechanism.

**Audience:**

Yes

**Claims And Evidence:**

Yes

**Requested Changes:**

## Critical Recommendations

1. Please clarify the contributions of this paper, especially disambiguating the doubling mechanism proposed and the representation learning and personalization approach of FedRep. Additionally, if there is any facet of the doubling mechanism that is intertwined with or reliant upon representation learning mechanisms (and is therefore not amenable to FedAvg), please discuss. Finally, please discuss obstacles (theoretically or otherwise) to studying FedAvg with the doubling mechanism (note that this might be a duplicate request with the previous sentence).
1. Please clarify the significance and limitations of Assumption 4.3. In particular, is this assumed to hold over all possible size $n$ sets simultaneously? If so, how does this impact the possible data heterogeneity possible, especially statistical correlation between data and system heterogeneity.
1. Please discuss why the initialization phase of Algorithm 2 is compatible with the general principles of federated learning, especially data minimization.

## Other Recommendations

1. Please clarify why the experiments do not use the "natural" client partitions of the FEMNIST and SENT140 datasets.
1. Please discuss how the learning rate for the optimizers were chosen, and why it is reasonable to use the same learning rate across all methods.

**Strengths And Weaknesses:**

My overall impression of the paper is that it has a good core (simple, in a postiive way) idea coupled with strong theoretical analysis. However, the paper has the following weaknesses.

1. Its claims in the abstract & introduction that overstate what the paper does.
1. The theoretical assumptions that are arguably too strong (as best as I can tell, though I would love to discuss with the authors about this).
1. An algorithmic choice that potentially goes in the face of the data minimization principle underlying federated learning.
1. The empirical results have some potential issues, especially related to how datasets are partitioned across clients, and a lack of reproducibility.

I discuss these in greater detail below.

## Strengths

The fundamental idea, the doubling mechanism, is interesting theoretically and empirically. The algorithm has a lot of intuitive appeal (ie. wait for the fastest $n$ clients to finish, and proceed), and its core simplicity is a virtue. In particular, the doubling mechanism can ostensibly be applied to any federated learning setting in which there are (1) sufficiently many clients available and (2) sufficiently many can participate in a given round. In particular, it is not actually tied to the FedRep method (though some of the wording in the first 2 sections obscures that fact, more on that below). The method can be applied to other methods, including FedAvg and LG-FedAvg (the latter of which is actually implemented in the empirical results).

Of course, simple mechanisms often still require nuanced theoretical analysis, and this is no difference. I think that the focus on the "linear representation" setting is actually a strength of the paper. It allows for much more fine-grained control over the optimization behavior, and therefore over the runtime speedups that the method can incur. Moreover, the theory involves a ton of great high-dimensional probability results that I enjoyed reading. I will caveat that i have not verified the theory 100%, though I believe that the theory holds **with the assumptions made in the main body** (more on that below).

For the empirical results, the authors do a good job of benchmarking a wide swath of methods. I do not believe that this is strictly necessary (as the paper's primary contributions are, in my opinion, theoretical) but this was nice to see nonetheless.

## Weaknesses

### Accomplishments versus Claims

One of the core themes of the abstract and introduction are that this work addresses both data heterogeneity and system heterogeneity simultaneously. From the introduction:
```
We propose Straggler-Resilient Personalized Federated Learning (SRPFL), an adaptive node-participation method which accommodates gradient-based subroutines and leverages ideas from representation learning theory to find personalized models in a federated and straggler resilient fashion.
```
This statement struck me as incongruous with what the paper is doing. The representation learning in Algorithms 1 and 2 are all from the FedRep algorithm from (Collins et al., 2021). This is freely discussed in later sections of the work, but section 1 completely removes this connection from the discussion.

More generally, I don't know that I agree that this paper really addresses data heterogeneity, as prior work has shown that FedRep does that. Rather, the contribution of this work is the doubling mechanism, which could be applied to any FL algorithm (whether or not it uses representation learning). This brings me to a fundamental question:

**Question:** Why does this paper couple the study of the doubling mechanism to the FedRep algorithm? Are there theoretical obstacles to studying algorithms such as FedAvg when using this mechanism? Are there aspects of FedRep that specifically motivated the doubling mechanism?

As it stands, I would greatly prefer that the authors focus on what they do provide and study. The question of data heterogeneity is also related to the following section, as I believe that the theoretical assumptions help mitigate issues at the intersection of data & system heterogeneity.

### Theoretical Assumptions

The authors make a number of assumptions, 4.1 - 4.4, that I think are worth considering. Many of these appear in (Collins et al., 2021). However, they take on new meaning under the doubling mechanism proposed by this paper.

Primarily, I am concerned with Assumption 4.3 (aka "client diversity"). This assumption states that if you take $n$ rows of a matrix $W^*$ (each row of which corresponds to a client), then you get a full rank matrix. As best as I can tell, this is the same $n$ from the doubling mechanism. This is the core issue: In work like (Collins et al., 2021), it is reasonable to assume that the $n$ rows are drawn randomly, and therefore this statement need only hold with high probability. In this work, $n$ is not random, it is simply the $n$ clients who finish fastest. The work seems to make no assumptions on this distribution. Thus, it is reasonable to imagine settings where the data heterogeneity is tied to the system heterogeneity, in which case this assumption is **much** stronger than it seems on first glance.

For example, if we consider a simple setting with 3 clients, and $n = 2$, and one of the clients is always the slowest, then this assumption seems strange in practice. In fact, you can construct counter-examples in this vein to Theorem 4.5, but these counter-examples fail Assumption 4.3 if you assume that it must hold for all possible subset of size $n$.

As discussed above, this relates back to data heterogeneity. As best as I can tell, this "full rank" assumption limits the amount of data heterogeneity possible. This is not necessarily bad, but it's certainly a limitation of the paper and probably deserves more discussion. I would love feedback from the authors about this.

### Data Minimization and SVD Initialization

One of the core tenets of federated learning (and indeed, one of the reasons it exists as a computational paradigm) is data minimization - the server should have access to no more data than is completely necessary. However, the initialization phase of Algorithm 2 involves sending sums of outer products of data. This seems potentially (not definitely) problematic from a data minimization perspective. I am no expert on privacy mechanisms for distributed SVD computations, so it is possible that there is some way that this could be integrated with something like differential privacy. I do not think that this integration is necessary, but I do believe that the authors need to discuss why this initialization phase is not tantamount to sending data in the clear to the server. In short, why is this initialization operation acceptable in the context of federated learning? There is no single answer to this, but I think it warrants discussion.

### Empirical Investigation and Datasets

I will preface this with saying that the empirical investigations are not the core of this paper, at least in my mind. That being said, they exhibit some characteristics that unfortunately reduces how convincing they are. First and foremost, two of the datasets that the paper mentions, FEMNIST and SENT140, have a natural federated structure, as discussed in the Leaf benchmark (Caldas et al., 2017). However, a close read of this paper suggests that this was not used, and instead the data was partitioned by some random process. Can the authors comment on why this natural federated structure was not used?

Second, I will mention briefly that the choice of optimizer hyperparameter seems somewhat arbitrary to me. In particular, all methods use the same learning rate (whether or not they do partial model participation or global optimization), which I believe needs some form of justification. On a more minor note, this work says that it sets hyperparmeters following the work of (Collins et al., 2021), but this doesn't seem to be the case. The latter uses SGD with momentum and a learning rate of 0.5, while the former uses SGD with a learning rate of 0.01.

These issues are not huge, but they are things that stuck out to me upon reading closely.

---

> ### Author Response · Authors · 2023-07-06
> **Rebuttal**
>
> $\textbf{Q}$.  Please clarify the contributions of this paper, especially disambiguating the doubling mechanism proposed and the representation learning and personalization approach of FedRep.
>
> $\textbf{A}$. As the reviewer correctly points out, prior works have utilized representation learning to address data heterogeneity in Federated Learning (Liang et al., 2020; Collins et al., 2021; Jiang and Lin, 2022). Specifically, (Collins et al., 2021) showed that FedRep can handle clients with heterogeneous data. Nevertheless, the element of system heterogeneity eludes these works and no straggler-resilience guarantees are provided. Similarly, a long line of work focuses on handling system heterogeneity in Federated Learning (Xie et al., 2019; Reisizadeh et al., 2019; Reisizadeh et al., 2022); however, theoretical results of these works rely heavily on some notion of data homogeneity across clients. Our paper is inspired by the work of (Collins et al., 2021) and is among the first studies that present a framework that can handle both data and system heterogeneity simultaneously while deriving theoretically proven speed-up guarantees.
> We agree with the reviewer that our first draft might have erroneously given the impression that the idea of studying data heterogeneity via representation learning is one of the contributions of this paper. For this reason, we have carefully revised the abstract and introduction of our paper to address this concern.
> To clarify our contribution we point out that although each of the two aforementioned hurdles (data heterogeneity and system heterogeneity) has been individually the subject of study in the literature, handling simultaneously both of them gives rise to new challenges which necessitate taking a novel approach that goes beyond the mere combination of prior works. More precisely, despite some similarities, our work differs substantially from (Collins et al., 2021) with regard to the framework and the analysis, and takes aim at the development of a straggler-resilient method with provably improved performance in the presence of system heterogeneity. Specifically, in (Collins et al., 2021) access to infinite noiseless samples is assumed, whereas in our work only a finite number of noisy samples are available. In this more challenging setting we prove Theorem $4.5$ which captures the level of achievable accuracy and the rate at which the representation model improves as a function of the participating clients. We note that no similar result is provided in (Collins et al., 2021) and their analysis differs from ours in multiple dimensions: (i) They consider a fixed number of nodes throughout their method and (ii) their algorithm has access to infinite noiseless samples. Additionally, our work proves that FedRep-SRPFL achieves logarithmic speedup over FedRep in the presence of data and system heterogeneity (illustrated in Theorem $4.6$), a result that lies entirely outside the scope of (Collins et al., 2021).
>
> Furthermore, as we mention in Remark 3.3 a participation scheme similar to ours was proposed in (Reisizadeh et al., 2022), however their analysis follows a different approach and their results apply to significantly more restrictive regimes. Specifically, in (Reisizadeh et al., 2022) the analysis relies heavily on deriving a connection between the ERM solutions of consecutive stages. In order to control the statistical accuracy of the corresponding ERM problems (i) data homogeneity across all clients is necessary and further (ii) clients who participate in early stages are required to remain active and connected to the server in all subsequent stages, maintaining fixed computational speeds
> throughout the whole training process. Furthermore, (iii) the results of their analysis hold only for strongly convex loss functions and (iv) their stage termination criterion requires the knowledge of the strong convexity parameter. The above restrictions are detrimental in the FL regime and severely undermine the applicability of the resulting algorithm. In our work we follow a different approach controlling -in terms of principal angle distance- a quantity analogous to statistical accuracy, therefore directly connecting the common representation (and overall solution) at every stage to the ground-truth representation. This novel approach allows our algorithm to accommodate (i) data heterogeneity, (ii) clients with dynamically changing speeds or equivalently clients that are replaced by new ones at every round, and (iii)non-convex loss functions. Additionally, a major part of our technical contribution focuses on (iv) analytically deriving the optimal number of rounds per stage, thus producing a simple and efficient doubling scheme.
> We kindly refer the reviewer to Theorems $4.5$ and $4.6$ as well as the corresponding Appendices A and B for a more extended exposition of our theoretical contributions. Thanks for your comment.

---

> > ### Author Response · Authors · 2023-07-06
> > **Rebuttal**
> >
> > $\textbf{Q}$. Please clarify why the experiments do not use the "natural" client partitions of the FEMNIST and SENT140 datasets.
> >
> > $\textbf{A}$. In our experiments, we opt to deviate from the default client partition of the FEMNIST and Sent140 datasets and instead manually assign data to clients. For FEMNIST we follow the same setting as in (Collins et a., 2021) with the exception that we equally assign $150$ samples to each client. For Sent140 we partition the data both in homogeneous and in heterogeneous fashion, while making sure that the sizes of the local datasets exhibit high concentration.
> >
> > Manually partitioning the datasets (i) gives us control over the level of data heterogeneity enabling us to explore both heterogeneous and homogeneous instances, (ii) allows us to avoid extreme cases where clients have very few samples (most clients in the default Sent140 partition are assigned less than $3$ samples) (iii) permits the (approximately) even allocation of samples to clients thus deriving a clear alignment between theory and experiments. To further elaborate on (iii) we point out that although it is not strictly necessary, assuming that all clients have the same number of local samples simplifies the exposition of our analysis and permits the clear demonstration of the main ideas of our work. Our results readily extend to clients with different number of samples, however the implementation of our algorithm becomes more tedious. Indeed, in such a setting consecutive participating sets should be selected such that the cumulative number of samples across participating clients (approximately) doubles from one stage to the next. This is a crucial condition that allows our algorithm to progressively improve the accuracy of the representation model.
> >
> > In most of our experiments the local datasets of the clients are of equal size. However, to illustrate the robustness of our method we provide numerical results in the FEMNIST and the SENT140 datasets, presented in Figure $8$, with disproportional sample allocation. Specifically, in the FEMNIST dataset each client is allocated $(100+u_i)$ samples where $u_i$ is a random variable uniformly distributed in $[0,50]$. In Sent140 we assign samples to clients following the log-normal distribution deriving a more challenging setting. In both cases the benefits of our doubling scheme persist, however unsurprisingly the performance of our algorithm degrades as the data allocation becomes more disproportional.
> >
> > $\textbf{Q}$. Please discuss how the learning rate for the optimizers were chosen, and why it is reasonable to use the same learning rate across all methods.
> >
> > $\textbf{A}$. Thanks for raising this point. In the revised paper, we have addressed this issue. For the implementation of FedRep and FedRep-SRPFL we follow (Collins et al., 2021) and use SGD with momentum with the same momentum parameter $0.5$ and local learning rate $0.1$. Further, similarly to (Collins et al., 2021) we set the local learning rate to $0.1$ for all other methods under consideration, which obtains optimal performance. In our revised  manuscript we have rephrased Appendix C to more clearly illustrate the details of our setting.

---

> > > ### Comment · Reviewer_rm5K · 2023-07-06
> > > **Regarding synthetic datasets**
> > >
> > > A quick follow-up question here: You already use CIFAR-10 and CIFAR-100 for synthetic partitioning that you can control the amount of partitioning explicitly. Isn't there some benefit to using non-synthetic partitions as well? In FEMNIST most clients have roughly the same number of examples (and repeating operations, for example, could be used if you wanted to ensure every client has equal amounts of data). I'm not trying to get you to run more experiments, I'm just honestly trying to assess what the experimental benefit is here. Given that much of the work (and your rebuttal above) is grounded in "realistic" assumptions on FL, it seems to me that there's only upside to investigating on more realistic datasets. This is a minor point though, and thanks for the clarifications.

---

> > > > ### Author Response · Authors · 2023-07-16
> > > > **Regarding synthetic datasets**
> > > >
> > > > We note that in the default partition of the FEMNIST dataset there is a mismatch between the labels of the local training sets and the labels of the corresponding local test sets. Put simply, the labels of the samples where each client is training their model on, is (to a certain extent) different from the labels of the samples where the same model is tested on. Since the focus of our work lies primarily on personalized methods this misalignment leads to an unfair evaluation of the derived personalized models and an inaccurate comparison between our methods.
> > > >
> > > > Manually assigning data to clients permits the removal of this mismatch by carefully aligning the training set and the test set for every client.
> > > > A different direction would involve training a global model on the training set and subsequently, personalize it utilizing the test set. Although, this approach would also resolve the misalignment issue of the FEMNIST dataset, it would differ from the methodology we have followed in our previous experiments. That said, we would be happy to receive additional feedback and include more experiments as the reviewer sees fit.

---

> > ### Comment · Reviewer_rm5K · 2023-07-06
> > **Regarding the paper's contributions**
> >
> > Thank you for the clarification! I believe that much of what you discuss in your statement about what the contributions of this work are, especially relative to (Collins et al., 2021) should be injected into the paper, which it sounds like you have done to some extent in a revised version. I also think that disambiguating algorithm development and theoretical contributions is important! For example, the fact that (Collins et al., 2021) does not consider finite-samples is important and should be highlighted! Conversely, the fact that you use the FedRep algorithm as a backbone (loosely speaking) for this work is not a drawback, but it is worth being explicit about.

---

> > > ### Author Response · Authors · 2023-07-16
> > > **Regarding the paper's contributions**
> > >
> > > To address your concern we have carefully revisited our abstract and introduction to better convey the contributions of our work. Furthermore, we have included Remark $4.2$ which highlights the differences between our work and (Collins et al., 2021). Thank you for your comment.

---

> ### Author Response · Authors · 2023-07-06
> **Rebuttal**
>
> $\textbf{Q}$. Additionally, if there is any facet of the doubling mechanism that is intertwined with or reliant upon representation learning mechanisms (and is therefore not amenable to FedAvg), please discuss. Finally, please discuss obstacles (theoretically or otherwise) to studying FedAvg with the doubling mechanism (note that this might be a duplicate request with the previous sentence).
>
> $\textbf{A}$. This is an excellent point. An important component of our doubling mechanism lies in the utilization of solutions from early stages (when some subset of $n$ nodes participates) as a warm start for subsequent stages (when a subset of $2n$ nodes, with potentially no overlap with the previous subset, participates). In the presence of data heterogeneity connecting the optimal solutions of consecutive stages is far from trivial and is, in fact, a major reason why theoretical results from prior work that utilized similar doubling schemes (Reisizadeh et al., 2022) were restricted to data-homogeneous regimes. To illustrate the hurdle introduced by data heterogeneity it is enough to consider subsequent subsets of participating clients with optimal solutions that differ substantially from each other. It is clear that obtaining a good quality solution for the earlier participating subset could serve as a poor starting point for the following stage. To overcome this challenge our analysis is crucially intertwined with the representation learning framework. In our setting, the presence of a shared, global representation serves as "common ground" across data-heterogeneous clients and allows us to show that intermediate solutions constitute sufficiently good starting points and substantial progress is achieved between stages. This intuition justifies the choice of FedRep as a suitable subroutine for our doubling scheme.
>
> Despite our analysis being tailored for FedRep, we expect similar theoretical results to hold for other subroutines that fall into the framework of common representation (our numerical results suggest that indeed Local-Global FedAvg enjoys similar benefits by the utilization of our doubling scheme). On the other hand, vanilla FedAvg is a general-purpose algorithm not designed for representation learning which suggests its incompatibility with our doubling mechanism. Recently, however, Collins et al., (2022) showed that FedAvg (with a sufficient number of local updates) recovers the ground-truth representation in the case of multi-task linear regression, casting FedAvg a potential candidate subroutine for our method. That said, further investigation is needed to formally prove the claim that FedAvg with adaptive node participation, as presented in our paper, can achieve the same theoretical guarantees as the FedRep method discussed in our study. Thanks for raising this point.

---

> > ### Comment · Reviewer_rm5K · 2023-07-06
> > **Regarding FedRep versus FedAvg**
> >
> > Thank you for the clarification on the theoretical difficulties of studying FedAvg with a doubling mechanism. This sounds (at least to me) like a good discussion in the paper, especially as it relates to the (Collins et al., 2022) paper.

---

> > > ### Author Response · Authors · 2023-07-16
> > > **Regarding FedRep versus FedAvg**
> > >
> > > In Remark $4.10$ of our revisited manuscript we have incorporated the above discussion  emphasizing on the compatibility of FedAvg with our doubling mechanism due to (Collins et al., 2022). Thanks for raising this point.

---

> ### Author Response · Authors · 2023-07-06
> **Rebuttal**
>
> $\textbf{Q}$. Please clarify the significance and limitations of Assumption $4.3$. In particular, is this assumed to hold over all possible size $n$
>  sets simultaneously? If so, how does this impact the possible data heterogeneity possible, especially statistical correlation between data and system heterogeneity.
>
> $\textbf{A}$. We thank the reviewer for their diligent study of our work and the insightful comments they provided with regard to Assumption $4.3$. Indeed, in order for Theorem $4.5$ to guarantee that sufficient progress is achieved at every round, the optimal heads of the participating clients should span $\mathbb{R}^k$ (to avoid any misunderstanding in our revised manuscript we clarify that Assumption $4.3$ needs to hold for any subset of size greater or equal to $n_0$). From a purely theoretical perspective Assumption $4.3$ is almost identical to Assumption $2$ in (Collins et al., 2021) since the former assumes that taking any $n_0$ rows from $\frac{1}{\sqrt{n}} \boldsymbol{W}^*$ results in a matrix with positive minimum singular value, whereas the latter assumes that taking any $rn$ rows (with $rn$ being a fraction of $n$) would also result in a matrix with positive minimum singular value.
>
> In order to accurately recover the ground-truth, common representation, our analysis heavily relies on Assumption $4.3$ which guarantees that the participating clients have sufficiently heterogeneous data. Our setting is well-suited for this condition with the client speeds being independent of the local data available to them. This is a natural assumption since the computational power of each client crucially depends on their device characteristics (battery, CPU, etc.), whereas any connection to their local data is unclear. More specifically, in most real-world settings, it is reasonable to assume that the similarity between the tasks of two clients (indicating data homogeneity) is independent of the difference or similarity in their computational power (representing system heterogeneity).
>
> That said, the reviewer is absolutely correct that when there is a strong correlation between data and system heterogeneity, Assumption $4.3$ may not hold, which can be seen as a potential limitation of our work and an interesting future direction to explore. We have added a remark regarding this point after Assumption $4.3$ of the revisited paper. Furthermore, in our revisited paper we have included experiments on the CIFAR10 dataset that explore the behavior of our method is such a regime. Specifically, clients are split into groups based on their speeds and each group is given access to data with specific labels. Our numerical results, illustrated in Figure $8$, indicate that the benefits of our doubling scheme persist, although to a lesser degree as the correlation between data and system heterogeneity grows.
>
> We note that in all of our experiments it suffices to initiate FedRep-SRPFL with participating subset size $n=10$ to consistently exhibit superior performance compared to the rest of the baselines.

---

> > ### Comment · Reviewer_rm5K · 2023-07-06
> > **Regarding Assumption 4.3**
> >
> > Thank you for the clarification! The note that you mention sounds great - I think it's worth emphasizing the limits of the assumption. On a more minor note, you say above that "in most real-world settings, it is reasonable to assume that the similarity between the tasks of two clients...is independent of the difference or similarity in their computational power." I would be very careful about such a statement, as I don't think it's obviously true! As a really basic example, different types of phones are popular in different parts of the world (due to, e.g., local manufacturing, price differences, etc.). That means that computation power *could* correlate with language usage, even if we only restrict to a single language! This is all to say - the actual characteristics of "realistic" FL data are largely unknown (which is to some degree intended), so I tend to be very cautious about making statements about these characteristics.

---

> > > ### Author Response · Authors · 2023-07-16
> > > **Regarding Assumption 4.3**
> > >
> > > We thank the reviewer for providing this insightful example and for raising this point. We have carefully reworded Remark $4.5$ excluding any presumptive statements regarding the connection between data and system heterogeneity of the clients.

---

> ### Author Response · Authors · 2023-07-06
> **Rebuttal**
>
> $\textbf{Q}$. Please discuss why the initialization phase of Algorithm $2$ is compatible with the general principles of Federated Learning, especially data minimization.
>
> $\textbf{A}$. The initialization phase of Algorithm $2$ is crucial for our method since it obtains a representation model that is sufficiently close to the ground-truth representation. This is a necessary starting condition that ensures the convergence of our algorithm and it has been used for almost all prior works in the context of linear representation learning.  In fact, deriving such an accurate estimator is a recurring theme in numerous settings with deep connections to our problem: matrix completion (Jain et al., 2013),  multi-task learning (Tripuraneni et al., 2020), federated learning with common representation (Collins et al., 2021; Jain et al., 2021), etc.
>
> In the framework of federated learning, clients are required to send local information. Note that this one-time exchange of information  between the clients and the server provides very limited information about the data points of each client as it simply reveals ``the sum of the outer products of their samples''. That said, the reviewer is right that such an exchange of information could potentially undermine privacy and contribute to information leakage. Indeed, one can utilize tools developed in the Differential Privacy (DP) literature to preserve the privacy of the users, while providing sufficient information for the server to initialize the algorithm. In fact, DP has been extensively studied in the context of federated learning deriving a series of results on the interplay between accuracy and local privacy (Shokri and Shmatikov, 2015; McMahan et al., 2017a, MAcMahan et al.,2017b). Although, one research direction aims at providing only record-level privacy (Chaudhuri et al., 2011; Hu et al., 2020) the majority of the studies focus on client/task-level privacy (Geyer et al., 2017; Li et al., 2019; Levy t al., 2021; Kairouz et al., 2021; Hu et al., 2021; Bietti et al., 2022), a notion suitable for our setting.
>
> Despite being orthogonal to our work, it is clear that implementing the initialization phase of Algorithm $2$ utilizing differential privacy techniques would benefit our method enhancing its robustness. Indeed, this has been the topic of recent works where (Jain et al., 2021; Shen et al., 2022) proposed variants of this initialization that can be used as a black box with strong privacy guarantees. We have added a remark regarding this point after Theorem $4.7$ of the revised paper.

---

> > ### Comment · Reviewer_rm5K · 2023-07-06
> > **Regarding the initialization phase**
> >
> > I'm not sure that I agree that sending the sum of outer products of their samples is limited - I believe it could be true, this just strikes me as an argument that needs some backing. As for the reference on DP - most of the references you give are on DP for SGD-like algorithms. The sum of outer products is a different thing, right? Some kind of differentially private SVD is probably a better grounding point for how one might get privacy here. I would certainly be interested to learn of any paper that does user-level DP with SVD computations!
> >
> > To reiterate - I am not suggesting that you solve this question. I just think that this is an important note for the reader! A lot of FL work uses DP implicitly as a "in the future this method could have formal privacy protections", and personally, the initialization phase was new enough to me that I would appreciate such a note on how one *might* inject formal privacy there (or if it's just fundamentally an open question, which is also a really useful note!).

---

> > > ### Author Response · Authors · 2023-07-16
> > > **Regarding the initialization phase**
> > >
> > > Although most of the references we provided consider Differential Privacy (DP) for SGD-like algorithms we point out that the works of (Jain et al., 2021; Shen et al., 2022) include initializations (similar to ours) implemented in private fashion. More precisely, Jain et al., (2021) took the first steps towards this direction in a slightly different setting providing the first DP guarantees (Lemma $4.5$, Jain et al., 2021) for such an initialization process. Subsequently, Shen et al., (2022) took aim specifically at the DP implementation of the FedRep initialization phase (the same process we consider in our work), a procedure that as the reviewer correctly noted requires DP SVD computations. The authors utilized the noisy power method coupled with cross-validation techniques to derive strong user-level DP guarantees (Corollary $F.1.$, Shen et al., 2022). Following the reviewer's recommendation we have included this note in Remark $4.11$ of our revisited manuscript. Thank you for raising this point.

---

> ### Author Response · Authors · 2023-07-06
> **Rebuttal**
>
> $\textbf{References}$
>
> Liang, Liu, Ziyin, Allen, Auerbach, Brent, Salakhutdinov, and Morency (2020). "Think locally, act globally: Federated learning with local and global representations", arXiv preprint, arXiv:2001.01523.
>
> Collins, Hassani, Mokhtari, and Shakkottai (2021). "Exploiting shared representations for personalized federated learning", International conference on machine learning.
>
> Jiang, and Lin (2022). "Test-time robust personalization for federated learning", arXiv preprint, arXiv:2205.10920.
>
> Xie, Koyejo, and Gupta (2019). "Asynchronous federated optimization", arXiv preprint, arXiv:1903.03934.
>
> Reisizadeh, Prakash, Pedarsani, and Avestimehr (2019). "Coded computation over heterogeneous clusters", IEEE Transactions on Information Theory.
>
> Reisizadeh, Tziotis, Hassani, Mokhtari, and Pedarsani (2022). "Straggler-resilient federated learning: Leveraging the interplay between statistical accuracy and system heterogeneity", IEEE Journal on Selected Areas in Information Theory.
>
> Collins, Hassani, Mokhtari, and Shakkottai (2022). "Fedavg with fine tuning: Local updates lead to representation learning", Advances in Neural Information Processing Systems.
>
> Jain, Netrapalli, and Sanghavi (2013). "Low-rank matrix completion using alternating minimization", Proceedings of the forty-fifth annual ACM symposium on Theory of computing.
>
> Tripuraneni, Jin, and Jordan (2021). "Provable meta-learning of linear representations", International Conference on Machine Learning.
>
> Jain, Rush, Smith, Song, and Thakurta (2021). "Differentially private model personalization", Advances in Neural Information Processing Systems.
>
> Shokri, and Shmatikov (2015). "Privacy-preserving deep learning", in Proceedings of the 22nd ACM SIGSAC conference on computer and communications security.
>
> McMahan, Moore, Ramage, Hampson, and Aguera y Arcas (2017a), "Communication-efficient learning of deep networks from decentralized data", in Artificial intelligence and statistics.
>
> McMahan, Ramage, Talwar, and Zhang (2017b). "Learning differentially private
> recurrent language models", arXiv preprint, arXiv:1710.06963.
>
> Chaudhuri, Monteleoni, and Sarwate (2011). "Differentially private empirical
> risk minimization", Journal of Machine Learning Research.
>
> Hu, Guo, Li, Pei, and Gong (2020). "Personalized federated learning with differential privacy", IEEE Internet Things J.
>
> Geyer, Klein, and Nabi (2017). "Differentially private federated learning: A
> client level perspective", CoRR.
>
> Li, Khodak, Caldas, and Talwalkar (2019). "Differentially private meta-learning." arXiv preprint, arXiv:1909.05830.
>
> Levy, Sun, Amin, Kale, Kulesza, Mohri, and Suresh (2021). "Learning with user-level privacy", Advances in Neural Information Processing Systems.
>
> Kairouz, McMahan, Song, Thakkar, Thakurta, and Xu (2021). "Practical and private (deep) learning without sampling or shuffling", in International Conference on Machine Learning.
>
> Hu, Wu, and Smith (2021). "Private multi-task learning: Formulation
> and applications to federated learning", arXiv preprint, arXiv:2108.12978.
>
> Bietti, Wei, Dudik, Langford, and Wu (2022). "Personalization improves privacy-accuracy tradeoffs in federated learning", International Conference on Machine Learning.
>
> Shen, Ye, Kang, Hassani, and Shokri (2022). "Share your representation only: Guaranteed improvement of the privacy-utility tradeoff in federated learning", The Eleventh International Conference on Learning Representations.

---

### Decision · Action_Editors · 2023-09-07

**Recommendation:** Accept as is

**Comment:**

All reviewers were ultimately in favor of this work appearing in TMLR---they agreed that this was an interesting approach, and appreciated the simplicity of the procedure and corresponding theoretical analysis. The reviewers also suggested several useful improvements to the paper, including more clearly explaining the main contributions, discussing potential limitations of the assumptions made (particularly Assumption 4.3), and including experiments in additional regimes including in homogeneous settings as well as scenarios when there may be correlations between data/systems heterogeneity. The authors have already thoughtfully addressed many of these suggestions in their revision, though we would suggest that they take a final pass to ensure that the feedback has been thoroughly incorporated---particularly in terms of clearly explaining key assumptions and potential limitations of the approach in addition to its benefits (e.g., in terms of data minimization/privacy, correlated heterogeneity, etc).

**Audience:**

As noted by the reviewers, the general area of systems heterogeneity is an interesting direction with federated learning and would be of interest to the TMLR audience.

**Claims And Evidence:**

This work explores issues of systems heterogeneity in federated settings and proposes a personalized FL solution where sampling is based on the computational capabilities of each client. The key ideas are well-supported through theoretical analyses and empirical studies.